# AN IMPROVED MODEL-FREE DECISION-ESTIMATION COEFFICIENT WITH APPLICATIONS IN ADVERSARIAL MDPS

**Haolin Liu**[1]*, **Chen-Yu Wei**[1], **Julian Zimmert**[2]
[1]University of Virginia, [2]Google Research
{srs8rh, chenyu.wei}@virginia.edu, zimmert@google.com

## ABSTRACT

We study decision making with structured observation (DMSO). Previous work [FKQR21, FGH23] has characterized the complexity of DMSO via the decision-estimation coefficient (DEC), but left a gap between the regret upper and lower bounds that scales with the size of the model class. To tighten this gap, [FGQ$^+$23] introduced optimistic DEC, achieving a bound that scales only with the size of the value-function class. However, their optimism-based exploration is only known to handle the stochastic setting, and it remains unclear whether it extends to the adversarial setting.

We introduce Dig-DEC, a model-free DEC that removes optimism and drives exploration purely by information gain. Dig-DEC is always no larger than optimistic DEC and can be much smaller in special cases. Importantly, the removal of optimism allows it to handle adversarial environments without explicit reward estimators. By applying Dig-DEC to hybrid MDPs with stochastic transitions and adversarial rewards, we obtain the first *model-free* regret bounds for *hybrid* MDPs with *bandit* feedback under linear reward and several *general* transition structures, resolving the main open problem left by [LWZ25].

We also improve the online function-estimation procedure in model-free learning: For average estimation error minimization, we refine [FGQ$^+$23]'s estimator to achieve sharper concentration, improving their regret bounds from $T^{\frac{3}{4}}$ to $T^{\frac{2}{3}}$ (on-policy) and from $T^{\frac{5}{6}}$ to $T^{\frac{7}{9}}$ (off-policy). For squared error minimization in Bellman-complete MDPs, we redesign their two-timescale procedure, improving the regret bound from $T^{\frac{2}{3}}$ to $\sqrt{T}$. This is the first time a DEC-based method achieves performance matching that of optimism-based approaches [JLM21, XFB$^+$23] in Bellman-complete MDPs.

## 1 INTRODUCTION

[FKQR21, FGH23] developed the framework of decision-estimation coefficient (DEC) that characterizes the complexity of general online decision making problems and provides a general algorithmic principle called Estimation-to-Decision (E2D). In the state-of-the-art result by [FGH23], regret lower and upper bounds are established with a gap of $\log |\mathcal{M}|$, where $\mathcal{M}$ is the model class where the underlying true model lies. This $\log |\mathcal{M}|$ reflects the price of *model estimation*. Essentially, the lower bound in [FGH23] only captures the complexity of decision-making / exploration, while the upper bound additionally includes the complexity of model estimation. Since E2D is a model-based algorithm that learns over models, it necessarily incurs this cost of model estimation.

On the other hand, a large class of existing reinforcement learning (RL) algorithms are model-free value-based algorithms, which only estimate value functions. To better capture the decision-making complexity in this case, [FGQ$^+$23] proposed a variant of E2D, called optimistic E2D, that achieves a regret upper bound characterized by the complexity measure called optimistic DEC. However, unlike the model-based DEC/E2D framework [FKQR21, FGH23] which drives exploration only

---

*Alphabetical order

through information gain, optimistic DEC/E2D leverages the *optimism* principle to drive exploration, which may not be fundamental and could lead to sub-optimal performance in certain cases. Overall, the precise tradeoff between model estimation complexity and decision-making complexity, along with the gap between upper and lower bounds, remain largely unsolved.

A parallel line of reserach seeks to relax the assumption that the environment remains stationary. [FRSS22] and [XZ23] studied the pure adversarial setting where the environment can choose a different model in every round. In this case, their algorithms only estimate the optimal policy and the price of estimation becomes $\log |\Pi|$ where $\Pi$ is the policy class. In such pure adversarial environment, however, the decision-making complexity could become prohibitively high and is often vacuous in Markov decision processes (MDPs). A simpler and more tractable setting is the that of *hybrid* MDPs where the transition is stochastic but the reward is adversarial. This setting has been studied in various settings: tabular MDPs [NGSA13, RM19, JJL$^+$20, SERM20], linear (mixture) MDPs [LWL21, DLWZ23, SKM23, LWZ24, KZWL23, LZZ24], and low-rank MDPs [ZYW$^+$24, LMWZ24]. The work of [LWZ25] first leveraged the DEC framework to obtain results for *bilinear classes*. However, they only gave a model-based algorithm (incurring large estimation error) and a model-free algorithm that requires full-information reward feedback, leaving the model-free bandit case open.

We provide a unified framework that advances both directions discussed above:

- In the stochastic setting, we introduce a new model-free DEC notion, Dig-DEC, that improves over the optimistic DEC of [FGQ$^+$23]. Our approach does not rely on the optimism principle, but adheres more closely to the general idea of DEC that drives exploration purely with information gain. For canonical settings such as bilinear classes or Bellman-complete MDPs with bounded Bellman eluder dimension or coverability, we recover their complexities with improved $T$-dependence in the regret, while in some constructed settings, the improvement can be arbitrarily large.
- We establish the first sublinear regret for *model-free* learning in *hybrid* bilinear classes and Bellman-complete coverable MDPs with linear reward and bandit feedback, resolving the open question in [LWZ25].
- We improve the online function estimation procedure both in the case of average estimation error and squared estimation error. This allows us to improve the $T^{\frac{3}{4}}/T^{\frac{5}{6}}$ regret of [FGQ$^+$23] to $T^{\frac{2}{3}}/T^{\frac{7}{9}}$ in the former case, and improve the $T^{\frac{2}{3}}$ regret of [FGQ$^+$23] to $\sqrt{T}$ in the latter case. The techniques we use to achieve them could be of independent interest.

Tables that compare our results with previous ones are provided in Appendix A. Notably, our framework generalizes the Algorithmic Information Ratio (AIR) framework of [XZ23] and [LWZ25], substantially simplifying the analysis while enhancing algorithmic flexibility (Section 4). This generalization may facilitate future development in this line of research.

We remark that, similar to [FGQ$^+$23], the term "model-free" learning in our work does not mean that the learner has no access to the model class $\mathcal{M}$ or has computational constraints. Instead, it only means that the regret bound is independent of the size of the model set $\mathcal{M}$. This implicitly restricts the learner from making fine-grained estimation over $\mathcal{M}$.

## 2 PRELIMINARY

We consider Decision Making with Structured Observations (DMSO) [FKQR21]. Let $\mathcal{M}$ be a model space, $\Pi$ a policy space, $\mathcal{O}$ an observation space, and $V$ a value function. For simplicity, we $|\Pi|$ is finite. Each model $M \in \mathcal{M}$ is a mapping from policy space $\Pi$ to a distribution over observations $\Delta(\mathcal{O})$. Every model $M \in \mathcal{M}$ is associated with a value function $V_M : \Pi \to [0, 1]$ that specifies the expected payoff of policy $\pi \in \Pi$ in model $M$. We denote $\pi_M = \operatorname{argmax}_{\pi \in \Pi} V_M(\pi)$.

The learner interacts with the environment for $T$ rounds. In each round $t = 1, \ldots, T$, the environment first chooses a model $M_t \in \mathcal{M}$ without revealing it to the learner. Then the learner selects a policy $\pi_t \in \Pi$, and observes an observation $o_t \sim M_t(\cdot|\pi_t)$. The regret with respect to policy $\pi^\star \in \Pi$ is

$$\operatorname{Reg}(\pi^\star) = \sum_{t=1}^{T} \left( V_{M_t}(\pi^\star) - V_{M_t}(\pi_t) \right).$$

**Markov Decision Process** A Markov decision process is defined by a tuple $(\mathcal{S}, \mathcal{A}, P, R, H, s_1)$, where $\mathcal{S}$ is the state space, $\mathcal{A}$ is the action space, $P : \mathcal{S} \times \mathcal{A} \to \Delta(\mathcal{S})$ is the transition kernel, $R : \mathcal{S} \times \mathcal{A} \to \Delta([0,1])$ is the reward distribution (with abuse of notation, we also use $R(s, a)$ to denote the expected reward $R(s, a) \in [0, 1]$), $H$ the horizon, and $s_1$ the initial state. Assume $\mathcal{S} = \bigcup_{h=1}^{H} \mathcal{S}_h$ with $\mathcal{S}_i \cap \mathcal{S}_j = \emptyset$ for $i \neq j$, and $\mathcal{S}_1 = \{s_1\}$. In every step $h = 1, 2, \ldots, H$ within an episode, the learner observes the state $s_h \in \mathcal{S}_h$ and selects an action $a_h \in \mathcal{A}$. The learner then transitions to the next state via $s_{h+1} \sim P(\cdot|s_h, a_h)$, which is only supported on $\mathcal{S}_{h+1}$, and receives the reward $r_h \sim R(s_h, a_h)$. We assume that the reward is constrained such that $\sum_{h=1}^{H} r_h \in [0, 1]$ for any policy almost surely. Given a policy $\pi : \mathcal{S} \to \mathcal{A}$, the $Q$-function and $V$-function for $s \in \mathcal{S}_h$ are defined by $Q^\pi(s, a) = \mathbb{E}^\pi[\sum_{h'=h}^{H} r_h \mid s_h = s, a_h = a]$ and $V^\pi(s) = Q^\pi(s, \pi(s))$. The $Q$-function and $V$-function of an optimal policy $\pi^\star$ are abbreviated with $Q^\star$ and $V^\star$. We use $Q^\pi(s, a; M)$ and $Q^\star(s, a; M)$ to denote the $Q$-functions under model $M = (P, R)$.

Learning in MDPs is a DMSO problem where $\mathcal{M} = \mathcal{P} \times \mathcal{R}$ with $\mathcal{P}$ being the set of transition kernels and $\mathcal{R}$ the set of reward functions. A *round* in DMSO corresponds to an MDP episode, and observation $o = (s_1, a_1, r_1, s_2, a_2, r_2, \ldots, r_H)$ is the trajectory. For any function $g$, we write $\mathbb{E}^{\pi,M}[g(o)] = \mathbb{E}_{o \sim M(\cdot|\pi)}[g(o)]$. If $g(o)$ only depends on $(s_1, a_1, s_2, a_2, \ldots, a_H)$, we also write it as $\mathbb{E}^{\pi,P}[g(o)]$. We use $V_M(\pi) = \mathbb{E}^{\pi,M}[\sum_{h=1}^{H} r_h]$ to denote the expected total reward obtained by policy $\pi$ in MDP $M$, and $d_h^{\pi,M}(s, a)$ (or $d_h^{\pi,P}(s, a)$) the occupancy measure on step $h$ under policy $\pi$ and model $M$ (or transition $P$).

## 2.1 $\Phi$-RESTRICTED LEARNING

For DMSO, [FKQR21, FGH23] and [CMB25] studied the *stochastic* setting where $M_t = M^\star$ for all $t$. They showed that the DEC characterizes the regret lower bound and captures the complexity of decision making. They proposed model-based algorithms with near-optimal upper bounds up to the model estimation complexity $\log|\mathcal{M}|$. On the other hand, [FRSS22] and [XZ23] studied the pure *adversarial* setting where $M_t$ arbitrarily changes over time. For this setting, they identified that DEC of the convexified model class characterizes the regret lower bound, which could be significantly larger than DEC of the original model class. Their upper bound replaces $\log|\mathcal{M}|$ by $\log|\Pi|$, reflecting that they perform policy-based learning without finegrained estimation of the model.

Several works go beyond pure model learning or pure policy learning. [FGQ$^+$23] considered model-free value learning in the stochastic setting where only the value function is estimated, aiming to only incur $\log|\mathcal{F}|$ estimation complexity, where $\mathcal{F}$ is the value function set. [LWZ25] and [CR25] considered the hybrid setting where part of the environment is stochastic and part adversarial, and the target of estimation is only on the optimal policy and the stochastic part of the environment.

We base our presentation in [LWZ25]'s formulation, which can cover all cases mentioned above.

**Definition 1** (Infosets and $\Phi$ [LWZ25, CR25]). *Let $\Phi$ be a collection of subsets of $\mathcal{M} \times \Pi$ satisfying: 1) The subsets are disjoint, i.e., for any $\phi, \phi' \in \Phi$, if $\phi \neq \phi'$, then $\phi \cap \phi' = \emptyset$. 2) Every $\phi$ contains a single policy, i.e., if $(M, \pi), (M', \pi') \in \phi$, then $\pi = \pi'$. We call a $\phi \in \Phi$ an information set (infoset). Due to 2) above, each $\phi \in \Phi$ is associated with a unique policy. We denote this policy as $\pi_\phi$. We also define $\Psi \triangleq \bigcup_{\phi \in \Phi} \phi \subseteq \mathcal{M} \times \Pi$.*

With Definition 1, for given $\rho \in \Delta(\Phi)$, $p \in \Delta(\Pi)$, $\nu \in \Delta(\Psi)$, and $\eta > 0$, [LWZ25] defined $\Phi$-AIR:

$$\text{AIR}_\eta^\Phi(p, \nu; \rho) = \mathbb{E}_{\pi \sim p} \mathbb{E}_{(M, \pi^\star) \sim \nu} \mathbb{E}_{o \sim M(\cdot|\pi)} \left[ V_M(\pi^\star) - V_M(\pi) - \frac{1}{\eta} \text{KL}(\nu_{\boldsymbol{\phi}}(\cdot|\pi, o), \rho) \right], \quad (1)$$

where $\nu_{\boldsymbol{\phi}}(\cdot|\pi, o)$[1] is the posterior over $\phi$ given $(\pi, o)$, which satisfies $\nu(\phi|\pi, o) \propto \sum_{(M, \pi^\star) \in \phi} \nu(M, \pi^\star) M(o|\pi)$. $\Phi$-AIR can characterize the decision-making complexity in the $\Phi$-restricted environment defined below:

**Definition 2** ($\Phi$-resitricted environment [LWZ25, CR25]). *A $\Phi$-restricted environment is an (adversarial) decision making problem in which the environment commits to $\phi^\star \in \Phi$ at the beginning of the game and henceforth selects $(M_t, \pi_{\phi^\star}) \in \phi^\star$ in every round $t$ arbitrarily based on the history.*

---

[1]We use the notational convention in [LWZ25]: the bold subscript in $\nu_{\boldsymbol{\phi}}(\cdot|\pi, o)$ specifies the *identity* of the variable represented by '$\cdot$', instead of a *realized value* of that variable. The subscript may be omitted when clear.

**Theorem 3** ([LWZ25])**.** *For $\Phi$-restricted environment defined in Definition 2, there exists an algorithm ensuring $\mathbb{E}[\text{Reg}(\pi_{\phi^\star})] \leq \mathbb{E}\big[\sum_t \min_p \max_\nu \text{AIR}_\eta^\Phi(p, \nu; \rho_t)\big] + \frac{\log |\Phi|}{\eta}$.*

## 2.2 Results and Open Questions in [LWZ25]

[LWZ25]'s main results are based on $\Phi$-AIR: For *model-free* learning in *stochastic* MDPs, [LWZ25] obtained $\sqrt{T}$ regret for linear $Q^\star/V^\star$ MDPs (before their result, the best known rate is $T^{\frac{2}{3}}$). Unfortunately, their algorithm cannot handle other canonical settings such as bilinear classes, MDPs with bounded Bellman-eluder dimension, or MDPs with bounded coverability. For *model-based* learning in *hybrid* MDPs where the transition is fixed but the reward function changes arbitrarily over time, [LWZ25] obtained near-optimal regret bounds for general cases up to a $\log(|\mathcal{P}||\Pi|)$ factor.

An attempt was made by [LWZ25] to handle *model-free* learning in *hybrid* MDPs based on an extension of the optimistic DEC approach [FGQ+23]. However, their result only handles *full-information* reward feedback. Extension to the bandit setting is challenging under this framework as the optimistic update requires an explicit construction of the reward estimator.

In this work, we focus on model-free learning in both stochastic and hybrid MDPs. Our results generalize those of [LWZ25] in both directions: Our framework handles all canonical settings for *model-free* learning in *stochastic* MDPs, improving previous results by [FGQ+23]. It also handles *model-free* learning in *hybrid* MDPs with *bandit* feedback under the same reward assumption as [LWZ25].

## 3 Settings and Assumptions

Below, we show how to view model-free learning in stochastic and hybrid MDPs as learning in $\Phi$-restricted environments (Definition 2), and introduce the assumptions used in the paper.

### 3.1 The Stochastic Setting

**Definition 4** (Stochastic setting)**.** *In the stochastic setting, the environment commits to $M^\star$ at the beginning of the game and sets $M_t = M^\star$ in every round $t$.*

For model-free learning in the stochastic setting, we assume the following:

**Assumption 1** ($\Phi$ for model-free learning in stochastic MDPs)**.** *In the stochastic setting, in addition to $(\mathcal{M}, \Pi, \mathcal{O}, V)$ in the DMSO framework (Section 2), the learner is provided with a function set $\mathcal{F}$. Each model $M \in \mathcal{M}$ induces a function $f \in \mathcal{F}$. Assume that models inducing the same $f$ have the same $Q^\star$ function and hence the same optimal policy $\pi_M$ (for example, an $\mathcal{F}$ that contains all possible $Q^\star$ functions satisfies this, though $\mathcal{F}$ could also provide additional information). With this, $\Phi$ is created by partitioning $\mathcal{M}$ according to the function they induces: Define $\Phi = \{\phi_f : f \in \mathcal{F}\}$ where $\phi_f = \{(M, \pi_M) : M \text{ induces } f\}$. With abuse of notation, we write $M \in \phi$ to indicate that $(M, \pi_M) \in \phi$. We denote by $\pi_\phi$ the common optimal policy for all $M \in \phi$, and by $f_\phi(s, a)$ the $Q^\star$ function induced by $M \in \phi$, i.e., $f_\phi(s, a) = Q^\star(s, a; M)$ for all $M \in \phi$. Define $f_\phi(s) = \max_a f_\phi(s, a)$. We also use $V_\phi(\pi_\phi) := f_\phi(s_1)$ to denote the value of policy $\pi_\phi$ under any model in $\phi$.*

### 3.2 The Hybrid Setting

**Definition 5** (Hybrid setting)**.** *In the hybrid setting, the environment commits to $P^\star \in \mathcal{P}$ at the beginning of the game. In every round, the environment selects $R_t \in \mathcal{R}$ arbitrarily based on the history and sets $M_t = (P^\star, R_t)$.*

For model-free learning in the hybrid setting, the definition of $\Phi$ becomes more involved as it partitions over three dimensions $(\Pi, \mathcal{P}, \mathcal{R})$ in different ways. Formally, the partition should satisfy the following Assumption 2. We provide an illustration in Figure 1 in Appendix B to help the reader understand this assumption.

**Assumption 2** ($\Phi$ for learning in hybrid MDPs [LWZ25])**.** *The learner is provided with a function set $\mathcal{F}^\pi$ for every $\pi \in \Pi$. For any fixed $\pi$, each transition $P \in \mathcal{P}$ induces a function $f \in \mathcal{F}^\pi$. $\Phi$ is created*

*by partitioning $\mathcal{P} \times \mathcal{R} \times \Pi$ firstly according to $\pi$, and then according to the $f$ the transition induces in $\mathcal{F}^\pi$: Define $\Phi = \{\phi_{\pi,f} : \pi \in \Pi, f \in \mathcal{F}^\pi\}$, where $\phi_{\pi,f} = \{(P, R, \pi) : P \text{ induces } f \text{ in } \mathcal{F}^\pi, R \in \mathcal{R}\}$. We write $P \in \phi$ if there exists $R, \pi$ such that $(P, R, \pi) \in \phi$, and write $M = (P, R) \in \phi$ if $P \in \phi$. We denote by $\pi_\phi$ the unique $\pi \in \Pi$ defining $\phi \in \Phi$.*

The next assumption describes the requirement for the function set in our work.

**Assumption 3** (Unique reward to value mapping given $\phi$ [LWZ25]). *Let $\Phi$ satisfy Assumption 2. Assume that for any fixed $\phi$ and $P, P' \in \phi$, it holds that $Q^{\pi_\phi}(s, a; (P, R)) = Q^{\pi_\phi}(s, a; (P', R))$ for any $s, a, R$. We denote $f_\phi(s, a; R) = Q^{\pi_\phi}(s, a; (P, R))$ for any $P \in \phi$, and define $f_\phi(s; R) = \mathbb{E}_{a \sim \pi_\phi(\cdot|s)}[f_\phi(s, a; R)]$. We also use $V_{\phi,R}(\pi_\phi) = f_\phi(s_1; R)$ to denote the value of policy $\pi_\phi$ under $(P, R)$ for any $P \in \phi$.*

To understand Assumption 2 and Assumption 3 better, we take adversarial linear MDP [LWZ24] for example. In adversarial linear MDPs, the learner is given a known feature mapping $\varphi(s, a) \in \mathbb{R}^d$, such that the reward function can be represented as $R(s, a) = \varphi(s, a)^\top \theta_R$ and the transition as $P(s'|s, a) = \varphi(s, a)^\top \omega_P(s')$. In this case, one can show that for any $\pi$, $Q^\pi(s, a; P_1, R) = Q^\pi(s, a; P_2, R) \; \forall s, a, R$ if and only if $\mathbb{E}^{\pi, P_1}[\phi(s_h, a_h)] = \mathbb{E}^{\pi, P_2}[\phi(s_h, a_h)]$ for all $h$. Based on Assumption 3, we would like to put such $P_1$ and $P_2$ in the same partition under $\pi$ (see Figure 1 for an illustration). In other words, in Assumption 2, each $f \in \mathcal{F}^\pi$ corresponds to a unique value of $(\mathbb{E}^{\pi, P}[\phi(s_h, a_h)])_{h \in [H]} \in \mathbb{R}^{dH}$, and as long as two $P$'s share this value, they both belong to $\phi_{\pi,f}$.

We remark that while Assumption 3 is a reasonable generalization of Assumption 1 to the hybrid setting, it does not capture all learnable hybrid MDPs we are aware of. For example, if the transition space is partitioned according to Assumption 3 for hybrid low-rank MDPs with *unknown reward feature*, then $\log |\Phi|$ will scale *polynomially* with the number of possible feature mappings. In contrast, the work of [LMWZ24] handles this case with the regret scaling only *logarithmically* with the number of possible feature mappings. There is still technical difficulty in handling this case in our framework, and we leave it as future work.[2] We also remark that the previous work by [LWZ25] has the same limitation even in the full-information case.

Therefore, in this work, for the hybrid setting, we consider linear reward with *known* features, formally stated in the next assumption.

**Assumption 4** (Linear reward with known feature). *There exists a feature mapping $\varphi : \mathcal{S} \times \mathcal{A} \to \mathbb{R}^d$ known to the learner such that for any $R \in \mathcal{R}$, $R(s_h, a_h) = \varphi(s_h, a_h)^\top \theta_h(R)$ for all $(s_h, a_h) \in \mathcal{S}_h \times \mathcal{A}$ for some $\theta_h(R) \in \mathbb{R}^d$.*

While the stochastic setting (Definition 4) and the hybrid setting (Definition 5) are special cases of $\Phi$-restricted environments (Definition 2), the adversary in these special cases has additional restriction: for example, in the stochastic setting, the adversary is allowed to choose $M^\star \in \phi^\star$ at the beginning of the game, but has to stick to $M^\star$ throughout interactions. Similarly, $P^\star$ has to be fixed in the hybrid setting. This is different from the general $\Phi$-restricted setting where the adversary is allowed to choose $M_t \in \phi^\star$ arbitrarily in every round. However, using such a "coarser" partition $\Phi$ to model these settings is crucial for obtaining an improved estimation error that only scales with the size of the value function set.

## 4 GENERAL FRAMEWORK

This section introduce a general framework and complexity measure for the $\Phi$-restricted environment, which covers model-free learning in stochastic and hybrid MDPs as special cases. For given $\rho \in \Delta(\Phi)$, define for $p \in \Delta(\Pi)$ and $\nu \in \Delta(\Psi)$

$$\mathsf{AIR}_\eta^{\Phi, D}(p, \nu; \rho) = \mathbb{E}_{\pi \sim p} \mathbb{E}_{(M, \pi^\star) \sim \nu} \left[ V_M(\pi^\star) - V_M(\pi) - \frac{1}{\eta} D^\pi(\nu \| \rho) \right], \tag{2}$$

---

[2]The algorithm of [LMWZ24] begins with reward-free exploration to learn a feature mapping, followed by online learning over that fixed feature mapping. While this two-phase approach could potentially be integrated into our DEC framework in special cases, our goal is to explore approaches that avoid such design to address more general scenarios.

for some divergence measure $D^\pi(\nu\|\rho)$ convex in $\nu$ for any $\pi$ and $\rho$. $\Phi$-AIR defined in Eq. (1) is a special case where $D^\pi(\nu\|\rho) = \mathbb{E}_{M\sim\nu}\mathbb{E}_{o\sim M(\cdot|\pi)}[\mathrm{KL}(\nu_\phi(\cdot|\pi, o), \rho)]$. The general algorithm designed based on Eq. (2) is shown in Algorithm 1.

---

**Algorithm 1** General Framework

---

**Input:** Set of partitions $\Phi$ and its union $\Psi$ (defined in Section 2.1).
$\rho_1(\phi) = 1/|\Phi|, \forall \phi \in \Phi$.
**for** $t = 1, 2, \ldots, T$ **do**

  Set $p_t, \nu_t$ as the solution of the following minimax optimization (defined in Eq. (2)):

$$\min_{p\in\Delta(\Pi)} \max_{\nu\in\Delta(\Psi)} \mathsf{AIR}_\eta^{\Phi,D}(p, \nu; \rho_t). \tag{3}$$

  Execute $\pi_t \sim p_t$, and observe $o_t \sim M_t(\cdot|\pi_t)$.
  Update $\rho_{t+1} = \textsc{PosteriorUpdate}(\nu_t, \rho_t, \pi_t, o_t)$. $\tag{4}$

---

Algorithm 1 has two main steps. First, given the infoset distribution $\rho_t \in \Delta(\Phi)$, solve the policy distribution $p_t$ and the worst-case world distribution $\nu_t$ in the saddle-point problem Eq. (3). This is similar to the previous AIR framework in [XZ23] and [LWZ25]. After taking policy $\pi_t \sim p_t$ and receiving the observation $o_t \sim M_t(\cdot|\pi_t)$, perform a posterior update by incorporating new information from $o_t$ (Eq. (4)) and obtain the new infoset distribution $\rho_{t+1} \in \Delta(\Phi)$. In [XZ23] and [LWZ25], this posterior update step is simply $\rho_{t+1}(\phi) = \nu_t(\phi|\pi_t, o_t)$, but it could take different forms in our case depending on the specific divergence $D$ instantiated later.

The ability of our algorithm to handle a general divergence $D$ is enabled by our new analysis techniques. The update rule $\rho_{t+1}(\phi) = \nu_t(\phi|\pi_t, o_t)$ in [XZ23] and [LWZ25] and the corresponding regret analysis heavily relies on a "constructive minimax theorem" [XZ23] that is restricted to strictly convex divergence measures and somewhat cumbersome to generalize to divergence other than KL. Our new analysis, on the other hand, is more flexible and nicely connects to the standard analysis of mirror descent.

Our analysis goes as follows. For any $(M, \pi) \in \mathcal{M} \times \Pi$, denote $\delta_{M,\pi} \in \Delta(\mathcal{M} \times \Pi)$ as the Kronecker delta function centered at $(M, \pi)$. That is, $\delta_{M,\pi}(M, \pi) = 1$ and $\delta_{M,\pi}(M', \pi') = 0$ for any other $(M', \pi')$. By a simple first-order optimality condition (Lemma 18) and the fact that $\nu_t$ is a best response to $p_t$ (Eq. (3)), we have (recall the definition of $\pi_{\phi^\star}$ in Definition 2)

$$\mathbb{E}_{\pi\sim p_t}\left[V_{M_t}(\pi_{\phi^\star}) - V_{M_t}(\pi) - \frac{1}{\eta}D^\pi(\delta_{M_t,\pi_{\phi^\star}}\|\rho_t)\right] \tag{5}$$

$$\leq \max_{\nu\in\Delta(\Psi)} \mathbb{E}_{\pi\sim p_t}\mathbb{E}_{(M,\pi^\star)\sim\nu}\left[V_M(\pi^\star) - V_M(\pi) - \frac{1}{\eta}D^\pi(\nu\|\rho_t)\right] - \mathbb{E}_{\pi\sim p_t}\left[\frac{1}{\eta}\mathrm{Breg}_{D^\pi(\cdot\|\rho_t)}(\delta_{M_t,\pi_{\phi^\star}}, \nu_t)\right]$$

where $\mathrm{Breg}_F(x, y) = F(x) - F(y) - \langle\nabla F(y), x - y\rangle \geq 0$ is the Bregman divergence defined with a convex function $F$. Since $p_t$ is minimax solution in Eq. (3), after rearrangement of Eq. (5) and summation over $t$, we get

$$\sum_{t=1}^T \left(V_{M_t}(\pi_{\phi^\star}) - \mathbb{E}_{\pi\sim p_t}[V_{M_t}(\pi)]\right) \tag{6}$$

$$\leq \sum_{t=1}^T \min_{p\in\Delta(\Pi)} \max_{\nu\in\Delta(\Psi)} \mathsf{AIR}_\eta^{\Phi,D}(p, \nu; \rho_t) + \overbrace{\frac{1}{\eta}\sum_{t=1}^T \mathbb{E}_{\pi\sim p_t}\left[D^\pi(\delta_{M_t,\pi_{\phi^\star}}\|\rho_t) - \mathrm{Breg}_{D^\pi(\cdot\|\rho_t)}(\delta_{M_t,\pi_{\phi^\star}}, \nu_t)\right]}^{\textbf{Est}},$$

where we use the definition in Eq. (2). From Eq. (6), we have the following theorem.

**Theorem 6.** *Algorithm 1 achieves* $\mathbb{E}[\mathrm{Reg}(\pi_{\phi^\star})] \leq \mathbb{E}\left[\sum_t \min_p \max_\nu \mathsf{AIR}_\eta^{\Phi,D}(p, \nu; \rho_t) + \frac{\textbf{Est}}{\eta}\right]$.

The PosteriorUpdate in Eq. (4) has to be further designed in order to minimize **Est**. In Appendix C, we show how our new analysis recovers previous results of [XZ23] and [LWZ25] easily. We remark that when recovering [LWZ25]'s result for model-based learning in hybrid MDPs with full-information feedback, we chooses $D$ such that **Est** does not even scale with $\log|\Phi|$, while they achieve it with a more complex two-level algorithm. This shows the flexibility of our framework. In the next two subsection, we discuss about the two terms in the regret bound of Theorem 6.

### 4.1 DIVERGENCE MEASURE IN ALGORITHM 1 AND dig-dec

To handle the MDPs of interest in Section 3, we will instantiate Algorithm 1 with the following divergence $D$:

$$D^\pi(\nu\|\rho) = \mathbb{E}_{M\sim\nu}\mathbb{E}_{o\sim M(\cdot|\pi)}\left[\text{KL}\left(\nu_{\boldsymbol{\phi}}(\cdot|\pi, o), \rho\right) + \mathbb{E}_{\phi\sim\rho}\left[\overline{D}^\pi(\phi\|M)\right]\right], \tag{7}$$

where $\overline{D}^\pi(\phi\|M)$ is another divergence that measures the discrepancy between infoset $\phi$ and model $M$. Two choices of $\overline{D}$ will be introduced later in Section 4.2: *averaged estimation error* and *squared estimation error*.

With this definition of $D^\pi(\nu\|\rho)$, the first term in the regret bound in Theorem 6 can be bounded by the following complexity:

$$\text{dig-dec}_\eta^{\Phi,\overline{D}} \triangleq \max_{\rho\in\Delta(\Phi)}\min_{p\in\Delta(\Pi)}\max_{\nu\in\Delta(\Psi)} \text{AIR}_\eta^{\Phi,D}(p,\nu;\rho)$$

$$= \max_{\rho\in\Delta(\Phi)}\min_{p\in\Delta(\Pi)}\max_{\nu\in\Delta(\Psi)}$$

$$\mathbb{E}_{\pi\sim p}\mathbb{E}_{(M,\pi^\star)\sim\nu}\left[V_M(\pi^\star) - V_M(\pi) - \frac{1}{\eta}\mathbb{E}_{o\sim M(\cdot|\pi)}\left[\text{KL}(\nu_{\boldsymbol{\phi}}(\cdot|\pi, o), \rho)\right] - \frac{1}{\eta}\mathbb{E}_{\phi\sim\rho}\left[\overline{D}^\pi(\phi\|M)\right]\right].$$

$$\tag{8}$$

As both the KL and the $\overline{D}$ terms in Eq. (8) are measures of information gain, we call this complexity notion *dual information gain decision-estimation coefficient* (Dig-DEC). In Section 6, we compare in more detail how DigDEC is upper bounded by optimistic DEC — the complexity achieved by the prior work [FGQ+23] in the stochastic setting, and when the improvement can be arbitrarily large.

### 4.2 POSTERIORUPDATE AND BOUNDS FOR **Est**

The $\overline{D}$ we would like to use in Eq. (7) depends on the MDP class we consider. Below, we describe two classes of problems that are associated with different choices of $\overline{D}$, under which the achievable rates for **Est** are different.

#### 4.2.1 AVERAGE ESTIMATION ERROR

**Assumption 5** (Average estimation error). *There exists an estimation function $\ell_h : \Phi \times \mathcal{O} \to [-B, B]^N$ for every $h$ such that for any $\phi \in \Phi$ and any $M \in \phi$, it holds that for any $\pi \in \Pi$,*

$$\mathbb{E}^{\pi,M}[\ell_h(\phi; o_h)] = 0.$$

*Additionally, assume that the adversary is restricted such that for any $\pi, \phi$ and $t, t' \in [T]$, it holds that $\mathbb{E}^{\pi,M_t}[\ell_h(\phi; o_h)] = \mathbb{E}^{\pi,M_{t'}}[\ell_h(\phi; o_h)]$.*

The estimation function $\ell$ in Assumption 5 will be instantiated as the average Bellman error in Lemma 8 for all concrete examples. In this case, Assumption 5 is essentially the standard realizability assumption. We adopt the more general terminology of "estimation error" following [DKL+21].

**Theorem 7.** *Assume Assumption 5 holds. Then Algorithm 4 with Algorithm 2 as POSTERIORUPDATE with $\overline{D}^\pi(\phi\|M) = \overline{D}_{\text{av}}^\pi(\phi\|M) \triangleq \max_{j\in[N]} \frac{1}{B^2 H}\sum_{h=1}^H \left(\mathbb{E}^{\pi,M}[\ell_h(\phi; o_h)_j]\right)^2$ ensures*

$$\mathbb{E}[\textbf{Est}] \lesssim N\log(|\Phi|)T^{\frac{1}{3}}.$$

**Lemma 8.** *In the stochastic setting, Assumption 1 implies Assumption 5 with $N = 1$ estimation function $\ell_h(\phi; o_h) = f_\phi(s_h, a_h) - r_h - f_\phi(s_{h+1})$. In the hybrid setting, Assumption 2, Assumption 3 and Assumption 4 imply Assumption 5 with $N = d$ estimation functions $\ell_h(\phi; o_h)_j = f_\phi(s_h, a_h; \boldsymbol{e}_j) - \varphi(s_h, a_h)^\top \boldsymbol{e}_j - f_\phi(s_{h+1}; \boldsymbol{e}_j)$, where $\boldsymbol{e}_j$ as a reward represents the reward function defined as $R(s, a) = \varphi(s, a)_j$.*

In order to minimize **Est** in Eq. (6), we have to obtain an estimator of $\overline{D}_{\text{av}}^{\pi_t}(\phi\|M^\star)$ for all $\phi$. This can only be achieved via *batching*, which results in the design of Algorithm 4: In each epoch $k = 1, 2, \ldots, T/\tau$, the learner uses the same policy $\pi_k$ to interact with the MDP for $\tau$ episodes. While

similar epoching mechanism has been proposed in [FGQ+23], our construction of the estimator improves their rate of **Est** from $\sqrt{T}$ to $T^{\frac{1}{3}}$. To see the difference, consider the case $N = 1$ in the stochastic setting, in which the goal is to approximate $\sum_{h=1}^{H} \left( \mathbb{E}^{\pi_k, M^\star}[\ell_h(\phi; o_h)] \right)^2$. With observations $(o^1, \ldots, o^\tau)$ drawn from $M^\star(\cdot|\pi_k)$ in epoch $k$, we construct an *unbiased* estimator as $L_k(\phi) = \sum_{h=1}^{H} \left( \frac{2}{\tau} \sum_{i=1}^{\tau/2} \ell_h(\phi; o_h^i) \right) \left( \frac{2}{\tau} \sum_{i=\tau/2+1}^{\tau} \ell_h(\phi; o_h^i) \right)$, while [FGQ+23] constructs a *biased* estimator as $L_k(\phi) = \sum_{h=1}^{H} \left( \frac{1}{\tau} \sum_{i=1}^{\tau} \ell_h(\phi; o_h^i) \right)^2$. The detail of this estimation procedure is provided in Appendix F.1.

### 4.2.2 SQUARED ESTIMATION ERROR

Under stronger assumptions on the estimation function, we can improve the rate further. This is motivated by the class of Bellman-complete MDPs, given as followed.

**Definition 9** (Bellman completeness for the stochastic setting). *A $\Phi$ satisfying Assumption 1 is Bellman complete under model $M = (P, R)$ if for any $\phi \in \Phi$, there exists an $\phi' \in \Phi$ such that for any $s, a$,*

$$f_{\phi'}(s, a) = R(s, a) + \mathbb{E}_{s' \sim P(\cdot|s,a)}[f_\phi(s')].$$

*A $\Phi$ is Bellman complete if it is Bellman complete under all model $M \in \mathcal{M}$.[3]*

**Definition 10** (Bellman completeness for the hybrid setting). *A $\Phi$ satisfying Assumption 3 is Bellman complete under transition $P$ if for any $\phi \in \Phi$, there exists an $\phi' \in \Phi$ such that $\pi_{\phi'} = \pi_\phi$ and for any $s, a, R$,*

$$f_{\phi'}(s, a; R) = R(s, a) + \mathbb{E}_{s' \sim P(\cdot|s,a)}[f_\phi(s'; R)].$$

*A $\Phi$ is Bellman complete if it is Bellman complete under all transition $P \in \mathcal{P}$.*

**Assumption 6.** *There exists $\xi_h : \Phi \times \Phi \times \mathcal{O} \to [0, B^2]$ for every $h$ and $\mathcal{T}_M : \Phi \to \Phi$ for every $M$ such that for any $\phi$ and any $M \in \phi$, it holds that $\phi = \mathcal{T}_M \phi$. Furthermore, for any $\phi', \phi \in \Phi$, any $M \in \mathcal{M}$, and any $\pi \in \Pi$,*

$$4B^2 \cdot \mathbb{E}^{\pi, M} \left[ \xi_h(\phi', \phi; o_h) - \xi_h(\mathcal{T}_M \phi, \phi; o_h) \right] \geq \mathbb{E}^{\pi, M} \left[ (\xi_h(\phi', \phi; o_h) - \xi_h(\mathcal{T}_M \phi, \phi; o_h))^2 \right].$$

*Additionally, assume that the adversary is restricted such that $\mathcal{T}_{M_t} \phi = \mathcal{T}_{M_{t'}} \phi$ for all $\phi$ and all $t, t' \in [T]$.*

Similar to Assumption 5, the function $\xi$ in Assumption 6 will be instantiated as the square Bellman error in Lemma 12 for all concrete examples. In this case, Assumption 6 corresponds to the standard realizability plus Bellman-completeness assumption.

**Theorem 11.** *Assume Assumption 6 holds. Then Algorithm 1 with Algorithm 3 as POSTERIORUP-DATE with $\overline{D}^\pi(\phi\|M) = \overline{D}_{\mathsf{sq}}^\pi(\phi\|M) \triangleq \frac{1}{B^2 H} \sum_{h=1}^{H} \mathbb{E}^{\pi, M} \left[ \xi_h(\phi, \phi; o_h) - \xi_h(\mathcal{T}_M \phi, \phi; o_h) \right]$ ensures*

$$\mathbb{E}[\mathbf{Est}] \lesssim \log^2 |\Phi|.$$

**Lemma 12.** *In the stochastic setting, Assumption 1 together with Bellman completeness (Definition 9) implies Assumption 6 with the estimation function $\xi_h(\phi', \phi; o_h) = (f_{\phi'}(s_h, a_h) - r_h - f_\phi(s_{h+1}))^2$ and $B^2 = 1$. In the hybrid setting, Assumption 2, Assumption 3 and Assumption 4 together with Bellman completeness (Definition 10) imply Assumption 6 with the estimation function $\xi_h(\phi', \phi; o_h) = \|(f_{\phi'}(s_h, a_h; \boldsymbol{e}_j) - \varphi(s_h, a_h)^\top \boldsymbol{e}_j - f_\phi(s_{h+1}; \boldsymbol{e}_j))_{j \in [d]}\|^2$ and $B^2 = d$, where $\boldsymbol{e}_j$ as a reward represents the reward function defined as $R(s, a) = \varphi(s, a)_j$.*

With Assumption 6, POSTERIORUPDATE no longer needs to rely on batching. We leverage a two-timescale POSTERIORUPDATE learning procedure similar to that of [FGQ+23], which in turn builds on [AZ22]. We refine their approach so **Est** can be bounded by a constant, improving over [FGQ+23]'s $T^{\frac{1}{3}}$ bound. In addition, our approach comes with a simpler regret analysis. Our POSTERIORUPDATE features a two-layer learning structure with a biased loss on the top layer. It is related to model selection algorithms with comparator-dependent second-order bounds (e.g., [CLW21]), but also has its special structure not seen in prior work. Thus, we believe it is of independent interest. The detail of this estimation procedure is provided in Appendix F.2.

---

[3]In fact, it suffices to assume Bellman completeness only under the ground-truth model $M^\star$ (as in [FGQ+23]). However, it is without loss of generality to assume Bellman completeness under all $M \in \mathcal{M}$, as one can preprocess the model set $\mathcal{M}$ by eliminating models under which Bellman completeness does not hold. For simplicity, we assume the latter. Similar for Definition 10.

## 5 APPLICATIONS

By Theorem 6, the worst-case regret of Algorithm 1 is $\sum_t \min_p \max_\nu \mathsf{AIR}_\eta^{\Phi,D}(p,\nu;\rho_t) + \mathbf{Est}/\eta \le T \mathsf{dig\text{-}dec}_\eta^{\Phi,\overline{D}} + \mathbf{Est}/\eta$. In Section 4.2, we provided bounds on $\mathbf{Est}$ for two types of $\overline{D}$, i.e., $\overline{D}_{\mathsf{av}}$ and $\overline{D}_{\mathsf{sq}}$. Below, we provide upper bounds for $\mathsf{dig\text{-}dec}_\eta^{\Phi,\overline{D}}$ in concrete settings associated with each $\overline{D}$.

### 5.1 STOCHASTIC SETTINGS

For the stochastic setting, we consider MDP class $\mathcal{M}$ and its associated $\Phi$ with bounded bilinear rank [DKL+21], Bellman-eluder dimension [JLM21], and coverability [XFB+23]. The results are summarized in Table 1.The on-policy/off-policy in Table 1 should not be confused with the standard on-policy/off-policy training in standard RL. Instead, they are two subclasses of the bilinear class [DKL+21] and correspond to the $Q$-type/$V$-type Bellman eluder dimension in [JLM21]. The on-policy case has smaller regret because the executed policies provides sufficient exploration to notice a model missmatch, while in the off-policy case, the learner needs to execute an additional exploration policy for this purpose.

Table 1: Summary of the applications in the stochastic settings. BE stands for MDPs with bounded Bellman-eluder dimensions. Dig-DEC bounds are provided in Appendix H.3 for bilinear classes, Appendix H.4 for BE, and Appendix H.5 for coverable MDPs. Bilinear classes marked with $\star$ are restricted to estimation function specified in Lemma 29, under which it holds that $\mathsf{dig\text{-}dec}_\eta^{\Phi,\overline{D}_{\mathsf{sq}}} \le \mathsf{dig\text{-}dec}_\eta^{\Phi,\overline{D}_{\mathsf{av}}}$. $B$ and $N$ are parameters specified in Assumption 5 or Assumption 6. The regret bound is given by $T \cdot \mathsf{dig\text{-}dec}_\eta^{\Phi,\overline{D}} + \mathbf{Est}/\eta$ with $\mathbf{Est}$ given in Theorem 7 or Theorem 11, with the optimal $\eta$.

| Setting | | | $\mathsf{dig\text{-}dec}_\eta^{\Phi,\overline{D}}$ | $\overline{D}$ | $B$ | $N$ | $\mathbb{E}[\mathrm{Reg}(\pi_{M^\star})]$ |
|---|---|---|---|---|---|---|---|
| class | sub-class | completeness | | | | | |
| bilinear | on-policy | | $H^2 d\eta$ | $\overline{D}_{\mathsf{av}}$ | $1$ | $1$ | $H\sqrt{d\log|\Phi|}T^{\frac{2}{3}}$ |
| bilinear | off-policy | | $\sqrt{H^3 d|\mathcal{A}|^2\eta}$ | $\overline{D}_{\mathsf{av}}$ | $|\mathcal{A}|$ | $1$ | $H(d|\mathcal{A}|^2\log|\Phi|)^{\frac{1}{3}}T^{\frac{7}{9}}$ |
| BE | $Q$-type | | $H^2 d\eta$ | $\overline{D}_{\mathsf{av}}$ | $1$ | $1$ | $H\sqrt{d\log|\Phi|}T^{\frac{2}{3}}$ |
| BE | $V$-type | | $\sqrt{H^3 d|\mathcal{A}|\eta}$ | $\overline{D}_{\mathsf{av}}$ | $1$ | $1$ | $H(d|\mathcal{A}|\log|\Phi|)^{\frac{1}{3}}T^{\frac{7}{9}}$ |
| bilinear$^\star$ | on-policy | ✓ | $H^2 d\eta$ | $\overline{D}_{\mathsf{sq}}$ | $1$ | – | $H\sqrt{dT}\log|\Phi|$ |
| bilinear$^\star$ | off-policy | ✓ | $\sqrt{H^3 d|\mathcal{A}|^2\eta}$ | $\overline{D}_{\mathsf{sq}}$ | $|\mathcal{A}|$ | – | $H(d|\mathcal{A}|^2\log^2|\Phi|)^{\frac{1}{3}}T^{\frac{2}{3}}$ |
| BE | $Q$-type | ✓ | $H^2 d\eta$ | $\overline{D}_{\mathsf{sq}}$ | $1$ | – | $H\sqrt{dT}\log|\Phi|$ |
| BE | $V$-type | ✓ | $\sqrt{H^3 d|\mathcal{A}|\eta}$ | $\overline{D}_{\mathsf{sq}}$ | $1$ | – | $H(d|\mathcal{A}|\log^2|\Phi|)^{\frac{1}{3}}T^{\frac{2}{3}}$ |
| coverable | – | ✓ | $H^2 d\eta$ | $\overline{D}_{\mathsf{sq}}$ | $1$ | – | $H\sqrt{dT}\log|\Phi|$ |

We remark without giving details that in the stochastic setting, we can achieve same results in Table 1 *with high-probability* if we replace the $\mathbb{E}_{M\sim\nu}\mathbb{E}_{o\sim M(\cdot|\pi)}[\mathrm{KL}(\nu_{\boldsymbol{\phi}}(\cdot|\pi,o),\rho_t)]$ term by $\mathrm{KL}(\nu_{\boldsymbol{\phi}},\rho_t)$ in the definition of $D$ in Eq. (7). This variant, however, cannot handle the hybrid setting.

### 5.2 HYBRID SETTINGS

For the hybrid setting, with known linear reward feature, we consider transition structure including hybrid bilinear classes [LWZ25] and coverability [XFB+23]. While it is possible to also extend Bellman-eluder dimension to the hybrid setting, we omit it for simplicity.

## 6 COMPARISON WITH PRIOR COMPLEXITIES IN STOCHASTIC MDPS

Compared with $\mathsf{dig\text{-}dec}_\eta^{\Phi,\overline{D}}$ in Eq. (8) achieved by our algorithm, the complexity of optimistic E2D [FGQ+23] defined for the stochastic setting is

$$\mathsf{o\text{-}dec}_\eta^{\Phi,\overline{D}} = \max_{\rho\in\Delta(\Phi)} \min_{p\in\Delta(\Pi)} \max_{\nu\in\Delta(\Psi)} \mathbb{E}_{\pi\sim p}\mathbb{E}_{M\sim\nu}\mathbb{E}_{\phi\sim\rho}\left[V_\phi(\pi_\phi) - V_M(\pi) - \frac{1}{\eta}\overline{D}^\pi(\phi\|M)\right] \quad (9)$$

Table 2: Summary of the applications in the hybrid settings. Dig-DEC bounds are provided in Appendix I.2 for hybrid bilinear classes and Appendix I.3 for coverable MDPs. Bilinear classes marked with $\star$ are restricted to estimation function specified in Lemma 36, under which it holds that $\text{dig-dec}_\eta^{\Phi,\overline{D}_{\text{sq}}} \leq \text{dig-dec}_\eta^{\Phi,\overline{D}_{\text{av}}}$.

| Setting | | | $\text{dig-dec}_\eta^{\Phi,\overline{D}}$ | $\overline{D}$ | $B$ | $N$ | $\mathbb{E}[\text{Reg}(\pi_{\phi^\star})]$ |
|---|---|---|---|---|---|---|---|
| class | sub-class | completeness | | | | | |
| bilinear | on-policy | | $(H^5 d^3 \eta)^{\frac{1}{3}}$ | $\overline{D}_{\text{av}}$ | $1$ | $d$ | $d(H^5 \log |\Phi|)^{\frac{1}{4}} T^{\frac{5}{6}}$ |
| bilinear | off-policy | | $(H^6 d^3 |\mathcal{A}|^2 \eta)^{\frac{1}{4}}$ | $\overline{D}_{\text{av}}$ | $|\mathcal{A}|$ | $d$ | $(H^6 d^4 |\mathcal{A}|^2 \log |\Phi|)^{\frac{1}{5}} T^{\frac{13}{15}}$ |
| bilinear$^\star$ | on-policy | ✓ | $(H^5 d^4 \eta)^{\frac{1}{3}}$ | $\overline{D}_{\text{sq}}$ | $\sqrt{d}$ | – | $d(H^5 \log^2 |\Phi|)^{\frac{1}{4}} T^{\frac{3}{4}}$ |
| bilinear$^\star$ | off-policy | ✓ | $(H^6 d^4 |\mathcal{A}|^2 \eta)^{\frac{1}{4}}$ | $\overline{D}_{\text{sq}}$ | $\sqrt{d}|\mathcal{A}|$ | – | $(H^6 d^4 |\mathcal{A}|^2 \log^2 |\Phi|)^{\frac{1}{5}} T^{\frac{4}{5}}$ |
| coverable | – | ✓ | $(H^5 d^4 \eta)^{\frac{1}{3}}$ | $\overline{D}_{\text{sq}}$ | $\sqrt{d}$ | – | $d(H^5 \log^2 |\Phi|)^{\frac{1}{4}} T^{\frac{3}{4}}$ |

for the same choices of $\overline{D}$. Another model-free DEC in [LWZ25] is

$$\text{dec}_\eta^\Phi = \max_{\rho \in \Delta(\Phi)} \min_{p \in \Delta(\Pi)} \max_{\nu \in \Delta(\Psi)} \mathbb{E}_{\pi \sim p} \mathbb{E}_{(M,\pi^\star) \sim \nu} \left[ V_M(\pi^\star) - V_M(\pi) - \frac{1}{\eta} \mathbb{E}_{o \sim M(\cdot|\pi)} \left[ \text{KL}(\nu_{\boldsymbol{\phi}}(\cdot|\pi,o), \rho) \right] \right].$$

It is clear that $\text{dig-dec}_\eta^{\Phi,\overline{D}} \leq \text{dec}_\eta^\Phi$ for any non-negative divergence $\overline{D}$. Furthermore, we have

**Theorem 13.** *In the stochastic setting,* $\text{dig-dec}_\eta^{\Phi,\overline{D}} \leq \text{o-dec}_\eta^{\Phi,\overline{D}} + \eta$ *for any* $\overline{D}$.

Since DECs with parameter $\eta$ is usually of order $(\eta d)^\alpha$ for some intrinsic dimension $d$ and exponent $\alpha \leq 1$, Theorem 13 implies that for any setting that can be handled by optimistic E2D with a certain $\overline{D}$, it can also be covered by our algorithm with the same $\overline{D}$. Compared to optimistic DEC (Eq. (9)), Dig-DEC (Eq. (8)) has an extra KL term $\mathbb{E}_{\pi \sim p} \mathbb{E}_{M \sim \nu} \mathbb{E}_{o \sim M(\cdot|\pi)} [\text{KL}(\nu_{\boldsymbol{\phi}}(\cdot|\pi,o), \rho)]$ that can be further decomposed into two terms $\text{KL}(\nu_{\boldsymbol{\phi}}, \rho) + \mathbb{E}_{\pi \sim p} \mathbb{E}_{M \sim \nu} \mathbb{E}_{o \sim M(\cdot|\pi)} [\text{KL}(\nu_{\boldsymbol{\phi}}(\cdot|\pi,o), \nu_{\boldsymbol{\phi}})]$. They have different purposes: The first term $\text{KL}(\nu_{\boldsymbol{\phi}}, \rho)$ is for *regularization*, which makes the marginal distribution of $\nu$ not overly distant from $\rho$. This is the key that allows us to avoid the optimism mechanism in [FGQ$^+$23] (i.e., the $V_\phi(\pi_\phi)$ in Eq. (9)). We remark that by *regularization only*, we can *recover* the bounds achieved by optimistic DEC in the stochastic setting (this can be seen from the proof of Theorem 13), though it is unclear whether it can give *strict improvement*. However, the removal of optimism turns out to be important in the hybrid setting (Section 5.2) as it avoids explicit construction of the reward estimator. The second term $\mathbb{E}_{\pi \sim p} \mathbb{E}_{M \sim \nu} \mathbb{E}_{o \sim M(\cdot|\pi)} [\text{KL}(\nu_{\boldsymbol{\phi}}(\cdot|\pi,o), \nu_{\boldsymbol{\phi}})]$ is an *information gain* that allows Dig-DEC to *strictly improve* over optimistic DEC even in the stochastic setting. This is because all common choices of $\overline{D}$ such as bilinear divergence and squared Bellman error are mean-based and ignore distributional differences, and the KL information gain term can capture them. We give a toy example in the next theorem to show this, with a detailed proof provided in Appendix J.

**Theorem 14.** *There exists a* 3-*armed bandit instance where for any* $T \geq 1$ *and* $\eta \leq 1$, *the algorithm in [FGQ$^+$23] suffers* $\max_a \mathbb{E}[\text{Reg}(a)] \geq \Omega(\sqrt{T})$, *while our algorithm achieves* $\max_a \mathbb{E}[\text{Reg}(a)] \leq 1$.

## 7 CONCLUSION

We introduced a new model-free DEC approach that removes optimism in prior work and incorporates two information-gain terms into the AIR objective for decision making. In addition, we refined the online function estimation procedure. Together, they yield improved regret bounds in the stochastic setting and establish the first regret bounds for model-free learning in hybrid MDPs with bandit feedback. Future directions include relaxing Assumption 3 and Assumption 4, and investigating the fundamental limits of model-free learning.

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

# Appendices

## A  REGRET BOUND COMPARISON WITH PREVIOUS WORK

Table 3: Regret for model-free learning in stochastic MDPs (only showing $T$ dependence). "Toy 3-arm" is defined in Theorem 14. The two bounds in the same cell correspond to the cases with on-policy and off-policy estimation.

| Algorithm | Bilinear or BE | {Bilinear or BE or Coverable} + Bellman Complete + On-Policy | Toy 3-arm | Exploration Mechanism |
|---|---|---|---|---|
| [DKL+21] [JLM21] [XFB+23] | $T^{\frac{2}{3}}/T^{\frac{2}{3}}$ | $\sqrt{T}$ | $\sqrt{T}$ | optimism |
| [FGQ+23] | $T^{\frac{3}{4}}/T^{\frac{5}{6}}$ | $T^{\frac{2}{3}}$ | $\sqrt{T}$ | information gain + optimism |
| Ours | $T^{\frac{2}{3}}/T^{\frac{7}{9}}$ | $\sqrt{T}$ | 1 | information gain |

Table 4: Regret for learning in hybrid MDPs (stochastic transition and adversarial reward). The model-free learning guarantees in [LWZ25] and our work cannot handle general reward but rely on Assumption 4.

| Algorithm | Bilinear | {Bilinear or Coverable} + Bellman Complete + On-Policy | Model-Free | Bandit Feedback | General Reward |
|---|---|---|---|---|---|
| [LWZ25] | $\sqrt{T}/T^{\frac{2}{3}}$ | $\sqrt{T}$ | ✗ | ✓ | ✓ |
| [LWZ25] | $T^{\frac{3}{4}}/T^{\frac{5}{6}}$ | – | ✓ | ✗ | ✗ |
| Ours | $T^{\frac{5}{6}}/T^{\frac{13}{15}}$ | $T^{\frac{3}{4}}$ | ✓ | ✓ | ✗ |

## B  PARTITIONING OVER $\mathcal{P} \times \mathcal{R} \times \Pi$ FOR HYBRID MDPS

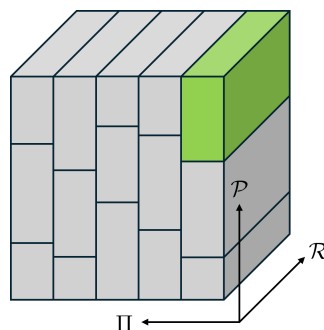

Figure 1: Partitioning for hybrid MDPs

Figure 1 illustrates the partition scheme over $\mathcal{M} \times \Pi = \mathcal{P} \times \mathcal{R} \times \Pi$ described in Assumption 2. Each infoset $\phi$ (represented by the green block in Figure 1) is associated with single policy $\pi_\phi$, a subset of transitions, and all reward functions. As shown in Figure 1, the partition over the $\mathcal{P}$ space could be different for different $\pi$.

# C    OMITTED DETAILS IN SECTION 2

In this section, we show that the algorithms in [XZ23] and [LWZ25] are special cases of Algorithm 1.

## C.1    RECOVERING THEOREM 3

The decision rule of [LWZ25]'s algorithm corresponds to Eq. (3) with $D^\pi(\nu\|\rho) = \mathbb{E}_{M\sim\nu}\mathbb{E}_{o\sim M(\cdot|\pi)}[\mathrm{KL}(\nu_\phi(\cdot|\pi,o),\rho)]$. It can be shown that $\mathrm{Breg}_{D^\pi(\cdot\|\rho)}(\nu,\nu') = \mathbb{E}_{M\sim\nu}\mathbb{E}_{o\sim M(\cdot|\pi)}\big[\mathrm{KL}(\nu_\phi(\cdot|\pi,o),\nu'_\phi(\cdot|\pi,o))\big]$ in this case. Furthermore, notice that when $\nu = \delta_{M_t,\pi_{\phi^\star}}$, we have $\nu_\phi(\cdot|\pi,o) = \delta_{\phi^\star}$ according to Definition 2. Thus, the estimation error term in Eq. (6) in [LWZ25]'s algorithm is

$$\mathbb{E}[\mathbf{Est}] = \mathbb{E}\left[\sum_{t=1}^T \Big(\mathrm{KL}(\delta_{\phi^\star},\rho_t) - \mathbb{E}_{o\sim M_t(\cdot|\pi_t)}\Big[\mathrm{KL}(\delta_{\phi^\star},(\nu_t)_\phi(\cdot|\pi_t,o))\Big]\Big)\right]$$

$$= \mathbb{E}\left[\sum_{t=1}^T (\mathrm{KL}(\delta_{\phi^\star},\rho_t) - \mathrm{KL}(\delta_{\phi^\star},(\nu_t)_\phi(\cdot|\pi_t,o_t)))\right] = \mathbb{E}\left[\sum_{t=1}^T \log\frac{\nu_t(\phi^\star|\pi_t,o_t)}{\rho_t(\phi^\star)}\right],$$

where in the second equality we use that $o_t$ is drawn from $M_t(\cdot|\pi_t)$. Thus, by letting $\rho_{t+1}(\phi) = \nu_t(\phi|\pi_t,o_t)$, their algorithm achieves $\mathbb{E}[\mathbf{Est}] = \mathbb{E}\left[\sum_{t=1}^T \log\frac{\rho_{t+1}(\phi^\star)}{\rho_t(\phi^\star)}\right] \le \log\frac{1}{\rho_1(\phi^\star)} = \log|\Phi|$. Using this in Eq. (6) proves Theorem 3. The results of [XZ23] can also be recovered as they are special cases of [LWZ25].

## C.2    RECOVERING RESULTS FOR ADVERSARIAL MDP WITH FULL-INFORMATION FEEDBACK [LWZ25]

For learning with full information feedback in the adversarial MDPs, the learner can observe the full reward function at the end of each episode. In other words, at episode $t$, the reward function $R_t : \mathcal{S} \times \mathcal{A} \to [0,1]$ is part of the observation $o_t$. In this setting, the $\log|\Pi|$ dependence in the regret bound can be improved to $\log|\mathcal{A}|$. To achieve this, [LWZ25] designed a two-level algorithm and define a new notion called InfoAIR. We can recover this result by instantiating our Algorithm 1 with $\Phi = \{\phi_{P,(a_s)_{s\in\mathcal{S}}} : P \in \mathcal{P}, a_s \in \mathcal{A}, \forall s \in \mathcal{S}\}$ where $\phi_{P,(a_s)_{s\in\mathcal{S}}} = \{((P,R),\pi^\star) : R \in \mathcal{R}, \pi^\star = (a_s)_{s\in\mathcal{S}}\}$, that is, partitioning $\mathcal{M} \times \Pi$ according to the transition kernel and the actions taken by the policy on all states. Then define

$$D^\pi(\nu\|\rho) = \mathbb{E}_{(P,R,\pi^\star)\sim\nu}\mathbb{E}_{o\sim M_{P,R}(\cdot|\pi)}\mathbb{E}_{s\sim d^{\pi,P}}\left[\mathrm{KL}(\nu_{\boldsymbol{a}_s,\boldsymbol{P}}(\cdot|\pi,o),\rho_{\boldsymbol{a}_s,\boldsymbol{P}})\right],$$

where $M_{P,R}$ denotes the MDP model with transition kernel $P$ and reward function $R$, and $\rho_{\boldsymbol{a}_s,\boldsymbol{P}}$ denotes $\rho$'s marginal distribution over $(a_s,P)$ following our notational convention. Finally, update the posterior as $\rho_{t+1} = \mathrm{argmin}_\rho \sum_{s\in\mathcal{S}} \mathrm{KL}(\rho_{\boldsymbol{a}_s,\boldsymbol{P}},\nu_{\boldsymbol{a}_s,\boldsymbol{P}}(\cdot|\pi_t,o_t))$. This recovers the same regret bound as in [LWZ25] without the need for the two-level design. We also note that the analysis for this result requires our new proof strategy in Eq. (5), as the $D^\pi(\nu\|\rho)$ here is not strictly convex in $\nu$ and the previous proof [XZ23, LWZ25] cannot be applied.

## D  CONCENTRATION INEQUALITY

**Lemma 15** (Freedman's inequality [BLL$^+$11]). *Let $X_1, X_2, \ldots$ be a martingale difference sequence with respect to a filtration $\mathfrak{F}_1 \subset \mathfrak{F}_2 \subset \cdots$ such that $\mathbb{E}[X_t|\mathfrak{F}_t] = 0$ and assume $X_t \leq B$ almost surely. Then for any $\alpha \geq B$, with probability at least $1 - \delta$,*

$$\sum_{t=1}^{T} X_t \leq \frac{1}{\alpha} \sum_{t=1}^{T} \mathbb{E}[X_t^2|\mathfrak{F}_t] + \alpha \log(1/\delta). \tag{10}$$

**Lemma 16** (Empirical Freedman's inequality). *Let $X_1, X_2, \ldots$ be a sequence with respect to a filtration $\mathfrak{F}_1 \subset \mathfrak{F}_2 \subset \cdots$ such that $\mathbb{E}[X_t|\mathfrak{F}_t] = \mu_t$ and assume $\max\{X_t - \mu_t, X_t\} \leq B$ almost surely. Then for any $\alpha \geq 4B$, with probability at least $1 - \delta$,*

$$\sum_{t=1}^{T} (\mu_t - X_t) \leq \frac{4}{\alpha} \sum_{t=1}^{T} X_t^2 + \alpha \log(1/\delta). \tag{11}$$

*Proof.* Denote $\mathbb{E}_t[\cdot] = \mathbb{E}[\cdot \mid \mathfrak{F}_t]$. We have at any time step

$$\mathbb{E}_t \left[ \exp\left( \frac{1}{\alpha}(\mu_t - X_t) - \frac{4}{\alpha^2} X_t^2 \right) \right]$$

$$\leq \mathbb{E}_t \left[ 1 + \frac{1}{\alpha}(\mu_t - X_t) - \frac{4}{\alpha^2} X_t^2 + \left( \frac{1}{\alpha}(\mu_t - X_t) - \frac{4}{\alpha^2} X_t^2 \right)^2 \right]$$

$$\leq 1 + \mathbb{E}_t \left[ -\frac{4}{\alpha^2} X_t^2 + \frac{2}{\alpha^2}((\mu_t - X_t)^2 + X_t^2) \right] \leq 1.$$

Markov inequality finishes the proof. $\square$

**Lemma 17.** *Let $(X_1, Y_1), (X_2, Y_2) \ldots$ be a sequence with respect to a filtration $\mathfrak{F}_1 \subset \mathfrak{F}_2 \subset \cdots$ such that $|X_t| \leq B$ and $0 \leq Y_t \leq B$ almost surely. Furthermore, $\mathbb{E}[X_t|\mathfrak{F}_t] \geq \mathbb{E}[Y_t|\mathfrak{F}_t]$ and $B\mathbb{E}[X_t|\mathfrak{F}_t] \geq \mathbb{E}[X_t^2|\mathfrak{F}_t]$. Then with probability at least $1 - \delta$,*

$$\frac{1}{2} \sum_{t=1}^{T} \mathbb{E}[X_t|\mathfrak{F}_t] \leq \sum_{t=1}^{T} \left( X_t - \frac{1}{4} Y_t \right) + 9B \log(1/\delta). \tag{12}$$

*Also, with probability at least $1 - \delta$,*

$$\frac{1}{2} \sum_{t=1}^{T} X_t \leq \sum_{t=1}^{T} \left( X_t - \frac{1}{4} Y_t \right) + 9B \log(1/\delta). \tag{13}$$

*Proof.* Denote $\mathbb{E}_t[\cdot] = \mathbb{E}[\cdot \mid \mathfrak{F}_t]$. Let $c \in [\frac{1}{2}, 1]$ be a fixed constant, and define $Z_t = cX_t - \frac{1}{4}Y_t$. Applying Lemma 15 with $\alpha = 9B$ gives

$$\sum_{t=1}^{T} (\mathbb{E}_t[Z_t] - Z_t) \leq \frac{1}{9B} \sum_{t=1}^{T} \mathbb{E}_t[(\mathbb{E}_t[Z_t] - Z_t)^2] + 9B \log(1/\delta)$$

$$\leq \frac{1}{9B} \sum_{t=1}^{T} \mathbb{E}_t[Z_t^2] + 9B \log(1/\delta)$$

$$\leq \frac{1}{9B} \sum_{t=1}^{T} \left( 2c^2 \mathbb{E}_t[X_t^2] + \frac{2}{16} \mathbb{E}_t[Y_t^2] \right) + 9B \log(1/\delta)$$

$$\leq \frac{1}{9} \sum_{t=1}^{T} \left( 2c^2 \mathbb{E}_t[X_t] + \frac{2}{16} \mathbb{E}_t[X_t] \right) + 9 \log(1/\delta)$$

$$\qquad\qquad\qquad (\mathbb{E}_t[Y_t^2] \leq B\mathbb{E}_t[Y_t] \text{ because } Y_t \in [0, B])$$

Rearranging:

$$\sum_{t=1}^{T} \mathbb{E}_t \left[ Z_t - \left( \frac{2c^2}{9} + \frac{1}{72} \right) X_t \right] \leq \sum_{t=1}^{T} Z_t + 9B \log(1/\delta). \tag{14}$$

To prove Eq. (12), let $c = 1$, which gives $\mathbb{E}_t \left[ Z_t - \left( \frac{2c^2}{9} + \frac{1}{72} \right) X_t \right] = \mathbb{E}_t \left[ X_t - \frac{1}{4} Y_t - \frac{17}{72} X_t \right] \geq \frac{1}{2} \mathbb{E}_t [X_t]$. Combining this with Eq. (14) proves Eq. (12). To prove Eq. (13), let $c = \frac{1}{2}$. which gives $\mathbb{E}_t \left[ Z_t - \left( \frac{2c^2}{9} + \frac{1}{72} \right) X_t \right] = \mathbb{E}_t \left[ \frac{1}{2} X_t - \frac{1}{4} Y_t - \frac{5}{72} X_t \right] \geq 0$. Combining this with Eq. (14) and rearranging proves Eq. (13). $\square$

## E  MIRROR DESCENT

**Lemma 18** (First-order optimality condition). *For any concave and differentiable function $F$, if $\nu' \in \arg\max_{\nu \in \Omega} F(\nu)$ for some convex set $\Omega$, then $F(\nu) \leq F(\nu') - \text{Breg}_{(-F)}(\nu, \nu')$ for any $\nu \in \Omega$.*

*Proof.* Define $G = -F$. Then $G$ is convex and $\nu' \in \arg\min_{\nu'} G(\nu')$. We have by the definition of Bregman divergence $\text{Breg}_G(\nu, \nu') = G(\nu) - G(\nu') - \langle \nabla G(\nu'), \nu - \nu' \rangle$, and first-order optimality condition $\langle \nabla G(\nu'), \nu - \nu' \rangle \geq 0$. Thus, $G(\nu) \geq G(\nu') + \text{Breg}_G(\nu, \nu')$, which is equivalent to $F(\nu) \leq F(\nu') + \text{Breg}_{(-F)}(\nu, \nu')$. $\qquad\square$

**Lemma 19.** *Let $g : \Phi \to [-1, 1]$ be any function and let $\nu, \rho \in \Delta(\Phi)$. Then for any $\eta > 0$,*

$$\mathbb{E}_{\phi \sim \nu}[g(\phi)] - \mathbb{E}_{\phi \sim \rho}[g(\phi)] - \frac{1}{\eta}KL(\nu, \rho) \leq \eta.$$

*Proof.*

$$\mathbb{E}_{\phi \sim \nu}[g(\phi)] - \mathbb{E}_{\phi \sim \rho}[g(\phi)] \leq 2D_{\text{TV}}(\nu, \rho) \leq 2\sqrt{KL(\nu, \rho)} \leq \frac{1}{\eta}KL(\nu, \rho) + \eta,$$

where we use Pinsker's inequality and AM-GM inequality. $\qquad\square$

**Lemma 20** (Mirror descent with auxiliary terms). *Let $F_t$ be a convex function over $\Delta_N$, and let $\ell_t, b_t \in \mathbb{R}^N$ with $\ell_t^2$ denoting $(\ell_t(1)^2, \ldots, \ell_t(N)^2)$. Then the update $p_1 = \frac{1}{N}\mathbf{1}$ and*

$$p_{t+1} = \arg\min_{p \in \Delta_N} \left\{ \langle p, \ell_t + 4\gamma\ell_t^2 + b_t \rangle + F_t(p) + \frac{1}{\gamma}KL(p, p_t) \right\}$$

*with $\gamma|\ell_t(i)| \leq \frac{1}{16}$ and $0 \leq \gamma b_t(i) \leq \frac{1}{4}$ for all $i \in [N]$ ensures for any $p^\star \in \Delta_N$,*

$$\sum_{t=1}^{T} \langle p_t, \ell_t \rangle$$

$$\leq \frac{\log N}{\gamma} + \sum_{t=1}^{T} \left( \langle p^\star, \ell_t + 4\gamma\ell_t^2 \rangle + \langle p^\star, b_t \rangle - \frac{1}{2}\langle p_t, b_t \rangle + F_t(p^\star) - F_t(p_{t+1}) - \text{Breg}_{F_t}(p^\star, p_{t+1}) \right).$$

*Proof.* By [Lemma 18](),

$$\langle p_{t+1}, \ell_t + 4\gamma\ell_t^2 + b_t \rangle + F_t(p_{t+1}) + \frac{1}{\gamma}KL(p_{t+1}, p_t)$$

$$\leq \langle p^\star, \ell_t + 4\gamma\ell_t^2 + b_t \rangle + F_t(p^\star) + \frac{1}{\gamma}KL(p^\star, p_t) - \text{Breg}_{F_t}(p^\star, p_{t+1}) - \frac{1}{\gamma}KL(p^\star, p_{t+1}).$$

Rearranging gives

$$\langle p_t, \ell_t + 4\gamma\ell_t^2 \rangle$$

$$\leq \langle p^\star, \ell_t + 4\gamma\ell_t^2 \rangle + \langle p_t - p_{t+1}, \ell_t + 4\gamma\ell_t^2 + b_t \rangle - \frac{1}{\gamma}KL(p_{t+1}, p_t)$$

$$+ \langle p^\star - p_t, b_t \rangle + \frac{KL(p^\star, p_t) - KL(p^\star, p_{t+1})}{\gamma} + F_t(p^\star) - F_t(p_{t+1}) - \text{Breg}_{F_t}(p^\star, p_{t+1}). \quad (15)$$

Since $\gamma|\ell_t(i) + 4\gamma\ell_t(i)^2 + b_t(i)| \leq \frac{1}{16} + 4 \times (\frac{1}{16})^2 + \frac{1}{4} \leq 1$, by [Lemma 19]() we have

$$\langle p_t - p_{t+1}, \ell_t + 4\gamma\ell_t^2 + b_t \rangle - \frac{1}{\gamma}KL(p_{t+1}, p_t)$$

$$\leq \gamma \langle p_t, (\ell_t + 4\gamma\ell_t^2 + b_t)^2 \rangle$$

$$\leq 2\gamma \langle p_t, (\tfrac{5}{4}\ell_t)^2 \rangle + 2\gamma \langle p_t, b_t^2 \rangle$$

$$\leq \langle p_t, 4\gamma\ell_t^2 \rangle + \frac{1}{2}\langle p_t, b_t \rangle.$$

Using this in Eq. (15) we get

$$
\begin{aligned}
\langle p_t, \ell_t \rangle \leq{} & \left\langle p^\star, \ell_t + 4\gamma \ell_t^2 \right\rangle + \langle p^\star, b_t \rangle - \frac{1}{2} \langle p_t, b_t \rangle \\
& + \frac{\mathrm{KL}(p^\star, p_t) - \mathrm{KL}(p^\star, p_{t+1})}{\gamma} + F_t(p^\star) - F_t(p_{t+1}) - \mathrm{Breg}_{F_t}(p^\star, p_{t+1}).
\end{aligned}
$$

Summing over $t$ gives the desired inequality. $\qquad\square$

# F ESTIMATION PROCEDURES

We present the choices of POSTERIORUPDATE as standalone online learning algorithms because they might be of independent interest.

## F.1 AVERAGE ESTIMATION ERROR MINIMIZATION VIA BATCHING

---

**Algorithm 2** Epoch-based learning algorithm for average estimation error

---

**Input**: An estimation function $\ell_h : \Phi \times \mathcal{O} \to [-B, B]^N$ satisfying Assumption 5.

**Parameter:** $\tau = T^{\frac{1}{3}}, \beta = 7\tau N \iota, \gamma = \frac{1}{2\beta}, \iota = \log(12NKH/\delta)$.

**for** $k = 1, 2, \ldots, K$ **do**

Receive observations $o_t \sim M_t(\cdot | \pi_k)$ for all $t \in \mathcal{I}_k = \{(k-1)\tau + 1, \ldots, k\tau\}$.
Split $\mathcal{I}_k$ into two sub-intervals of equal size:

$$\mathcal{I}_k^- = \{(k-1)\tau + 1, \ldots, (k-1)\tau + \tfrac{\tau}{2}\} \quad \text{and} \quad \mathcal{I}_k^+ = \{(k-1)\tau + \tfrac{\tau}{2} + 1, \ldots, k\tau\}.$$

Define for all $j \in [N]$,

$$L_k(\phi)_j = \frac{\tau}{B^2 H} \sum_{h=1}^{H} \left( \frac{1}{|\mathcal{I}_k^-|} \sum_{t \in \mathcal{I}_k^-} \ell_h(\phi; o_{t,h})_j \right) \left( \frac{1}{|\mathcal{I}_k^+|} \sum_{t \in \mathcal{I}_k^+} \ell_h(\phi; o_{t,h})_j \right), \quad L_k(\phi) = \sum_{j=1}^{N} L_k(\phi)_j.$$

Let $(F_t)_{t \in \mathcal{I}_k} : \Delta(\Phi) \to \mathbb{R}$ be convex functions. Calculate

$$\rho_{k+1} = \underset{\rho \in \Delta(\Phi)}{\operatorname{argmin}} \left\{ \langle \rho, L_k + (4\gamma + 2\beta^{-1})L_k^2 \rangle + \sum_{t \in \mathcal{I}_k} F_t(\rho) + \frac{1}{\gamma} \mathrm{KL}(\rho, \rho_k) \right\}. \tag{16}$$

---

**Lemma 21.** *With probability at least* $1 - \delta/3$, *Algorithm 2 satisfies*

$$\frac{1}{B^2 H} \sum_{k=1}^{K} \sum_{\phi} \rho_k(\phi) \sum_{t \in \mathcal{I}_k} \max_{j \in [N]} \sum_{h=1}^{H} \left( \mathbb{E}^{\pi_k, M_t}[\ell_h(\phi; o_h)_j] \right)^2 \leq \sum_{k=1}^{K} \sum_{\phi} \rho_k(\phi) \left( L_k(\phi) + \frac{1}{\beta} L_k(\phi)^2 \right) + 4\beta \log(3/\delta).$$

*Proof.* By Assumption 5, for any $t, t' \in \mathcal{I}_k$ it holds that

$$\mathbb{E}^{\pi_k, M_t}[\ell_h(\phi; o_h)] = \mathbb{E}^{\pi_k, M_{t'}}[\ell_h(\phi; o_h)].$$

We denote $\bar{\ell}_{k,h}(\phi) = \mathbb{E}^{\pi_k, M_t}[\ell_h(\phi; o_h)]$ for any $t \in \mathcal{I}_k$.

Clearly, the left-hand side of the desired inequality is upper bounded by

$$\frac{1}{B^2 H} \sum_{k=1}^{K} \sum_{\phi} \rho_k(\phi) \sum_{t \in \mathcal{I}_k} \sum_{j=1}^{N} \sum_{h=1}^{H} \left( \mathbb{E}^{\pi_k, M_t}[\ell_h(\phi; o_h)_j] \right)^2 = \frac{\tau}{B^2 H} \sum_{k=1}^{K} \sum_{\phi} \rho_k(\phi) \sum_{j=1}^{N} \sum_{h=1}^{H} \bar{\ell}_{k,h}(\phi)_j^2$$

By construction, $\mathbb{E}_k[L_k(\phi)] = \frac{\tau}{B^2 H} \sum_{j=1}^{N} \sum_{h=1}^{H} \bar{\ell}_{k,h}(\phi)_j^2$ due to the conditional independence of the observations. Furthermore, we have $L_k(\phi) \in [-\tau N, \tau N]$. Therefore, we can use Lemma 16 on the sequence $X_k = -\sum_{\phi} \rho_k(\phi) L_k(\phi)$ with $\beta \geq 7\tau N$:

$$\frac{\tau}{B^2 H} \sum_{k=1}^{K} \sum_{\phi} \rho_k(\phi) \sum_{j=1}^{N} \sum_{h=1}^{H} \bar{\ell}_{k,h}(\phi)_j^2 \leq \sum_{k=1}^{K} \sum_{\phi} \rho_k(\phi) \left( L_k(\phi) + \frac{1}{\beta} L_k(\phi)^2 \right) + 4\beta \log(3/\delta).$$

$\square$

**Lemma 22.** *With probability at least* $1 - \delta/3$,

$$\sum_{k=1}^{K} L_k(\phi^\star)^2 \leq KN^2 \log^2(12NKH/\delta).$$

*Proof.* By Assumption 5 and Lemma 15, for any $j, k, h$, we have with probability $1 - \delta$,

$$\left| \sum_{t \in \mathcal{I}_k^-} \ell_h(\phi^\star, o_{t,h})_j \right| = \left| \sum_{t \in \mathcal{I}_k^-} \ell_h(\phi^\star, o_{t,h})_j - \sum_{t \in \mathcal{I}_k^-} \mathbb{E}^{\pi_k, M_t}[\ell_h(\phi^\star; o_h)] \right| \leq B\sqrt{\tau \log(12/\delta)}$$

$$\left| \sum_{t \in \mathcal{I}_k^+} \ell_h(\phi^\star, o_{t,h})_j \right| = \left| \sum_{t \in \mathcal{I}_k^+} \ell_h(\phi^\star, o_{t,h})_j - \sum_{t \in \mathcal{I}_k^+} \mathbb{E}^{\pi_k, M_t}[\ell_h(\phi^\star; o_h)] \right| \leq B\sqrt{\tau \log(12/\delta)}\,.$$

Via a union bound over all these events, this holds simultaneously for all $j, k, h$. Hence with probability $1 - \delta$, we have $|L_k(\phi^\star)_j| \leq \frac{\tau}{B^2 H} H \left( \frac{1}{\tau} B\sqrt{\tau \log(12NKH/\delta)} \right)^2 = \log(12NKH/\delta)$ for all $j, k$ simultaneously. Summing over $j$ and $k$ finishes the proof. $\qquad\square$

**Lemma 23.** *With probability at least $1 - \delta/3$, we have*

$$\sum_{k=1}^{K} L_k(\phi^\star) \leq \frac{1}{\beta} \sum_{k=1}^{K} L_k(\phi^\star)^2 + 4\beta \log(6/\delta)$$

*Proof.* Define the random variable $X_k = \min\{L_k(\phi^\star), N \log(12NKH/\delta)\}$. By Lemma 16 we have with probability at least $1 - \delta/6$,

$$\sum_{k=1}^{K} X_t \leq \frac{1}{\beta} \sum_{k=1}^{K} L_k(\phi^\star)^2 + 4\beta \log(6/\delta)\,,$$

where we use that $\mathbb{E}_k[X_k] \leq \mathbb{E}_k[L_k(\phi^\star)] = 0$. Finally note that with probability $1 - \delta/6$ we have $L_k(\phi^\star) = X_k$ for all $k$ by the proof of Lemma 22. Combining both events finishes the proof. $\quad\square$

**Lemma 24.** *With probability at least $1 - \delta$, Algorithm 2 satisfies*

$$\frac{1}{B^2 H} \sum_{k=1}^{K} \sum_{\phi} \rho_k(\phi) \sum_{t \in \mathcal{I}_k} \max_{j \in [N]} \sum_{h=1}^{H} \left( \mathbb{E}^{\pi_k, M_t}[\ell_h(\phi; o_h)_j] \right)^2 \leq O\left( NT^{\frac{1}{3}} \log |\Phi| \right)$$

$$+ \sum_{k=1}^{K} \sum_{t \in \mathcal{I}_k} (F_t(\delta_{\phi^\star}) - F_t(\rho_{k+1}) - \mathrm{Breg}_{F_t}(\delta_{\phi^\star}, \rho_{k+1}))\,.$$

*Proof of Lemma 24.* By union bound, the events of Lemma 21, Lemma 22, and Lemma 23 hold simultaneously with probability $1 - \delta$. Observe that the update of $\rho_k$ (Eq. (16)) is in the form specified in Lemma 20. Invoking Lemma 20 with $b_k = \frac{2}{\beta} L_k^2$, we get

$$\sum_{k=1}^{K} \left\langle \rho_k, L_k + \frac{1}{\beta} L_k^2 \right\rangle \leq \frac{\log |\Phi|}{\gamma} \tag{17}$$

$$+ \sum_{k=1}^{K} \left( L_k(\phi^\star) + \left( 4\gamma + \frac{2}{\beta} \right) L_k(\phi^\star)^2 + \sum_{t \in \mathcal{I}_k} (F_t(\delta_{\phi^\star}) - F_t(\rho_{k+1}) - \mathrm{Breg}_{F_t}(\delta_{\phi^\star}, \rho_{k+1})) \right)\,.$$

Chaining Lemma 22 and Lemma 23,

$$\sum_{k=1}^{K} \left( L_k(\phi^\star) + \left( 4\gamma + \frac{2}{\beta} \right) L_k(\phi^\star)^2 \right)$$

$$\leq 4\beta \log(6/\delta) + \left( 4\gamma + \frac{3}{\beta} \right) K N^2 \log^2(12NKH/\delta)\,.$$

Using Lemma 21 and substituting $\beta = 7\tau N \iota$, $\gamma = \frac{1}{2\beta}$ yields

$$\frac{1}{B^2 H} \sum_{k=1}^{K} \sum_{\phi} \rho_k(\phi) \sum_{t \in \mathcal{I}_k} \max_{j \in [N]} \sum_{h=1}^{H} \left( \mathbb{E}^{\pi_k, M_t}[\ell_h(\phi; o_h)_j] \right)^2 \leq 35\tau N \iota + 20 \frac{K N \iota}{\tau}$$

Using $K = T/\tau$ and tuning $\tau = T^{\frac{1}{3}}$ yields $O(T^{\frac{1}{3}} N \iota)$. $\qquad\square$

### F.2 SQUARED ESTIMATION ERROR MINIMIZATION VIA BI-LEVEL LEARNING

---

**Algorithm 3** Bi-level learning algorithm for squared estimation error

---

**Input:** An estimation function $\xi_h : \Phi \times \Phi \times \mathcal{O} \to [0, B^2]$ satisfying Assumption 6.
**Parameter:** $\iota = 64 \log |\Phi|$, $\gamma = \frac{1}{4t}$.
$\rho_1(\phi) = 1/|\Phi|$, $\forall \phi \in \Phi$ and $q_1(\phi'|\phi) = 1/|\Phi|$, $\forall \phi', \phi \in \Phi$.
**for** $t = 1, 2, \ldots, T$ **do**

  Receive observation $o_t \sim M_t(\cdot|\pi_t)$.
  Define

$$\Delta_t(\phi', \phi) = \frac{1}{B^2 H} \sum_{h=1}^{H} \xi_h(\phi', \phi, o_{t,h}),$$

$$L_t(\phi) = \Delta_t(\phi, \phi) - \mathbb{E}_{\phi' \sim q_t(\cdot|\phi)} [\Delta_t(\phi', \phi)],$$

$$b_t(\phi) = \frac{[\rho_t(\phi) - \max_{s<t} \rho_s(\phi)]_+}{\rho_t(\phi)} \iota.$$

  Let $F_t : \Delta(\Phi) \to \mathbb{R}$ be a convex function. Calculate

$$\rho_{t+1} = \underset{\rho \in \Delta(\Phi)}{\operatorname{argmin}} \left\{ \langle \rho, L_t + 4\gamma L_t^2 + b_t \rangle + F_t(\rho) + \frac{1}{\gamma} \mathrm{KL}(\rho, \rho_t) \right\}, \tag{18}$$

$$q_{t+1}(\phi'|\phi) \propto \exp \left( -\alpha_t(\phi) \sum_{s=1}^{t} \rho_s(\phi) \Delta_s(\phi', \phi) \right) \quad \text{where} \quad \alpha_t(\phi) = \frac{1}{16 \max_{s \le t} \rho_s(\phi)}.$$

---

**Lemma 25.** *With probability at least $1 - \delta$,*

$$\sum_{t=1}^{T} \langle \rho_t, L_t \rangle \le \frac{\log |\Phi|}{\gamma}$$

$$+ \sum_{t=1}^{T} \left( -\frac{1}{2} \langle \rho_t, b_t \rangle + b_t(\phi^\star) + F_t(\delta_{\phi^\star}) - F_t(\rho_{t+1}) - \mathrm{Breg}_{F_t}(\delta_{\phi^\star}, \rho_{t+1}) \right) + O\left(\log(1/\delta)\right).$$

*Proof of Lemma 25.* Observe that the update of $\rho_t$ (Eq. (18)) is in the form specified in Lemma 20. Invoking Lemma 20, we get

$$\sum_{t=1}^{T} \langle \rho_t, L_t \rangle \le \frac{\log |\Phi|}{\gamma} \tag{19}$$

$$+ \sum_{t=1}^{T} \left( L_t(\phi^\star) + 4\gamma L_t(\phi^\star)^2 + b_t(\phi^\star) - \frac{1}{2} \langle \rho_t, b_t \rangle + F_t(\delta_{\phi^\star}) - F_t(\rho_{t+1}) - \mathrm{Breg}_{F_t}(\delta_{\phi^\star}, \rho_{t+1}) \right).$$

By Assumption 6 we have

$$0 \le \mathbb{E}_t[L_t(\phi^\star)^2] = \mathbb{E}_t \left[ \left( \Delta_t(\phi^\star, \phi^\star) - \mathbb{E}_{\phi' \sim q_t(\cdot|\phi^\star)} [\Delta_t(\phi', \phi^\star)] \right)^2 \right]$$

$$\le \mathbb{E}_{\phi' \sim q_t(\cdot|\phi^\star)} \left[ \mathbb{E}_t \left[ (\Delta_t(\phi^\star, \phi^\star) - \Delta_t(\phi', \phi^\star))^2 \right] \right] \quad \text{(Jensen's inequality)}$$

$$\le \mathbb{E}_{\phi' \sim q_t(\cdot|\phi^\star)} \left[ \mathbb{E}_t \left[ (\Delta_t(\mathcal{T}_{M_t}\phi^\star, \phi^\star) - \Delta_t(\phi', \phi^\star))^2 \right] \right]$$
$$(M_t \in \phi^\star \text{ and thus } \mathcal{T}_{M_t}\phi^\star = \phi^\star)$$

$$\le 4\mathbb{E}_{\phi' \sim q_t(\cdot|\phi^\star)} \left[ \mathbb{E}_t \left[ \Delta_t(\phi', \phi^\star) - \Delta_t(\mathcal{T}_{M_t}\phi^\star, \phi^\star) \right] \right] \quad \text{(by Assumption 6)}$$

$$= 4\mathbb{E}_{\phi' \sim q_t(\cdot|\phi^\star)} \left[ \mathbb{E}_t \left[ \Delta_t(\phi', \phi^\star) - \Delta_t(\phi^\star, \phi^\star) \right] \right]$$

$$= -4\mathbb{E}_t[L_t(\phi^\star)].$$

This allows us to apply Lemma 17 with $X_t = -L_t(\phi^\star)$ and $Y_t = \frac{1}{4}X_t^2$, which gives

$$\sum_{t=1}^{T} \left(L_t(\phi^\star) + 4\gamma L_t(\phi^\star)^2\right) \leq \sum_{t=1}^{T} \left(L_t(\phi^\star) + \frac{1}{16}L_t(\phi^\star)^2\right)$$

$$\leq \frac{1}{2}\sum_{t=1}^{T} \mathbb{E}_t[L_t(\phi^\star)] + 36\log(1/\delta) \leq 36\log(1/\delta).$$

Combining this with Eq. (19) finishes the proof. $\qquad\square$

**Lemma 26.** *With probability at least $1 - \delta$,*

$$\sum_{t=1}^{T} \mathbb{E}_{\phi\sim\rho_t}\mathbb{E}_{\phi'\sim q_t(\cdot|\phi)}[\Delta_t(\phi', \phi) - \Delta_t(\mathcal{T}_{M_t}\phi, \phi)] \leq 32\sum_{\phi} \max_{t\leq T}\rho_t(\phi)\log|\Phi| + 72\log(1/\delta).$$

*Proof.* By Assumption 6, we have $\mathcal{T}_{M_t}\phi = \mathcal{T}_{M_{t'}}\phi$ for all $\phi$ and all $t, t' \in [T]$. We denote $\mathcal{T}\phi = \mathcal{T}_{M_t}\phi$ for any $t$. By the exponential weight update, for any $\phi$,

$$\sum_{t=1}^{T}\sum_{\phi'} q_t(\phi'|\phi)\rho_t(\phi)\left(\Delta_t(\phi', \phi) - \Delta_t(\mathcal{T}_{M_t}\phi, \phi)\right)$$

$$= \sum_{t=1}^{T}\sum_{\phi'} q_t(\phi'|\phi)\rho_t(\phi)\left(\Delta_t(\phi', \phi) - \Delta_t(\mathcal{T}\phi, \phi)\right)$$

$$\leq \frac{\log|\Phi|}{\alpha_T(\phi)} + \sum_{t=1}^{T}\sum_{\phi'} \alpha_t(\phi)q_t(\phi'|\phi)\rho_t(\phi)^2\left(\Delta_t(\phi', \phi) - \Delta_t(\mathcal{T}\phi, \phi)\right)^2$$

$$\leq 16\max_{t\leq T}\rho_t(\phi)\log|\Phi| + \frac{1}{16}\sum_{t=1}^{T}\sum_{\phi'} q_t(\phi'|\phi)\rho_t(\phi)\left(\Delta_t(\phi', \phi) - \Delta_t(\mathcal{T}\phi, \phi)\right)^2.$$

Rearranging and summing over $\phi$:

$$\sum_{t=1}^{T} \mathbb{E}_{\phi\sim\rho_t}\mathbb{E}_{\phi'\sim q_t(\cdot|\phi)}\left[\Delta_t(\phi', \phi) - \Delta_t(\mathcal{T}\phi, \phi) - \frac{1}{16}(\Delta_t(\phi', \phi) - \Delta_t(\mathcal{T}\phi, \phi))^2\right]$$

$$\leq 16\sum_{\phi} \max_{t\leq T}\rho_t(\phi)\log|\Phi|. \tag{20}$$

Define

$$X_t = \mathbb{E}_{\phi\sim\rho_t}\mathbb{E}_{\phi'\sim q_t(\cdot|\phi)}\left[\Delta_t(\phi', \phi) - \Delta_t(\mathcal{T}\phi, \phi)\right],$$

$$Y_t = \frac{1}{4}\mathbb{E}_{\phi\sim\rho_t}\mathbb{E}_{\phi'\sim q_t(\cdot|\phi)}\left[(\Delta_t(\phi', \phi) - \Delta_t(\mathcal{T}\phi, \phi))^2\right].$$

By Assumption 6 we have $\mathbb{E}_t[Y_t] \leq \mathbb{E}_t[X_t]$. By Jensen's inequality, $\mathbb{E}_t[X_t^2] \leq 4B^2H\mathbb{E}_t[Y_t] \leq 4B^2H\mathbb{E}_t[X_t]$. Invoking Lemma 17 and using Eq. (20) give

$$\frac{1}{2}\sum_{t=1}^{T} X_t \leq \sum_{t=1}^{T}\left(X_t - \frac{1}{4}Y_t\right) + 36\log(1/\delta) \leq 16\sum_{\phi} \max_{t\leq T}\rho_t(\phi)\log|\Phi| + 36\log(1/\delta),$$

proving the desired inequality.

$\qquad\square$

**Lemma 27.** *With probability at least $1 - \delta$,*

$$\sum_{t=1}^{T} \mathbb{E}_{\phi\sim\rho_t}\left[\Delta_t(\phi, \phi) - \Delta_t(\mathcal{T}_{M_t}\phi, \phi)\right]$$

$$\leq \sum_{t=1}^{T}\left(F_t(\delta_{\phi^\star}) - F_t(\rho_{t+1}) - \text{Breg}_{F_t}(\delta_{\phi^\star}, \rho_{t+1})\right) + O\left(\log^2(|\Phi|/\delta)\right).$$

*Proof.* By Assumption 6, we have $\mathcal{T}_{M_t}\phi = \mathcal{T}_{M_{t'}}\phi$ for all $\phi$ and all $t, t' \in [T]$. We denote $\mathcal{T}\phi = \mathcal{T}_{M_t}\phi$ for any $t$.

$$\mathbb{E}_{\phi \sim \rho_t}[L_t(\phi)] = \mathbb{E}_{\phi \sim \rho_t}[\Delta_t(\phi, \phi) - \mathbb{E}_{\phi' \sim q_t(\cdot|\phi)}[\Delta_t(\phi', \phi)]]$$
$$= \mathbb{E}_{\phi \sim \rho_t}\left[\Delta_t(\phi, \phi) - \Delta_t(\mathcal{T}\phi, \phi) - \left(\mathbb{E}_{\phi' \sim q_t(\cdot|\phi)}[\Delta_t(\phi', \phi)] - \Delta_t(\mathcal{T}\phi, \phi)\right)\right].$$

Combining this with Lemma 25, we get

$$\sum_{t=1}^{T} \mathbb{E}_{\phi \sim \rho_t}\left[\Delta_t(\phi, \phi) - \Delta_t(\mathcal{T}\phi, \phi)\right]$$

$$\leq \frac{\log|\Phi|}{\gamma} + \sum_{t=1}^{T}\left(-\frac{1}{2}\langle\rho_t, b_t\rangle + b_t(\phi^\star) + F_t(\delta_{\phi^\star}) - F_t(\rho_{t+1}) - \mathrm{Breg}_{F_t}(\delta_{\phi^\star}, \rho_{t+1})\right)$$

$$+ O\left(\log(1/\delta)\right) + \sum_{t=1}^{T} \mathbb{E}_{\phi \sim \rho_t}\mathbb{E}_{\phi' \sim q_t(\cdot|\phi)}\left[\Delta_t(\phi', \phi) - \Delta_t(\mathcal{T}\phi, \phi)\right]$$

$$\leq \sum_{t=1}^{T}\left(-\frac{1}{2}\langle\rho_t, b_t\rangle + b_t(\phi^\star) + F_t(\delta_{\phi^\star}) - F_t(\rho_{t+1}) - \mathrm{Breg}_{F_t}(\delta_{\phi^\star}, \rho_{t+1})\right)$$

$$+ O\left(\log^2(|\Phi|/\delta)\right) + 32\sum_{\phi}\max_{t\leq T}\rho_t(\phi)\log|\Phi|. \qquad \text{(by Lemma 26 and the value of } \gamma\text{)}$$

Note that

$$32\log|\Phi|\sum_{\phi}\max_{t\leq T}\rho_t(\phi) = 32\log|\Phi|\sum_{\phi}\left(\rho_1(\phi) + \sum_{t=2}^{T}[\rho_t(\phi) - \max_{s<t}\rho_s(\phi)]_+\right)$$

$$= 32\log|\Phi|\sum_{\phi}\left(\rho_1(\phi) + \sum_{t=2}^{T}\rho_t(\phi)\frac{[\rho_t(\phi) - \max_{s<t}\rho_s(\phi)]_+}{\rho_t(\phi)}\right)$$

$$= \frac{1}{2}\sum_{t=1}^{T}\langle\rho_t, b_t\rangle$$

and

$$\sum_{t=1}^{T}b_t(\phi^\star) = O(\log|\Phi|) \times \sum_{t=1}^{T}\frac{\max_{s\leq t}\rho_s(\phi^\star) - \max_{s\leq t-1}\rho_s(\phi^\star)}{\max_{s\leq t}\rho_s(\phi^\star)}$$

$$\leq O(\log|\Phi|) \times \left(1 + \sum_{t=2}^{T}\ln\frac{\max_{s\leq t}\rho_s(\phi^\star)}{\max_{s\leq t-1}\rho_s(\phi^\star)}\right) \qquad (1 - x \leq \ln\frac{1}{x})$$

$$\leq O\left(\log^2|\Phi|\right).$$

Combining inequalities above proves the lemma.

$\square$

# G  OMITTED DETAILS IN SECTION 4

We define a batched version of Algorithm 1 in Algorithm 4. When the batch size $\tau = 1$, it is exactly Algorithm 1. One can also think of Algorithm 4 as a special case of Algorithm 1 where POSTERIORUPDATE makes a real update only when $t = k\tau$ for $k = 1, 2, \ldots$, and keeps $\rho_{t+1} = \rho_t$ otherwise.

---

**Algorithm 4** General Batched Framework

---

**Input:** Partition set $\Phi$ and its union $\Psi$ (defined in Section 2.1). Batch size $\tau$.
$\rho_1(\phi) = 1/|\Phi|$, $\forall \phi \in \Phi$.
**for** $k = 1, 2, \ldots, K$ **do**

  Set $p_k, \nu_k$ as the solution of the following minimax optimization (defined in Eq. (2)):

$$\min_{p \in \Delta(\Pi)} \max_{\nu \in \Delta(\Psi)} \mathsf{AIR}_\eta^{\Phi, D}(p, \nu; \rho_k). \tag{21}$$

  Execute $\pi_k$ in rounds $t \in \{(k-1)\tau + 1, \ldots, k\tau\} = \mathcal{I}_k$ and receive observations $(o_t)_{t \in \mathcal{I}_k}$.
  Update $\rho_{k+1} = \text{POSTERIORUPDATE}(\nu_k, \rho_k, \pi_k, (o_t)_{t \in \mathcal{I}_k})$. $\tag{22}$

---

## G.1  ASSUMPTION REDUCTIONS

*Proof of Lemma 8.* In the stochastic setting, by Assumption 1 we have $f_\phi(s, a) = Q^\star(s, a; M)$ and $f_\phi(s) = V^\star(s; M)$ for any $M \in \phi$. Hence

$$\mathbb{E}^{\pi, M}[\ell_h(\phi; o_h)] = \mathbb{E}^{\pi, M}[f_\phi(s_h, a_h) - r_h - f_\phi(s_{h+1})]$$
$$= \mathbb{E}^{\pi, M}[Q^\star(s_h, a_h; M) - r_h - V^\star(s_{h+1}; M)] = 0.$$

In the hybrid setting, we have by Assumption 2 and Assumption 3 that $f_\phi(s, a; R) = Q^{\pi_\phi}(s, a; (P, R))$ and $f_\phi(s; R) = V^{\pi_\phi}(s; (P, R))$ for any $P \in \phi$. Hence, for any $j \in [d]$, defining $R'$ such that $R'(s, a) = \varphi(s, a)_j$, we have for $(P, R) \in \phi$,

$$\mathbb{E}^{\pi, (P, R)}[\ell_h(\phi; o_h)_j] = \mathbb{E}^{\pi, P}[f_\phi(s_h, a_h; R') - R'(s_h, a_h) - f_\phi(s_{h+1}; R')]$$
$$= \mathbb{E}^{\pi, P}[Q^{\pi_\phi}(s_h, a_h; (P, R')) - R'(s, a) - V^{\pi_\phi}(s_{h+1}; (P, R'))] = 0.$$

Finally, note that in the stochastic setting $M_t = M^\star$, and in the hybrid setting $P_t = P^\star$, so the additional assumption always holds. $\qquad\square$

*Proof of Lemma 12.* In the stochastic setting, with Assumption 1 and the Bellman completeness assumption (Definition 9), for any $M = (P, R)$, we define $\mathcal{T}_M \phi \in \Phi$ as the $\phi'$ such that

$$f_{\phi'}(s, a) = R(s, a) + \mathbb{E}_{s' \sim P(\cdot|s, a)}[f_\phi(s')].$$

By Definition 9, such $\phi'$ always exists.

In the hybrid setting, with Assumption 2, Assumption 3 and Assumption 4 and the Bellman completeness assumption (Definition 10), for any $M = (P, R)$, we define $\mathcal{T}_M \phi \in \Phi$ to be the $\phi'$ such that $\pi_{\phi'} = \pi_\phi$ and for all $\tilde{R}$,

$$f_{\phi'}(s, a; \tilde{R}) = \tilde{R}(s, a) + \mathbb{E}_{s' \sim P(\cdot|s, a)}[f_\phi(s'; \tilde{R})].$$

By Definition 10, such $\phi'$ always exists.

Below, with a slight overload of notation, we denote in the hybrid setting $f_\phi(s_h, a_h) \in \mathbb{R}^d$ as the vector $(f_\phi(s_h, a_h; \boldsymbol{e}_j))_{j \in [d]}$ and $f_\phi(s_{h+1}) \in \mathbb{R}^d$ as the vector $\mathbb{E}_{a \sim \pi_\phi(\cdot|s_{h+1})}[(f_\phi(s_{h+1}, a; \boldsymbol{e}_j))_{j \in [d]}]$. Furthermore, we use the notation $y_h$ to denote $r_h \in \mathbb{R}$ in the stochastic setting, and $\varphi(s_h, a_h) \in \mathbb{R}^d$ in the hybrid setting.

Then we have by our choice of $\xi_h$:

$$
\begin{aligned}
&\mathbb{E}^{\pi,M}\left[\xi_h(\phi',\phi;o_h)-\xi_h(\mathcal{T}_M\phi,\phi;o_h)\right] \\
&= \mathbb{E}^{\pi,M}\left[\|f_{\phi'}(s_h,a_h)-y_h-f_\phi(s_{h+1})\|^2 - \|f_{\mathcal{T}_M\phi}(s_h,a_h)-y_h-f_\phi(s_{h+1})\|^2\right] \\
&= \mathbb{E}^{\pi,M}\left[\|f_{\phi'}(s_h,a_h)-f_{\mathcal{T}_M\phi}(s_h,a_h)\|^2\right] \\
&\qquad + 2\cdot\mathbb{E}^{\pi,M}\left[\langle f_{\phi'}(s_h,a_h)-f_{\mathcal{T}_M\phi}(s_h,a_h), f_{\mathcal{T}_M\phi}(s_h,a_h)-y_h-f_\phi(s_{h+1})\rangle\right] \\
&= \mathbb{E}^{\pi,M}\left[\|f_{\phi'}(s_h,a_h)-f_{\mathcal{T}_M\phi}(s_h,a_h)\|^2\right],
\end{aligned}
\tag{23}
$$

where the last line follows from $\mathbb{E}^{\pi,M}[y_h+f_\phi(s_{h+1})]=f_{\mathcal{T}_M\phi}(s_h,a_h)$ by definition of $\mathcal{T}_M\phi$. On the other hand,

$$
\begin{aligned}
&\mathbb{E}^{\pi,M}\left[\left(\xi_h(\phi',\phi;o_h)-\xi_h(\mathcal{T}_M\phi,\phi;o_h)\right)^2\right] \\
&= \mathbb{E}^{\pi,M}\left[\left(\|f_{\phi'}(s_h,a_h)-y_h-f_\phi(s_{h+1})\|^2 - \|f_{\mathcal{T}_M\phi}(s_h,a_h)-y_h-f_\phi(s_{h+1})\|^2\right)^2\right] \\
&= \mathbb{E}^{\pi,M}\left[\langle f_{\phi'}(s_h,a_h)-f_{\mathcal{T}_M\phi}(s_h,a_h),\ f_{\mathcal{T}_M\phi}(s_h,a_h)+f_{\phi'}(s_h,a_h)-2y_h-2f_\phi(s_{h+1})\rangle^2\right] \\
&\leq 4B^2\mathbb{E}^{\pi,M}\left[\|f_{\phi'}(s_h,a_h)-f_{\mathcal{T}_M\phi}(s_h,a_h)\|^2\right],
\end{aligned}
$$

where $B^2=1$ in the stochastic setting and $B^2=d$ in the hybrid setting. Combining both finishes the proof.

$\square$

## G.2 Bounds on **Est**

With the specific form of divergence

$$
D^\pi(\nu\|\rho) = \mathbb{E}_{M\sim\nu}\mathbb{E}_{o\sim M(\cdot|\pi)}\left[\mathrm{KL}\left(\nu_{\boldsymbol{\phi}}(\cdot|\pi,o),\rho\right) + \mathbb{E}_{\phi\sim\rho}\left[\overline{D}^\pi(\phi\|M)\right]\right],
\tag{24}
$$

the estimation term in Eq. (6) for an epoch algorithm with epoch length $\tau'$ and $K$ epochs is given by

**Lemma 28.** **Est** *in Eq. (6) can be written as*

$$
\mathbf{Est} = \sum_{t=1}^{T}\mathbb{E}_{\pi\sim p_t}\mathbb{E}_{o\sim M_t(\cdot|\pi)}\left[\log\left(\frac{\nu_t(\phi^\star|\pi,o)}{\rho_t(\phi^\star)}\right)\right] + \sum_{t=1}^{T}\mathbb{E}_{\pi\sim p_t}\mathbb{E}_{\phi\sim\rho_t}\left[\overline{D}^\pi(\phi\|M_t)\right].
\tag{25}
$$

*Proof of Lemma 28.* From the definition of divergence in Eq. (24) and Eq. (25), let $\delta_{\phi^\star}\in\Delta(\Phi)$ be the Kronecker delta function centered at $\phi^\star$. Then

$$
\begin{aligned}
\mathbf{Est} &= \sum_{t=1}^{T}\Bigg(\log\left(\frac{1}{\rho_t(\phi^\star)}\right) + \mathbb{E}_{\pi\sim p_t}\mathbb{E}_{\phi\sim\rho_t}\left[\overline{D}^\pi(\phi\|M_t)\right] \\
&\qquad\qquad - \mathbb{E}_{\pi\sim p_t}\mathbb{E}_{o\sim M_t(\cdot|\pi)}\left[\mathrm{KL}\left(\delta_{\phi^\star},(\nu_t)_{\boldsymbol{\phi}}(\cdot|\pi,o)\right)\right]\Bigg) \\
&= \sum_{t=1}^{T}\mathbb{E}_{\pi\sim p_t}\mathbb{E}_{o\sim M_t(\cdot|\pi)}\left[\log\left(\frac{\nu_t(\phi^\star|\pi,o)}{\rho_t(\phi^\star)}\right)\right] + \sum_{t=1}^{T}\mathbb{E}_{\pi\sim p_t}\mathbb{E}_{\phi\sim\rho_t}\left[\overline{D}^\pi(\phi\|M_t)\right]
\end{aligned}
\tag{26}
$$

where the first equality uses the fact that for any $\rho$,

$$
\mathrm{Breg}_{D^\pi(\cdot\|\rho)}(\nu,\nu') = \mathbb{E}_{M\sim\nu}\mathbb{E}_{o\sim M(\cdot|\pi)}\left[\mathrm{KL}\left(\nu_{\boldsymbol{\phi}}(\cdot|\pi,o),\nu'_{\boldsymbol{\phi}}(\cdot|\pi,o)\right)\right].
$$

$\square$

*Proof of Theorem 7.* With abuse of notation, we use $p_t, \nu_t, \rho_t$ to denote the $p_k, \nu_k, \rho_k$ where $k$ is the epoch where episode $t$ lies. We start from the estimation term in Eq. (25) using the definition of $\overline{D}$:

$$
\begin{aligned}
\textbf{Est} &= \sum_{t=1}^{T} \mathbb{E}_{\pi \sim p_t} \mathbb{E}_{o \sim M_t(\cdot|\pi)} \left[ \log\left( \frac{\nu_t(\phi^\star|\pi, o)}{\rho_t(\phi^\star)} \right) \right] + \frac{1}{B^2 H} \sum_{t=1}^{T} \mathbb{E}_{\pi \sim p_t} \mathbb{E}_{\phi \sim \rho_t} \left[ \max_{j \in [N]} \sum_{h=1}^{H} \left( \mathbb{E}^{\pi, M_t} [\ell_h(\phi; o_h)_j] \right)^2 \right] \\
&= \sum_{k=1}^{K} \mathbb{E}_{\pi \sim p_k} \sum_{t \in \mathcal{I}_k} \mathbb{E}_{o \sim M_t(\cdot|\pi)} \left[ \log\left( \frac{\nu_k(\phi^\star|\pi, o)}{\rho_k(\phi^\star)} \right) \right] \\
&\qquad + \frac{1}{B^2 H} \sum_{k=1}^{K} \mathbb{E}_{\pi \sim p_k} \mathbb{E}_{\phi \sim \rho_k} \left[ \sum_{t \in \mathcal{I}_k} \max_{j \in [N]} \sum_{h=1}^{H} \left( \mathbb{E}^{\pi, M_t} [\ell_h(\phi; o_h)_j] \right)^2 \right].
\end{aligned}
$$

Applying Lemma 24 with $F_t(\rho) = \mathrm{KL}(\rho, (\nu_k)_{\boldsymbol{\phi}}(\cdot|\pi_k, o_t))$ for $t \in \mathcal{I}_k$, we get

$$
\begin{aligned}
\mathbb{E}[\textbf{Est}] &\leq \mathbb{E}\left[ \sum_{k=1}^{K} \mathbb{E}_{\pi \sim p_k} \sum_{t \in \mathcal{I}_k} \mathbb{E}_{o \sim M_t(\cdot|\pi)} \left[ \log\left( \frac{\nu_k(\phi^\star|\pi, o)}{\rho_k(\phi^\star)} \right) \right] \right] + O\left( N \log(|\Phi|) T^{\frac{1}{3}} \right) \\
&\qquad + \mathbb{E}\left[ \sum_{k=1}^{K} \sum_{t \in \mathcal{I}_k} \left( \log\left( \frac{1}{\nu_k(\phi^\star|\pi_k, o_t)} \right) - \mathrm{KL}(\rho_{k+1}, (\nu_k)_{\boldsymbol{\phi}}(\pi_k, o_t)) - \log\left( \frac{1}{\rho_{k+1}(\phi^\star)} \right) \right) \right] \\
&\leq \mathbb{E}\left[ \sum_{k=1}^{K} \sum_{t \in \mathcal{I}_k} \left( \log\left( \frac{\nu_k(\phi^\star|\pi_k, o_t)}{\rho_k(\phi^\star)} \right) + \log\left( \frac{\rho_{k+1}(\phi^\star)}{\nu_k(\phi^\star|\pi_k, o_t)} \right) \right) \right] + O\left( N \log(|\Phi|) T^{\frac{1}{3}} \right) \\
&\leq \tau \log\left( \frac{1}{\rho_1(\phi^\star)} \right) + O\left( N \log(|\Phi|) T^{\frac{1}{3}} \right) \\
&= O\left( N \log(|\Phi|) T^{\frac{1}{3}} \right).
\end{aligned}
$$

$\square$

*Proof of Theorem 11.* We start from the estimation term in Eq. (25), using the definition of $\overline{D}$:

$$
\begin{aligned}
\textbf{Est} &= \sum_{t=1}^{T} \mathbb{E}_{\pi \sim p_t} \mathbb{E}_{o \sim M_t(\cdot|\pi)} \left[ \log\left( \frac{\nu_t(\phi^\star|\pi, o)}{\rho_t(\phi^\star)} \right) \right] \\
&\qquad + \frac{1}{B^2 H} \sum_{t=1}^{T} \mathbb{E}_{\pi \sim p_t} \mathbb{E}_{\phi \sim \rho_t} \left[ \sum_{h=1}^{H} \mathbb{E}^{\pi, M_t} [\xi_h(\phi, \phi; o_h) - \xi_h(\mathcal{T}_{M_t} \phi, \phi; o_h)] \right].
\end{aligned}
$$

Applying Lemma 27 with $F_t(\rho) = \mathrm{KL}(\rho, (\nu_t)_{\boldsymbol{\phi}}(\cdot|\pi_t, o_t))$, we get

$$
\begin{aligned}
\mathbb{E}[\textbf{Est}] &\leq \mathbb{E}\left[ \sum_{t=1}^{T} \mathbb{E}_{\pi \sim p_t} \mathbb{E}_{o \sim M_t(\cdot|\pi)} \left[ \log\left( \frac{\nu_t(\phi^\star|\pi, o)}{\rho_t(\phi^\star)} \right) \right] \right] + O\left( \log^2 |\Phi| \right) \\
&\qquad + \mathbb{E}\left[ \sum_{t=1}^{T} \left( \log\left( \frac{1}{\nu_t(\phi^\star|\pi_t, o_t)} \right) - \mathrm{KL}(\rho_{t+1}, (\nu_t)_{\boldsymbol{\phi}}(\pi_t, o_t)) - \log\left( \frac{1}{\rho_{t+1}(\phi^\star)} \right) \right) \right] \\
&\leq \mathbb{E}\left[ \sum_{t=1}^{T} \log\left( \frac{\rho_{t+1}(\phi^\star)}{\rho_t(\phi^\star)} \right) \right] + O\left( \log^2 |\Phi| \right) = O\left( \log^2 |\Phi| \right).
\end{aligned}
$$

$\square$

# H  RELATING dig-dec TO EXISTING COMPLEXITIES IN THE STOCHASTIC SETTING

## H.1  SUPPORTING LEMMAS

**Lemma 29.** *Suppose that* $(\mathcal{M}, \Phi)$ *satisfy [Assumption 5](#) with estimation function* $\ell_h(\phi; o_h) = f_\phi(s_h, a_h) - r_h - f_\phi(s_{h+1})$. *Furthermore, assume that* $(\mathcal{M}, \Phi)$ *is Bellman complete ([Definition 9](#)).* *Then [Assumption 6](#) holds with* $\xi_h(\phi', \phi; o_h) = (f_{\phi'}(s_h, a_h) - r_h - f_\phi(s_{h+1}))^2$ *and*

$$\mathsf{dig\text{-}dec}_\eta^{\Phi, \overline{D}_{\mathsf{sq}}} \leq \mathsf{dig\text{-}dec}_\eta^{\Phi, \overline{D}_{\mathsf{av}}}.$$

*Proof.* It suffices to show that $\overline{D}_{\mathsf{av}}^\pi(\phi \| M) \leq \overline{D}_{\mathsf{sq}}^\pi(\phi \| M)$ for any $\pi, \phi, M$:

$$
\begin{aligned}
\overline{D}_{\mathsf{sq}}^\pi(\phi \| M) &= \frac{1}{B^2 H} \sum_{h=1}^H \mathbb{E}^{\pi, M} \left[ \xi_h(\phi, \phi; o_h) - \xi_h(\mathcal{T}_M \phi, \phi; o_h) \right] \\
&= \frac{1}{B^2 H} \sum_{h=1}^H \mathbb{E}^{\pi, M} \left[ \left( f_\phi(s_h, a_h) - f_{\mathcal{T}_M \phi}(s_h, a_h) \right)^2 \right] \\
&\qquad\qquad\qquad\qquad\qquad\qquad \text{(by the same calculation as Eq. (23))} \\
&\geq \frac{1}{B^2 H} \sum_{h=1}^H \left( \mathbb{E}^{\pi, M} \left[ f_\phi(s_h, a_h) - f_{\mathcal{T}_M \phi}(s_h, a_h) \right] \right)^2 \qquad \text{(Jensen's inequality)} \\
&= \frac{1}{B^2 H} \sum_{h=1}^H \left( \mathbb{E}^{\pi, M} \left[ f_\phi(s_h, a_h) - r_h - f_\phi(s_{h+1}) \right] \right)^2 \\
&= \overline{D}_{\mathsf{av}}(\phi \| M).
\end{aligned}
$$

$\square$

## H.2  RELATING dig-dec TO o-dec

*Proof of [Theorem 13](#).* In the stochastic setting, by definition,

$$\mathsf{dig\text{-}dec}_\eta^{\Phi, \overline{D}} = \max_{\rho \in \Delta(\Phi)} \min_{p \in \Delta(\Pi)} \max_{\nu \in \Delta(\mathcal{M})}$$

$$\mathbb{E}_{\pi \sim p} \mathbb{E}_{M \sim \nu} \left[ V_M(\pi_M) - V_M(\pi) - \frac{1}{\eta} \mathbb{E}_{o \sim M(\cdot | \pi)} \left[ \mathrm{KL}(\nu_\phi(\cdot | \pi, o), \rho) \right] - \frac{1}{\eta} \mathbb{E}_{\phi \sim \rho} \left[ \overline{D}^\pi(\phi \| M) \right] \right]$$

and

$$\mathsf{o\text{-}dec}_\eta^{\Phi, \overline{D}} = \max_{\rho \in \Delta(\Phi)} \min_{p \in \Delta(\Pi)} \max_{\nu \in \Delta(\mathcal{M})} \mathbb{E}_{\pi \sim p} \mathbb{E}_{M \sim \nu} \mathbb{E}_{\phi \sim \rho} \left[ V_\phi(\pi_\phi) - V_M(\pi) - \frac{1}{\eta} \overline{D}^\pi(\phi \| M) \right].$$

For any $\rho, p, \nu$, we have

$$\mathbb{E}_{\pi \sim p} \mathbb{E}_{M \sim \nu} \left[ V_M(\pi_M) - V_M(\pi) - \frac{1}{\eta} \mathbb{E}_{o \sim M(\cdot | \pi)} \left[ \mathrm{KL}(\nu_\phi(\cdot | \pi, o), \rho) \right] - \frac{1}{\eta} \mathbb{E}_{\phi \sim \rho} \left[ \overline{D}^\pi(\phi \| M) \right] \right]$$

$$= \underbrace{\mathbb{E}_{M \sim \nu} \mathbb{E}_{\phi \sim \rho} \left[ V_M(\pi_M) - V_\phi(\pi_\phi) \right] - \frac{1}{\eta} \mathrm{KL}(\nu_\phi, \rho) - \frac{1}{\eta} \mathbb{E}_{\pi \sim p} \mathbb{E}_{M \sim \nu} \mathbb{E}_{o \sim M(\cdot | \pi)} [\mathrm{KL}(\nu_\phi(\cdot | \pi, o), \nu_\phi)]}_{\mathbf{term1}}$$

$$+ \mathbb{E}_{\pi \sim p} \mathbb{E}_{M \sim \nu} \mathbb{E}_{\phi \sim \rho} \left[ V_\phi(\pi_\phi) - V_M(\pi) - \frac{1}{\eta} \overline{D}^\pi(\phi \| M) \right].$$

To bound **term1**, observe that

$$\mathbb{E}_{M \sim \nu} \left[ V_M(\pi_M) \right] = \mathbb{E}_{\phi \sim \nu} \left[ V_\phi(\pi_\phi) \right].$$

Thus,

$$\mathbf{term1} = \mathbb{E}_{\phi\sim\nu}[V_\phi(\pi_\phi)] - \mathbb{E}_{\phi\sim\rho}[V_\phi(\pi_\phi)] - \frac{1}{\eta}\mathrm{KL}(\nu_{\boldsymbol{\phi}},\rho) \le \eta. \qquad \text{(Lemma 19)}$$

This implies

$\mathsf{dig\text{-}dec}_\eta^{\Phi,\overline{D}}$

$$\le \eta + \max_{\rho\in\Delta(\Phi)} \min_{p\in\Delta(\Pi)} \max_{\nu\in\Delta(\mathcal{M})} \mathbb{E}_{\pi\sim p}\mathbb{E}_{M\sim\nu}\mathbb{E}_{\phi\sim\rho}\left[V_\phi(\pi_\phi) - V_M(\pi) - \frac{1}{\eta}\overline{D}^\pi(\phi\|M) - \frac{1}{\eta}\mathbb{E}_{o\sim M(\cdot|\pi)}[\mathrm{KL}(\nu_{\boldsymbol{\phi}}(\cdot|\pi,o),\nu_{\boldsymbol{\phi}})]\right]$$

$$\le \eta + \max_{\rho\in\Delta(\Phi)} \min_{p\in\Delta(\Pi)} \max_{\nu\in\Delta(\mathcal{M})} \mathbb{E}_{\pi\sim p}\mathbb{E}_{M\sim\nu}\mathbb{E}_{\phi\sim\rho}\left[V_\phi(\pi_\phi) - V_M(\pi) - \frac{1}{\eta}\overline{D}^\pi(\phi\|M)\right]$$

$$= \eta + \mathsf{o\text{-}dec}_\eta^{\Phi,\overline{D}}.$$

□

## H.3   RELATING dig-dec TO BILINEAR RANK

Bilinear rank is a complexity measure proposed in [DKL+21]. It is defined as the following.

**Assumption 7** (Bilinear class [DKL+21])**.** *A model class $\mathcal{M}$ and its associated $\Phi$ satisfying Assumption 1 is a bilinear class with rank $d$ if there exists functions $X_h : \Phi \times \mathcal{M} \to \mathbb{R}^d$ and $W_h : \Phi \times \mathcal{M} \to \mathbb{R}^d$ for all $h \in [H]$ such that*

1. *For $M \in \phi$, it holds that $W_h(\phi; M) = 0$.*
2. *For any $\phi \in \Phi$ and any $M \in \mathcal{M}$,*

$$|V_\phi(\pi_\phi) - V_M(\pi_\phi)| \le \sum_{h=1}^{H} |\langle X_h(\phi; M), W_h(\phi; M)\rangle|.$$

3. *For every policy $\pi$, there exists an estimation policy $\pi^{\mathrm{est}}$. Also, there exists a discrepancy function $\ell_h : \Phi \times \mathcal{O} \to \mathbb{R}$ such that for any $\phi', \phi \in \Phi$ and any $M \in \mathcal{M}$,*

$$|\langle X_h(\phi'; M), W_h(\phi; M)\rangle| = \left|\mathbb{E}^{\pi_{\phi'}\circ_h\pi_{\phi'}^{\mathrm{est}},M}[\ell_h(\phi;o_h)]\right|$$

    *where $o_h = (s_h, a_h, r_h, s_{h+1})$ and $\pi \circ_h \pi^{\mathrm{est}}$ denotes a policy that plays $\pi$ for the first $h-1$ steps and plays policy $\pi^{\mathrm{est}}$ at the $h$-th step.*

*We call it an on-policy bilinear class if $\pi^{\mathrm{est}} = \pi$ for all $\pi \in \Pi$, and otherwise an off-policy bilinear class. As in prior work [DKL+21, FKQR21], for the off-policy case, we assume $|\mathcal{A}|$ is finite, and $\pi^{\mathrm{est}}$ is always $\mathrm{unif}(\mathcal{A})$. We denote by $\pi^\alpha$ the policy that in every step $h = 1, \ldots, H$ chooses $\pi$ with probability $1 - \frac{\alpha}{H}$ and chooses $\pi^{\mathrm{est}}$ with probability $\frac{\alpha}{H}$.*

**Lemma 30.** *Bilinear classes (Assumption 7) satisfy Assumption 5.*

*Proof of Lemma 30.* For any $\phi' \in \Phi$ and any $(M, \phi)$ such that $M \in \phi$,

$$\left|\mathbb{E}^{\pi_{\phi'}\circ_h\pi_{\phi'}^{\mathrm{est}},M}[\ell_h(\phi;o_h)]\right| = |\langle X_h(\phi'; M), W_h(\phi, M)\rangle| \qquad \text{(by Assumption 7.3)}$$

$$= 0. \qquad \text{(by Assumption 7.1 and that } M \in \phi)$$

□

**Lemma 31.** *Let $(\mathcal{M}, \Phi)$ be a bilinear class (Assumption 7). Then*

- $\mathsf{dig\text{-}dec}_\eta^{\Phi,\overline{D}_{\mathsf{av}}} \le O(B^2 H^2 d\eta)$ *in the on-policy case.*
- $\mathsf{dig\text{-}dec}_\eta^{\Phi,\overline{D}_{\mathsf{av}}} \le O(\sqrt{B^2 H^3 d\eta})$ *in the off-policy case.*

*Proof of Lemma 31.* We first use Theorem 13 to bound $\mathsf{dig\text{-}dec}_\eta^{\Phi,\overline{D}_{\mathsf{av}}}$ by $\mathsf{o\text{-}dec}_\eta^{\Phi,\overline{D}_{\mathsf{av}}} + \eta$, and then use Lemma 32 to relate $\mathsf{o\text{-}dec}_\eta^{\Phi,\overline{D}_{\mathsf{av}}}$ to bilinear rank. □

**Lemma 32** (Proposition 2.2 of [FGQ$^+$23])**.** *Let* $(\mathcal{M}, \Phi)$ *be a bilinear class (Assumption 7). Then*

- o-dec$_\eta^{\Phi, \overline{D}_{\mathsf{av}}} \leq O(B^2 H^2 d\eta)$ *in the on-policy case;*
- o-dec$_\eta^{\Phi, \overline{D}_{\mathsf{av}}} \leq O(\sqrt{B^2 H^3 d|\mathcal{A}|\eta})$ *in the off-policy case.*[4]

## H.4 RELATING dig-dec TO BELLMAN-ELUDER DIMENSION

**Lemma 33.** *Let* $\ell_h(\phi; o_h) = f_\phi(s_h, a_h) - r_h - f_\phi(s_{h+1})$, *and let* $\overline{D}_{\mathsf{av}}$ *be defined with respect to this* $\ell_h$. *Then*

- *If the* $Q$-*type Bellman-eluder dimension of* $(\mathcal{M}, \Phi)$ *is bounded by* $d$, *then* dig-dec$_\eta^{\Phi, \overline{D}_{\mathsf{av}}} \leq O(Hd\eta)$.
- *If the* $V$-*type Bellman-eluder dimension of* $(\mathcal{M}, \Phi)$ *is bounded by* $d$, *then* dig-dec$_\eta^{\Phi, \overline{D}_{\mathsf{av}}} \leq O(H\sqrt{d|\mathcal{A}|\eta})$.

*Proof.* We first consider the $Q$-type setting. Define $g_h(\phi', \phi; M) = \mathbb{E}^{\pi_{\phi'}, M}[\ell_h(\phi; o_h)]$. For a fixed $M$, we have by the AM-GM inequality

$$\mathbb{E}_{\phi \sim \rho}[g_h(\phi, \phi; M)] \leq \frac{\lambda}{4} \cdot \mathbb{E}_{\phi \sim \rho}\left[\frac{g_h(\phi, \phi; M)^2}{\mathbb{E}_{\phi' \sim \rho}[g_h(\phi', \phi; M)^2]}\right] + \frac{1}{\lambda}\mathbb{E}_{\phi \sim \rho}\mathbb{E}_{\phi' \sim \rho}[g_h(\phi', \phi; M)^2]$$

for any $\lambda > 0$, implying that

o-dec$_\eta^{\Phi, \overline{D}_{\mathsf{av}}}$

$$= \max_\rho \min_p \max_\nu \mathbb{E}_{\pi \sim p}\mathbb{E}_{\phi \sim \rho}\mathbb{E}_{M \sim \nu}\left[V_\phi(\pi_\phi) - V_M(\pi) - \frac{1}{\eta B^2 H}\sum_{h=1}^{H}\left(\mathbb{E}^{\pi, M}[\ell_h(\phi; o_h)]\right)^2\right]$$

$$\leq \max_\rho \max_\nu \mathbb{E}_{\phi' \sim \rho}\mathbb{E}_{\phi \sim \rho}\mathbb{E}_{M \sim \nu}\left[V_\phi(\pi_\phi) - V_M(\pi_\phi) - \frac{1}{\eta B^2 H}\sum_{h=1}^{H}\left(\mathbb{E}^{\pi_{\phi'}, M}[\ell_h(\phi; o_h)]\right)^2\right]$$

$$= \max_\rho \max_\nu \mathbb{E}_{\phi' \sim \rho}\mathbb{E}_{\phi \sim \rho}\mathbb{E}_{M \sim \nu}\left[\sum_{h=1}^{H} g_h(\phi, \phi; M) - \frac{1}{\eta B^2 H}\sum_{h=1}^{H} g_\phi(\phi', \phi, M)^2\right]$$

$$\leq \frac{\eta B^2 H}{4}\max_\rho \max_\nu \sum_{h=1}^{H}\mathbb{E}_{\phi \sim \rho}\left[\frac{g_h(\phi, \phi; M)^2}{\mathbb{E}_{\phi' \sim \rho}[g_h(\phi', \phi; M)^2]}\right].$$

The rest of the proof goes through standard steps. First, bound $\mathbb{E}_{\phi \sim \rho}\left[\frac{g_h(\phi, \phi; M)^2}{\mathbb{E}_{\phi' \sim \rho}[g_h(\phi', \phi; M)^2]}\right]$ by the *disagreement coefficient* of the function class $\mathcal{F}_M = \{f_\phi - \mathcal{T}_M f_\phi : \phi \in \Phi\}$ where $(\mathcal{T}_M f)(s, a) \triangleq R(s, a) + \mathbb{E}_{s' \sim P(\cdot|s,a)}[f(s')]$ under the probability measure $\mathbb{E}_{\phi \sim \rho}[d_h^{\pi_\phi, M}]$ (Lemma E.2 of [FKQR21]). Taking a maximum over $\rho$, this can be further bounded by the *distributional eluder dimension* of $\mathcal{F}_M$ over the probability measure space $\mathcal{D}_{\Phi, M} = \{d_h^{\pi_\phi, M} : \phi \in \Phi\}$ (Lemma 6.1 of [FKQR21] and Theorem 2.10 of [FRSLX21]), which is equivalent to the *$Q$-type Bellman-eluder dimension* in $M$ defined in [JLM21]. This then allows us to bound o-dec$_\eta^{\Phi, \overline{D}_{\mathsf{av}}} \leq \eta d B^2 H^2$, where $d$ is the maximum $Q$-type Bellman-eluder dimension over all possible $M$.

Next, we consider the $V$-type setting. Define $g_h(\phi', \phi; M) = \mathbb{E}^{\pi_{\phi'} \circ_h \pi_\phi, M}[\ell_h(\phi; o_h)]$. For a fixed $M$, we have by the AM-GM inequality

$$\mathbb{E}_{\phi \sim \rho}[g_h(\phi, \phi; M)] \leq \frac{\lambda}{4} \cdot \mathbb{E}_{\phi \sim \rho}\left[\frac{g_h(\phi, \phi; M)^2}{\mathbb{E}_{\phi' \sim \rho}[g_h(\phi', \phi; M)^2]}\right] + \frac{1}{\lambda}\mathbb{E}_{\phi \sim \rho}\mathbb{E}_{\phi' \sim \rho}[g_h(\phi', \phi; M)^2]$$

---

[4]In [FGQ$^+$23], the bounds on o-dec$_\eta^{\Phi, \overline{D}_{\mathsf{av}}}$ have different scaling of $B, H$ than ours. This is because their average estimation error does not involve the normalization factor $\frac{1}{B^2 H}$ like ours (Theorem 7). We normalize $\overline{D}_{\mathsf{av}}$ to keep the two information gain terms in Dig-DEC of the same unit. Equivalently, one can view our $\eta$ as a scaled version of theirs.

for any $\lambda > 0$. Below, let $\pi^\alpha$ be the policy that in every step $h$, with probability $1 - \frac{\alpha}{H}$ executes policy $\pi$, and with probability $\frac{\alpha}{H}$ executes $\mathrm{unif}(\mathcal{A})$. Then we have

$$\mathsf{o\text{-}dec}_\eta^{\Phi, \overline{D}_{\mathsf{av}}}$$

$$= \max_\rho \min_p \max_\nu \mathbb{E}_{\pi \sim p} \mathbb{E}_{\phi \sim \rho} \mathbb{E}_{M \sim \nu} \left[ V_\phi(\pi_\phi) - V_M(\pi) - \frac{1}{\eta B^2 H} \sum_{h=1}^H \left( \mathbb{E}^{\pi, M} [\ell_h(\phi; o_h)] \right)^2 \right]$$

$$\leq \max_\rho \max_\nu \mathbb{E}_{\phi' \sim \rho} \mathbb{E}_{\phi \sim \rho} \mathbb{E}_{M \sim \nu} \left[ V_\phi(\pi_\phi) - V_M(\pi_\phi^\alpha) - \frac{1}{\eta B^2 H} \sum_{h=1}^H \left( \mathbb{E}^{\pi_{\phi'}^\alpha, M} [\ell_h(\phi; o_h)] \right)^2 \right]$$

$$\leq \alpha + \max_\rho \max_\nu \mathbb{E}_{\phi' \sim \rho} \mathbb{E}_{\phi \sim \rho} \mathbb{E}_{M \sim \nu} \left[ V_\phi(\pi_\phi) - V_M(\pi_\phi) - \frac{1}{\eta B^2 H} \cdot \frac{\alpha}{3H|\mathcal{A}|} \sum_{h=1}^H \left( \mathbb{E}^{\pi_{\phi'} \circ_h \pi_\phi, M} [\ell_h(\phi; o_h)] \right)^2 \right]$$

$$= \alpha + \max_\rho \max_\nu \mathbb{E}_{\phi' \sim \rho} \mathbb{E}_{\phi \sim \rho} \mathbb{E}_{M \sim \nu} \left[ \sum_{h=1}^H g_h(\phi, \phi; M) - \frac{\alpha}{3\eta B^2 H^2 |\mathcal{A}|} \sum_{h=1}^H g_\phi(\phi', \phi, M)^2 \right]$$

$$\leq \alpha + \frac{3\eta B^2 H^2 |\mathcal{A}|}{4\alpha} \max_\rho \max_\nu \sum_{h=1}^H \mathbb{E}_{\phi \sim \rho} \left[ \frac{g_h(\phi, \phi; M)^2}{\mathbb{E}_{\phi' \sim \rho} [g_h(\phi', \phi; M)^2]} \right].$$

where the second inequality is because with probability at least $\left(1 - \frac{\alpha}{H}\right)^{h-1} \frac{\alpha}{H|\mathcal{A}|} \geq \frac{\alpha}{3H|\mathcal{A}|}$, the policy $\pi_{\phi'}^\alpha$ chooses the same actions in steps $1, \ldots, h$ as the policy $\pi_{\phi'} \circ_h \pi_\phi$. Similar to the $Q$-type analysis, the last expression can be related to $V$-type Bellman-eluder dimension (notice that the definition of $g_h$ is different for $Q$-type and $V$-type). This gives $\mathsf{o\text{-}dec}_\eta^{\Phi, \overline{D}_{\mathsf{av}}} \lesssim \alpha + \frac{B^2 H^3 d |\mathcal{A}| \eta}{\alpha} = O\left( \sqrt{B^2 H^3 d |\mathcal{A}| \eta} \right)$ by choosing the optimal $\alpha$.

Finally, using [Theorem 13](#) finishes the proof. $\qquad \square$

## H.5 RELATING dig-dec TO COVERABILITY UNDER BELLMAN COMPLETENESS

**Lemma 34.** *Let $(\mathcal{M}, \Phi)$ be Bellman complete ([Definition 9](#)), and suppose the coverability of every model in $\mathcal{M}$ is bounded by $d$. Then it holds that $\mathsf{o\text{-}dec}_\eta^{\Phi, \overline{D}_{\mathsf{sq}}} \leq \eta d H$ where $\overline{D}_{\mathsf{sq}}$ is defined with*

$$\xi_h(\phi', \phi; o_h) = (f_{\phi'}(s_h, a_h) - r_h - f_\phi(s_{h+1}))^2.$$

*Proof.* For $M = (P, R)$, define

$$g_h(s, a, \phi; M) = f_\phi(s, a) - R(s, a) - \mathbb{E}_{s' \sim P(\cdot | s, a)}[f_\phi(s')] = f_\phi(s, a) - f_{\mathcal{T}_M \phi}(s, a),$$

$$d_h^{\rho, M}(s, a) = \mathbb{E}_{\phi \sim \rho} \left[ d_h^{\pi_\phi, M}(s, a) \right].$$

By the AM-GM inequality, for any $\lambda > 0$,

$$\mathbb{E}_{\phi \sim \rho} \mathbb{E}^{\pi_\phi, M} [g_h(s_h, a_h, \phi; M)]$$

$$= \mathbb{E}_{\phi \sim \rho} \mathbb{E}_{(s,a) \sim d_h^{\pi_\phi, M}} [g_h(s, a, \phi; M)]$$

$$= \mathbb{E}_{\phi \sim \rho} \mathbb{E}_{(s,a) \sim d_h^{\rho, M}} \left[ \frac{d_h^{\pi_\phi, M}(s, a)}{d_h^{\rho, M}(s, a)} g_h(s, a, \phi; M) \right]$$

$$\leq \mathbb{E}_{\phi \sim \rho} \mathbb{E}_{(s,a) \sim d_h^{\rho, M}} \left[ \frac{\lambda}{4} \frac{d_h^{\pi_\phi, M}(s, a)^2}{d_h^{\rho, M}(s, a)^2} + \frac{1}{\lambda} g_h(s, a, \phi; M)^2 \right]$$

$$= \frac{\lambda}{4} \mathbb{E}_{\phi \sim \rho} \left[ \sum_{s,a} \frac{d_h^{\pi_\phi, M}(s, a)^2}{d_h^{\rho, M}(s, a)} \right] + \frac{1}{\lambda} \mathbb{E}_{\phi \sim \rho} \mathbb{E}_{\phi' \sim \rho} \mathbb{E}^{\pi_{\phi'}, M} [g_h(s_h, a_h, \phi, M)^2]. \qquad (27)$$

Note that

$$\sum_{h=1}^H \mathbb{E}_{\phi \sim \rho} \mathbb{E}^{\pi_\phi, M} [g_h(s_h, a_h, \phi; M)] = \mathbb{E}_{\phi \sim \rho} [V_\phi(\pi_\phi) - V_M(\pi_\phi)],$$

and by the same calculation as Eq. (23), we have

$$\frac{1}{B^2 H}\sum_{h=1}^{H}\mathbb{E}^{\pi_{\phi'},M}\left[g_h(s_h,a_h,\phi,M)^2\right] = \frac{1}{B^2 H}\sum_{h=1}^{H}\mathbb{E}^{\pi_{\phi'},M}\left[\xi_h(\phi',\phi;o_h) - \xi_h(\mathcal{T}_M\phi,\phi;o_h)\right] = \overline{D}_{\mathsf{sq}}^{\pi_{\phi'}}(\phi\|M).$$

By the definition of o-dec and combining the inequalities above,

$\mathsf{o\text{-}dec}_\eta^{\Phi,\overline{D}_{\mathsf{sq}}}$

$$= \max_\rho \min_p \max_\nu \mathbb{E}_{\pi\sim p}\mathbb{E}_{\phi\sim\rho}\mathbb{E}_{M\sim\nu}\left[V_\phi(\pi_\phi) - V_M(\pi) - \frac{1}{\eta}\overline{D}_{\mathsf{sq}}^\pi(\phi\|M)\right]$$

$$\leq \max_\rho \max_\nu \mathbb{E}_{\phi'\sim\rho}\mathbb{E}_{\phi\sim\rho}\mathbb{E}_{M\sim\nu}\left[V_\phi(\pi_\phi) - V_M(\pi_\phi) - \frac{1}{\eta}\overline{D}_{\mathsf{sq}}^{\pi_{\phi'}}(\phi\|M)\right]$$

$$= \max_\rho \max_\nu \mathbb{E}_{\phi'\sim\rho}\mathbb{E}_{\phi\sim\rho}\mathbb{E}_{M\sim\nu}\left[\sum_{h=1}^{H}\mathbb{E}^{\pi_\phi,M}[g_h(s_h,a_h,\phi;M)] - \frac{1}{\eta B^2 H}\sum_{h=1}^{H}\mathbb{E}^{\pi_{\phi'},M}\left[g_h(s_h,a_h,\phi,M)^2\right]\right]$$

$$\leq \frac{\eta B^2 H}{4}\max_\rho \max_\nu \mathbb{E}_{M\sim\nu}\mathbb{E}_{\phi\sim\rho}\left[\sum_{h=1}^{H}\sum_{s,a}\frac{d_h^{\pi_\phi,P}(s,a)^2}{d_h^{\rho,P}(s,a)}\right]. \qquad\qquad \text{(by Eq. (27))}$$

Let $\mu_h^P$ be any occupancy measure over layer $h$ that depends on $P$. Then

$$\mathbb{E}_{\phi\sim\rho}\left[\sum_{s,a}\frac{d_h^{\pi_\phi,P}(s,a)^2}{d_h^{\rho,P}(s,a)}\right] = \mathbb{E}_{\phi\sim\rho}\left[\sum_{s,a}\frac{d_h^{\pi_\phi,P}(s,a)\mu_h^P(s,a)}{d_h^{\rho,P}(s,a)}\cdot\frac{d_h^{\pi_\phi,P}(s,a)}{\mu_h^P(s,a)}\right]$$

$$\leq \mathbb{E}_{\phi\sim\rho}\left[\sum_{s,a}\frac{d_h^{\pi_\phi,P}(s,a)\mu_h^P(s,a)}{d_h^{\rho,P}(s,a)}\right]\cdot\max_{s,a,\pi}\frac{d_h^{\pi,P}(s,a)}{\mu_h^P(s,a)}$$

$$= \sum_{s,a}\mu_h^P(s,a)\cdot\max_{s,a,\pi}\frac{d_h^{\pi,P}(s,a)}{\mu_h^P(s,a)}$$

$$= \max_{s,a,\pi}\frac{d_h^{\pi,P}(s,a)}{\mu_h^P(s,a)}.$$

We let $\mu_h^P$ be the minimizer of $\max_{s,a,\pi}\frac{d_h^{\pi,P}(s,a)}{\mu_h^P(s,a)}$. The coverability in MDP $M$ is defined as $\min_\mu \max_{s,a,\pi,h}\frac{d_h^{\pi,P}(s,a)}{\mu_h^P(s,a)}$ [XFB+23]. Combining the inequalities proves $\mathsf{o\text{-}dec}_\eta^{\Phi,\overline{D}_{\mathsf{sq}}} \leq \eta d B^2 H^2$.

$\square$

# I   RELATING dig-dec TO EXISTING COMPLEXITIES IN THE HYBRID SETTING

## I.1   SUPPORTING LEMMAS

**Lemma 35.** *Let $g : \Phi \to [0, G]$. For $\nu, \rho \in \Delta(\Phi)$, we have*

$$\mathbb{E}_{\phi \sim \rho}[g(\phi)] \le 3\mathbb{E}_{\phi \sim \nu}[g(\phi)] + 2G \cdot D_{\mathrm{H}}^2(\nu, \rho),$$

*where $D_{\mathrm{H}}^2$ is the Hellinger distance.*

*Proof.*

$$
\begin{aligned}
|\mathbb{E}_{\phi \sim \rho}[g(\phi)] - \mathbb{E}_{\phi \sim \nu}[g(\phi)]| &= \left| \sum_{\phi} (\rho(\phi) - \nu(\phi)) g(\phi) \right| \\
&\le \sqrt{\sum_{\phi} (\rho(\phi) + \nu(\phi)) g(\phi)^2} \sqrt{\sum_{\phi} \frac{(\rho(\phi) - \nu(\phi))^2}{\rho(\phi) + \nu(\phi)}} \\
&\le \frac{1}{2} \mathbb{E}_{\phi \sim \rho}[g(\phi)] + \frac{1}{2} \mathbb{E}_{\phi \sim \nu}[g(\phi)] + \frac{G}{2} D_{\Delta}(\nu, \rho),
\end{aligned}
\tag{28}
$$

where

$$D_{\Delta}(\nu, \rho) = \sum_{\phi} \frac{(\rho(\phi) - \nu(\phi))^2}{\rho(\phi) + \nu(\phi)}$$

is the triangular discrimination. We can further bound it as

$$D_{\Delta}(\nu, \rho) = \sum_{\phi} \frac{(\rho(\phi) - \nu(\phi))^2}{\rho(\phi) + \nu(\phi)} = \sum_{\phi} \frac{(\sqrt{\rho(\phi)} - \sqrt{\nu(\phi)})^2 (\sqrt{\rho(\phi)} + \sqrt{\nu(\phi)})^2}{\rho(\phi) + \nu(\phi)} \le 2 D_{\mathrm{H}}^2(\nu, \rho).$$

Using this in Eq. (28) and rearranging gives the desired inequality. $\qquad\square$

**Lemma 36.** *Suppose that $(\mathcal{M}, \Phi)$ satisfy Assumption 5 with estimation function $\ell_h(\phi; o_h)_j = f_\phi(s_h, a_h; \boldsymbol{e}_j) - \varphi(s_h, a_h)^\top \boldsymbol{e}_j - f_\phi(s_{h+1}; \boldsymbol{e}_j)$. Furthermore, assume that $(\mathcal{M}, \Phi)$ is Bellman complete (Definition 10). Then Assumption 6 holds with $\xi_h(\phi', \phi; o_h) = \sum_{j=1}^d (f_{\phi'}(s_h, a_h; \boldsymbol{e}_j) - \varphi(s_h, a_h)^\top \boldsymbol{e}_j - f_\phi(s_{h+1}; \boldsymbol{e}_j))^2$ and*

$$\mathsf{dig\text{-}dec}_\eta^{\Phi, \overline{D}_{\mathsf{sq}}} \le \mathsf{dig\text{-}dec}_\eta^{\Phi, \overline{D}_{\mathsf{av}}}.$$

*Proof.* The proof is similar to that in the stochastic setting (Lemma 29). $\qquad\square$

**Lemma 37.** *Under Assumption 3 and Assumption 4, if $P, P' \in \phi$, then they share the same $d \times H$ dimensional vector:*

$$\left( \mathbb{E}^{\pi_\phi, P} [\varphi(s_h, a_h)] \right)_{h \in [H]} = \left( \mathbb{E}^{\pi_\phi, P'} [\varphi(s_h, a_h)] \right)_{h \in [H]}$$

*Proof.* Given a linear reward with known feature (Assumption 4), we have $R(s_h, a_h) = \varphi(s_h, a_h)^\top \theta_h(R)$ where $\varphi$ is a known feature. For any $P, R, \pi$, we have

$$V_{P,R}(\pi) = \sum_{h=1}^H \mathbb{E}^{\pi, P} \left[ \varphi(s_h, a_h)^\top \theta_h(R) \right].$$

Fix a $\phi$ and consider $P, P' \in \phi$. By Assumption 4, $V_{P,R}(\pi_\phi) = V_{P',R}(\pi_\phi)$ for any $R$. For each $h$, by instantiating $\theta_h(R)$ as all basis vectors in the $d$ dimensional space, we prove that $\mathbb{E}^{\pi_\phi, P}[\varphi(s_h, a_h)_j] = \mathbb{E}^{\pi_\phi, P'}[\varphi(s_h, a_h)_j]$ for any $h \in [H]$ and any $j \in [d]$. $\qquad\square$

**Definition 38.** *We define several quantities that will be reused in [Appendix I.2](#) for hybrid bilinear classes and [Appendix I.3](#) for coverable MDPs. We fix $\alpha \in [0, 1]$, and define $\pi^\alpha$ as the policy that in every step $h = 1, 2, \ldots, H$ chooses $\pi$ with probability $1 - \frac{\alpha}{H}$ and chooses $\mathrm{unif}(\mathcal{A})$ with probability $\frac{\alpha}{H}$. We also fix $\overline{D}$, which will be instantiated as $\overline{D}_{\mathsf{av}}$ and $\overline{D}_{\mathsf{sq}}$ in later subsections.*

*With them, we define (with $M = (P, R)$)*

$$\mathbf{TermA}_\eta^{\Phi,\overline{D}}(\nu) = \alpha + \mathbb{E}_{M\sim\nu}\mathbb{E}_{\phi\sim\nu}\mathbb{E}_{\phi'\sim\nu}\left[V_{\phi,R}(\pi_\phi) - V_M(\pi_\phi) - \frac{1}{9\eta}\overline{D}^{\pi_{\phi'}^\alpha}(\phi\|M)\right]$$

$$\mathbf{TermB}_\eta^{\Phi,\overline{D}}(\nu) = 6\sqrt{dH}\sqrt{3\mathbb{E}_{\phi\sim\nu}\mathbb{E}_{(P,R)\sim\nu}\left[(V_{\phi,R}(\pi_\phi) - V_{P,R}(\pi_\phi))^2\right]} - \frac{2}{9\eta}\mathbb{E}_{\phi'\sim\nu}\mathbb{E}_{M\sim\nu}\mathbb{E}_{\phi\sim\nu}\left[\overline{D}^{\pi_{\phi'}^\alpha}(\phi\|M)\right]$$

$$\mathbf{TermC}_\eta^{\Phi,\overline{D}}(\nu) = \mathbb{E}_{(M,\pi^\star)\sim\nu}\mathbb{E}_{\phi\sim\nu}\mathbb{E}_{\phi'\sim\nu}$$
$$\left[V_M(\pi^\star) - V_{\phi,R}(\pi_\phi) - \frac{1}{\eta}\mathbb{E}_{o\sim M(\cdot|\pi_{\phi'}^\alpha)}\left[KL(\nu_{\boldsymbol{\phi}}(\cdot|\pi_{\phi'}^\alpha, o), \nu_{\boldsymbol{\phi}})\right] - \frac{2}{9\eta}\overline{D}^{\pi_{\phi'}^\alpha}(\phi\|M)\right]$$

**Lemma 39.**

$$\min_p \max_\nu \mathsf{AIR}_\eta^{\Phi,D}(p, \nu; \rho) \leq \max_\nu \mathbf{TermA}_\eta^{\Phi,\overline{D}}(\nu) + \max_\nu \mathbf{TermC}_\eta^{\Phi,\overline{D}}(\nu).$$

*Proof.*

$\mathsf{AIR}_\eta^{\Phi,D}(p, \nu; \rho)$

$$= \mathbb{E}_{\pi\sim p}\mathbb{E}_{(M,\pi^\star)\sim\nu}\left[V_M(\pi^\star) - V_M(\pi) - \frac{1}{\eta}\mathbb{E}_{o\sim M(\cdot|\pi)}\left[\mathrm{KL}(\nu_{\boldsymbol{\phi}}(\cdot|\pi, o), \rho)\right] - \frac{1}{\eta}\mathbb{E}_{\phi\sim\rho}\left[\overline{D}^\pi(\phi\|M)\right]\right]$$

$$= \mathbb{E}_{\pi\sim p}\mathbb{E}_{(M,\pi^\star)\sim\nu}\left[V_M(\pi^\star) - V_M(\pi) - \frac{1}{\eta}\mathbb{E}_{o\sim M(\cdot|\pi)}\left[\mathrm{KL}(\nu_{\boldsymbol{\phi}}(\cdot|\pi, o), \nu_{\boldsymbol{\phi}})\right] - \frac{1}{\eta}\mathbb{E}_{\phi\sim\rho}\left[\overline{D}^\pi(\phi\|M)\right] - \frac{1}{\eta}\mathrm{KL}(\nu_{\boldsymbol{\phi}}, \rho)\right]$$

$$\leq \mathbb{E}_{\pi\sim p}\mathbb{E}_{(M,\pi^\star)\sim\nu}\left[V_M(\pi^\star) - V_M(\pi) - \frac{1}{\eta}\mathbb{E}_{o\sim M(\cdot|\pi)}\left[\mathrm{KL}(\nu_{\boldsymbol{\phi}}(\cdot|\pi, o), \nu_{\boldsymbol{\phi}})\right]\right.$$
$$\left. - \frac{1}{3\eta}\mathbb{E}_{\phi\sim\nu}\left[\overline{D}^\pi(\phi\|M)\right] + \frac{2}{3\eta}D_{\mathrm{H}}^2(\nu_{\boldsymbol{\phi}}, \rho) - \frac{1}{\eta}\mathrm{KL}(\nu_{\boldsymbol{\phi}}, \rho)\right] \qquad\qquad \text{([Lemma 35](#))}$$

$$\leq \mathbb{E}_{\pi\sim p}\mathbb{E}_{M\sim\nu}\mathbb{E}_{\phi\sim\nu}\left[V_{\phi,R}(\pi_\phi) - V_M(\pi) - \frac{1}{9\eta}\overline{D}^\pi(\phi\|M)\right]$$
$$+ \mathbb{E}_{\pi\sim p}\mathbb{E}_{(M,\pi^\star)\sim\nu}\mathbb{E}_{\phi\sim\nu}\left[V_M(\pi^\star) - V_{\phi,R}(\pi_\phi) - \frac{1}{\eta}\mathbb{E}_{o\sim M(\cdot|\pi)}\left[\mathrm{KL}(\nu_{\boldsymbol{\phi}}(\cdot|\pi, o), \nu_{\boldsymbol{\phi}})\right] - \frac{2}{9\eta}\overline{D}^\pi(\phi\|M)\right].$$

We have $\min_p \max_\nu \mathsf{AIR}_\eta^{\Phi,D}(p, \nu; \rho) = \max_\nu \min_p \mathsf{AIR}_\eta^{\Phi,D}(p, \nu; \rho)$ because $\mathsf{AIR}$ is convex in $p$ and concave in $\nu$. After the min-max swap, for each $\nu$, we choose $p$ to be such that $\pi \sim p$ is equivalent to first sampling $\phi' \sim \nu$ and then setting $\pi = \pi_{\phi'}^\alpha$. This gives

$$\min_p \max_\nu \mathsf{AIR}_\eta^{\Phi,D}(p, \nu; \rho)$$

$$\leq \max_\nu \mathbb{E}_{\phi'\sim\nu}\mathbb{E}_{M\sim\nu}\mathbb{E}_{\phi\sim\nu}\left[V_{\phi,R}(\pi_\phi) - V_M(\pi_{\phi'}^\alpha) - \frac{1}{9\eta}\overline{D}^{\pi_{\phi'}^\alpha}(\phi\|M)\right]$$
$$+ \mathbb{E}_{\phi'\sim\nu}\mathbb{E}_{(M,\pi^\star)\sim\nu}\mathbb{E}_{\phi\sim\nu}\left[V_M(\pi^\star) - V_{\phi,R}(\pi_\phi) - \frac{1}{\eta}\mathbb{E}_{o\sim M(\cdot|\pi_{\phi'}^\alpha)}\left[\mathrm{KL}(\nu_{\boldsymbol{\phi}}(\cdot|\pi_{\phi'}^\alpha, o), \nu_{\boldsymbol{\phi}})\right] - \frac{2}{9\eta}\overline{D}^{\pi_{\phi'}^\alpha}(\phi\|M)\right]$$

$$\leq \max_\nu \mathbf{TermA}_\eta^{\Phi,\overline{D}}(\nu) + \max_\nu \mathbf{TermC}_\eta^{\Phi,\overline{D}}(\nu).$$

$\square$

**Lemma 40.**

$$\mathbf{TermC}_\eta^{\Phi,\overline{D}}(\nu) \leq O(\eta dH + \alpha) + \mathbf{TermB}_\eta^{\Phi,\overline{D}}(\nu).$$

*Proof.* By [Lemma 37](#) we can define with any $P \in \phi$,

$$X_h(\phi) = \mathbb{E}^{\pi_\phi, P}\left[\varphi(s_h, a_h)\right].$$

Furthermore, define

$$X(\phi) = (X_h(\phi))_{h \in [H]} \in \mathbb{R}^{dH},$$
$$\theta(R) = (\theta_h(R))_{h \in [H]} \in \mathbb{R}^{dH}.$$

With this, we have

$$\mathbb{E}_{(M,\pi^\star) \sim \nu} \mathbb{E}_{\phi \sim \nu} [V_M(\pi^\star) - V_{\phi,R}(\pi_\phi)]$$
$$= \mathbb{E}_{\phi \sim \nu} \mathbb{E}_{R \sim \nu(\cdot|\phi)} [V_{\phi,R}(\pi_\phi)] - \mathbb{E}_{\phi \sim \nu} \mathbb{E}_{R \sim \nu} [V_{\phi,R}(\pi_\phi)]$$
$$= \mathbb{E}_{\phi \sim \nu} \left[ X(\phi)^\top \left( \mathbb{E}_{R \sim \nu(\cdot|\phi)} [\theta(R)] - \mathbb{E}_{R \sim \nu} [\theta(R)] \right) \right]$$
$$\leq \mathbb{E}_{\phi \sim \nu} \left[ \|X(\phi)\|_{\Sigma_\nu^{-1}} \left\| \mathbb{E}_{R \sim \nu(\cdot|\phi)} [\theta(R)] - \mathbb{E}_{R \sim \nu} [\theta(R)] \right\|_{\Sigma_\nu} \right] \quad (\Sigma_\nu = \mathbb{E}_{\phi \sim \nu} \left[ X(\phi) X(\phi)^\top \right])$$
$$\leq \sqrt{\mathbb{E}_{\phi \sim \nu} \left[ \|X(\phi)\|_{\Sigma_\nu^{-1}}^2 \right]} \sqrt{\mathbb{E}_{\phi \sim \nu} \left[ \left\| \mathbb{E}_{R \sim \nu(\cdot|\phi)} [\theta(R)] - \mathbb{E}_{R \sim \nu} [\theta(R)] \right\|_{\Sigma_\nu}^2 \right]}$$
$$= \sqrt{dH} \sqrt{\mathbb{E}_{\phi' \sim \nu} \mathbb{E}_{\phi \sim \nu} \left[ \left( X(\phi')^\top \mathbb{E}_{R \sim \nu(\cdot|\phi)} [\theta(R)] - X(\phi')^\top \mathbb{E}_{R \sim \nu} [\theta(R)] \right)^2 \right]}$$
$$\leq 3\sqrt{dH} \sqrt{\underbrace{\mathbb{E}_{\phi' \sim \nu} \mathbb{E}_{\phi \sim \nu} \left[ \left( \mathbb{E}_{(P,R) \sim \nu(\cdot|\phi)} [V_{P,R}(\pi_{\phi'})] - \mathbb{E}_{(P,R) \sim \nu} [V_{P,R}(\pi_{\phi'})] \right)^2 \right]}_{\textbf{Div1}}}$$
$$+ 3\sqrt{dH} \sqrt{\underbrace{\mathbb{E}_{\phi' \sim \nu} \mathbb{E}_{\phi \sim \nu} \left[ \left( X(\phi')^\top \mathbb{E}_{R \sim \nu(\cdot|\phi)} [\theta(R)] - \mathbb{E}_{(P,R) \sim \nu(\cdot|\phi)} [V_{P,R}(\pi_{\phi'})] \right)^2 \right]}_{\textbf{Div2}}}$$
$$+ 3\sqrt{dH} \sqrt{\underbrace{\mathbb{E}_{\phi' \sim \nu} \mathbb{E}_{\phi \sim \nu} \left[ \left( X(\phi')^\top \mathbb{E}_{R \sim \nu} [\theta(R)] - \mathbb{E}_{(P,R) \sim \nu} [V_{P,R}(\pi_{\phi'})] \right)^2 \right]}_{\textbf{Div3}}}. \tag{29}$$

For any observation $o = (s_1, a_1, r_1, \cdots, s_H, a_H, r_H)$, let $r(o) = \sum_{h=1}^H r_h$, we have

$$\textbf{Div1} = \mathbb{E}_{\phi' \sim \nu} \mathbb{E}_{\phi \sim \nu} \left[ \left( \mathbb{E}_{(P,R) \sim \nu(\cdot|\phi)} [V_{P,R}(\pi_{\phi'})] - \mathbb{E}_{(P,R) \sim \nu} [V_{P,R}(\pi_{\phi'})] \right)^2 \right]$$
$$\leq 2\mathbb{E}_{\phi' \sim \nu} \mathbb{E}_{\phi \sim \nu} \left[ \left( \mathbb{E}_{(P,R) \sim \nu(\cdot|\phi)} [V_{P,R}(\pi_{\phi'}^\alpha)] - \mathbb{E}_{(P,R) \sim \nu} [V_{P,R}(\pi_{\phi'}^\alpha)] \right)^2 \right] + 8\alpha^2$$
$$= 2\mathbb{E}_{\phi' \sim \nu} \mathbb{E}_{\phi \sim \nu} \left[ \left( \mathbb{E}_{(P,R) \sim \nu(\cdot|\phi)} \left[ \mathbb{E}_{o \sim M_{P,R}(\cdot|\pi_{\phi'}^\alpha)} [r(o)] \right] - \mathbb{E}_{(P,R) \sim \nu} \left[ \mathbb{E}_{o \sim M_{P,R}(\cdot|\pi_{\phi'}^\alpha)} [r(o)] \right] \right)^2 \right] + 8\alpha^2$$
$$= 2\mathbb{E}_{\phi' \sim \nu} \mathbb{E}_{\phi \sim \nu} \left[ \left( \mathbb{E}_{o \sim \nu(\cdot|\phi, \pi_{\phi'}^\alpha)} [r(o)] - \mathbb{E}_{o \sim \nu(\cdot|\pi_{\phi'}^\alpha)} [r(o)] \right)^2 \right] + 8\alpha^2$$
$$\leq 2\mathbb{E}_{\phi' \sim \nu} \mathbb{E}_{\phi \sim \nu} \left[ \left( \sum_o |\nu(o|\phi, \pi_{\phi'}^\alpha) - \nu(o|\pi_{\phi'}^\alpha)| \right)^2 \right] + 8\alpha^2$$
$$= 8\mathbb{E}_{\phi' \sim \nu} \mathbb{E}_{\phi \sim \nu} \left[ D_{\text{TV}}^2 \left( \nu_o(\cdot|\phi, \pi_{\phi'}^\alpha), \nu_o(\cdot|\pi_{\phi'}^\alpha) \right) \right] + 8\alpha^2$$
$$\leq 8\mathbb{E}_{\phi' \sim \nu} \mathbb{E}_{\phi \sim \nu} \left[ \text{KL} \left( \nu_o(\cdot|\phi, \pi_{\phi'}^\alpha), \nu_o(\cdot|\pi_{\phi'}^\alpha) \right) \right] + 8\alpha^2$$
$$= 8\mathbb{E}_{\phi' \sim \nu} \mathbb{E}_{M \sim \nu} \mathbb{E}_{o \sim M(\cdot|\pi_{\phi'}^\alpha)} \left[ \text{KL} \left( \nu_\phi(\cdot|\pi_{\phi'}^\alpha, o), \nu_\phi \right) \right] + 8\alpha^2.$$

On the other hand

$$\textbf{Div2} = \mathbb{E}_{\phi' \sim \nu} \mathbb{E}_{\phi \sim \nu} \left[ \left( X(\phi')^\top \mathbb{E}_{R \sim \nu(\cdot|\phi)} [\theta(R)] - \mathbb{E}_{(P,R) \sim \nu(\cdot|\phi)} [V_{P,R}(\pi_{\phi'})] \right)^2 \right]$$
$$= \mathbb{E}_{\phi' \sim \nu} \mathbb{E}_{\phi \sim \nu} \left[ \left( \mathbb{E}_{R \sim \nu(\cdot|\phi)} [V_{\phi',R}(\pi_{\phi'})] - \mathbb{E}_{(P,R) \sim \nu(\cdot|\phi)} [V_{P,R}(\pi_{\phi'})] \right)^2 \right]$$
$$\leq \mathbb{E}_{\phi' \sim \nu} \mathbb{E}_{\phi \sim \nu} \mathbb{E}_{(P,R) \sim \nu(\cdot|\phi)} \left[ (V_{\phi',R}(\pi_{\phi'}) - V_{P,R}(\pi_{\phi'}))^2 \right]$$
$$= \mathbb{E}_{\phi' \sim \nu} \mathbb{E}_{(P,R) \sim \nu} \left[ (V_{\phi',R}(\pi_{\phi'}) - V_{P,R}(\pi_{\phi'}))^2 \right]$$

Similarly,

$$
\begin{aligned}
\mathbf{Div3} &= \mathbb{E}_{\phi'\sim\nu}\mathbb{E}_{\phi\sim\nu}\left[\left(X(\phi')^\top\mathbb{E}_{R\sim\nu}\left[\theta(R)\right]-\mathbb{E}_{(P,R)\sim\nu}\left[V_{P,R}(\pi_{\phi'})\right]\right)^2\right]\\
&= \mathbb{E}_{\phi'\sim\nu}\mathbb{E}_{\phi\sim\nu}\left[\left(\mathbb{E}_{R\sim\nu}\left[V_{\phi',R}(\pi_{\phi'})\right]-\mathbb{E}_{(P,R)\sim\nu}\left[V_{P,R}(\pi_{\phi'})\right]\right)^2\right]\\
&\leq \mathbb{E}_{\phi'\sim\nu}\mathbb{E}_{(P,R)\sim\nu}\left[\left(V_{\phi',R}(\pi_{\phi'})-V_{P,R}(\pi_{\phi'})\right)^2\right]
\end{aligned}
$$

Combining these equations back to Eq. (29) and using the definition of $\mathbf{TermC}_\eta^{\Phi,\overline{D}}(\nu)$, we have

$$
\mathbf{TermC}_\eta^{\Phi,\overline{D}}(\nu)
$$
$$
\begin{aligned}
&\leq 3\sqrt{8dH\mathbb{E}_{\phi'\sim\nu}\mathbb{E}_{M\sim\nu}\mathbb{E}_{o\sim M(\cdot|\pi_{\phi'}^\alpha)}\left[\mathrm{KL}\left(\nu_{\boldsymbol{\phi}}(\cdot|\pi_{\phi'}^\alpha,o),\nu_{\boldsymbol{\phi}}\right)\right]+8\alpha^2}\\
&\quad+6\sqrt{dH}\sqrt{3\mathbb{E}_{\phi\sim\nu}\mathbb{E}_{(P,R)\sim\nu}\left[\left(V_{\phi,R}(\pi_\phi)-V_{P,R}(\pi_\phi)\right)^2\right]}\\
&\quad-\frac{1}{\eta}\mathbb{E}_{\phi'\sim\nu}\mathbb{E}_{M\sim\nu}\mathbb{E}_{o\sim M(\cdot|\pi_{\phi'}^\alpha)}\left[\mathrm{KL}\left(\nu_{\boldsymbol{\phi}}(\cdot|\pi_{\phi'}^\alpha,o),\nu_{\boldsymbol{\phi}}\right)\right]-\frac{2}{9\eta}\mathbb{E}_{\phi'\sim\nu}\mathbb{E}_{M\sim\nu}\mathbb{E}_{\phi\sim\nu}\left[\overline{D}^{\pi_{\phi'}^\alpha}(\phi\|M)\right]\\
&\leq O\left(\eta dH+\alpha\right)+6\sqrt{dH}\sqrt{3\mathbb{E}_{\phi\sim\nu}\mathbb{E}_{(P,R)\sim\nu}\left[\left(V_{\phi,R}(\pi_\phi)-V_{P,R}(\pi_\phi)\right)^2\right]}-\frac{2}{9\eta}\mathbb{E}_{\phi'\sim\nu}\mathbb{E}_{M\sim\nu}\mathbb{E}_{\phi\sim\nu}\left[\overline{D}^{\pi_{\phi'}^\alpha}(\phi\|M)\right]\\
&= O\left(\eta dH+\alpha\right)+\mathbf{TermB}_\eta^{\Phi,\overline{D}}(\nu).
\end{aligned}
$$

$\square$

## I.2   Relating dig-dec to hybrid bilinear rank

**Assumption 8** (Hybrid bilinear class [LWZ25]). *A model class $\mathcal{M}$ and its associated $\Phi$ satisfying Assumption 3 is a hybrid bilinear class with rank $d$ if there exists functions $X_h:\Phi\times\mathcal{P}\to\mathbb{R}^d$ and $W_h:\Phi\times\mathcal{R}\times\mathcal{P}\to\mathbb{R}^d$ for all $h\in[H]$ such that*

*1. For any $M=(P,R)\in\phi$, it holds that $W_h(\phi,\tilde{R};P)=0$ for any $\tilde{R}\in\mathcal{R}$.*
*2. For any $\phi\in\Phi$ and any $(P,R)\in\mathcal{M}$,*

$$
|V_{\phi,R}(\pi_\phi)-V_{P,R}(\pi_\phi)|\leq\sum_{h=1}^H|\langle X_h(\phi;P),W_h(\phi,R;P)\rangle|.
$$

*3. For every policy $\pi$, there exists an estimation policy $\pi^{\mathrm{est}}$. Also, there exists a discrepancy function $\ell_h:\Phi\times\mathcal{R}\times\mathcal{O}\to\mathbb{R}$ such that for any $\phi',\phi\in\Phi$ and any $M=(P,R)\in\mathcal{M}$,*

$$
|\langle X_h(\phi';P),W_h(\phi,R;P)\rangle|=\left|\mathbb{E}^{\pi_{\phi'}\circ_h\pi_{\phi'}^{\mathrm{est}},P}\left[\ell_h(\phi,R;o_h)\right]\right|
$$

*where $o_h=(s_h,a_h,r_h,s_{h+1})$ and $\pi\circ_h\pi^{\mathrm{est}}$ denotes a policy that plays $\pi$ for the first $h-1$ steps and plays policy $\pi^{\mathrm{est}}$ at the $h$-th step.*

*We call it an on-policy bilinear class if $\pi^{\mathrm{est}}=\pi$ for all $\pi\in\Pi$, and otherwise an off-policy bilinear class. We denote by $\pi^\alpha$ the policy that in every step $h=1,\ldots,H$ chooses $\pi$ with probability $1-\frac{\alpha}{H}$ and chooses $\pi^{\mathrm{est}}$ with probability $\frac{\alpha}{H}$.*

**Lemma 41.** *Hybrid bilinear classes (Assumption 8) with known-feature linear reward (Assumption 4) satisfy Assumption 5 with $N=d$.*

*Proof.* With the estimation function $\ell_h(\phi,R;o_h)$ defined in Assumption 8, we define for $j\in[d]$,

$$
\ell_h(\phi;o_h)_j=\ell_h(\phi,\boldsymbol{e}_j;o_h),
$$

where $\boldsymbol{e}_j$ as a reward represents the reward function defined as $R(s,a)=\varphi(s,a)^\top\boldsymbol{e}_j=\varphi(s,a)_j$.

For any $\phi' \in \Phi$ and any $M = (P, R) \in \phi$,

$$\left| \mathbb{E}^{\pi_{\phi'} \circ_h \pi_{\phi'}^{\text{est}}, P} [\ell_h(\phi; o_h)_j] \right|$$

$$= \left| \mathbb{E}^{\pi_{\phi'} \circ_h \pi_{\phi'}^{\text{est}}, P} [\ell_h(\phi, \boldsymbol{e}_j; o_h)] \right|$$

$$= |\langle X_h(\phi'; P), W_h(\phi, \boldsymbol{e}_j; P) \rangle| \qquad \text{(by Assumption 8.3)}$$

$$= 0. \qquad \text{(by Assumption 8.1)}$$

$\square$

**Lemma 42** (Lemma 20 of [LWZ25]). *Let* $(\mathcal{M}, \Phi)$ *be a hybrid bilinear class (Assumption 8). Then*

- $\max_\nu \mathbf{TermA}_\eta^{\Phi, \overline{D}_{\text{av}}}(\nu) \leq O(B^2 H^2 d\eta)$ *in the on-policy case.*

- $\max_\nu \mathbf{TermA}_\eta^{\Phi, \overline{D}_{\text{av}}}(\nu) \leq O(\alpha + B^2 H^3 d\eta/\alpha)$ *in the off-policy case.*[5]

**Lemma 43.** *Let* $(\mathcal{M}, \Phi)$ *be a hybrid bilinear class (Assumption 8). Then*

- $\max_\nu \mathbf{TermB}_\eta^{\Phi, \overline{D}_{\text{av}}}(\nu) \leq O\left( \left(B^2 H^5 d^3 \eta\right)^{\frac{1}{3}} \right)$ *in the on-policy case.*

- $\max_\nu \mathbf{TermB}_\eta^{\Phi, \overline{D}_{\text{av}}}(\nu) \leq O\left( \left(B^2 H^6 d^3 \eta/\alpha\right)^{\frac{1}{3}} \right)$ *in the off-policy case.*

*Proof.* From the definition of hybrid bilinear class in Assumption 8, we have

$$\mathbb{E}_{\phi \sim \nu} \mathbb{E}_{(P,R) \sim \nu} \left[ \left( V_{\phi, R}(\pi_\phi) - V_{P, R}(\pi_\phi) \right)^2 \right]$$

$$\leq \mathbb{E}_{\phi \sim \nu} \mathbb{E}_{(P,R) \sim \nu} \left[ \left( \sum_{h=1}^H |\langle X_h(\phi; P), W_h(\phi, R; P) \rangle| \right)^2 \right]$$

$$\leq H \sum_{h=1}^H \mathbb{E}_{\phi \sim \nu} \mathbb{E}_{(P,R) \sim \nu} \left[ |\langle X_h(\phi; P), W_h(\phi, R; P) \rangle|^2 \right].$$

Define $\Sigma_{h,P} = \mathbb{E}_{\phi \sim \nu} \left[ X_h(\phi; P) X_h(\phi; P)^\top \right]$. We have

$$\mathbb{E}_{\phi \sim \nu} \left[ |\langle X_h(\phi; P), W_h(\phi, R; P) \rangle|^2 \right]$$

$$\leq \mathbb{E}_{\phi \sim \nu} \left[ |\langle X_h(\phi; P), W_h(\phi, R; P) \rangle| \right]$$

$$\leq \sqrt{\mathbb{E}_{\phi \sim \nu} \left[ \|X_h(\phi; P)\|_{\Sigma_{h,P}^{-1}}^2 \right]} \sqrt{\mathbb{E}_{\phi \sim \nu} \left[ \|W_h(\phi, R; P)\|_{\Sigma_{h,P}}^2 \right]}$$

$$= \sqrt{d \mathbb{E}_{\phi \sim \nu} \mathbb{E}_{\phi' \sim \nu} \left[ \left( \mathbb{E}^{\pi_{\phi'} \circ_h \pi_{\phi'}^{\text{est}}, P} [\ell_h(\phi, R; o_h)] \right)^2 \right]}. \qquad \text{(Assumption 8)}$$

Thus,

$$\sqrt{\mathbb{E}_{\phi \sim \nu} \mathbb{E}_{(P,R) \sim \nu} \left[ \left( V_{\phi, R}(\pi_\phi) - V_{P, R}(\pi_\phi) \right)^2 \right]}$$

$$\leq \sqrt{H \sum_{h=1}^H \mathbb{E}_{(P,R) \sim \nu} \left[ \sqrt{d \mathbb{E}_{\phi \sim \nu} \mathbb{E}_{\phi' \sim \nu} \left[ \left( \mathbb{E}^{\pi_{\phi'} \circ_h \pi_{\phi'}^{\text{est}}, P} [\ell_h(\phi, R; o_h)] \right)^2 \right]} \right]}.$$

---

[5]As in Footnote 4, the bounds are different from [LWZ25]'s as we adopt a different scaling.

**(1)** In the on-policy case, we have $\alpha = 0$ and

$$6\sqrt{3dH\mathbb{E}_{\phi\sim\nu}\mathbb{E}_{(P,R)\sim\nu}\left[(V_{\phi,R}(\pi_\phi) - V_{P,R}(\pi_\phi))^2\right]} - \frac{2}{9\eta}\mathbb{E}_{\phi'\sim\nu}\mathbb{E}_{M\sim\nu}\mathbb{E}_{\phi\sim\nu}\left[D_{\mathsf{av}}^{\pi_{\phi'}}(\phi\|M)\right]$$

$$\leq 6\sqrt{3d^{\frac{3}{2}}H^2\sum_{h=1}^{H}\mathbb{E}_{(P,R)\sim\nu}\left[\sqrt{\mathbb{E}_{\phi\sim\nu}\mathbb{E}_{\phi'\sim\nu}\left[\left(\mathbb{E}^{\pi_{\phi'},P}\left[\ell_h(\phi,R;o_h)\right]\right)^2\right]}\right]}$$

$$- \frac{2}{9\eta B^2 H}\sum_{h=1}^{H}\mathbb{E}_{\phi\sim\nu}\mathbb{E}_{\phi'\sim\nu}\mathbb{E}_{(P,R)\sim\nu}\left[\sum_{j=1}^{d}\left(\mathbb{E}^{\pi_{\phi'},P}\left[\ell_h(\phi;o_h)_j\right]\right)^2\right]$$

$$\leq O\left(d^{\frac{3}{2}}H^2\beta\right) + \frac{1}{4\beta}\sum_{h=1}^{H}\mathbb{E}_{(P,R)\sim\nu}\left[\sqrt{\mathbb{E}_{\phi\sim\nu}\mathbb{E}_{\phi'\sim\nu}\left[\left(\mathbb{E}^{\pi_{\phi'},P}\left[\ell_h(\phi,R;o_h)\right]\right)^2\right]}\right]$$

$$- \frac{2}{9\eta B^2 H}\sum_{h=1}^{H}\mathbb{E}_{\phi\sim\nu}\mathbb{E}_{\phi'\sim\nu}\mathbb{E}_{(P,R)\sim\nu}\left[\left(\mathbb{E}^{\pi_{\phi'},P}\left[\ell_h(\phi,R;o_h)\right]\right)^2\right]$$

$$\leq O\left(d^{\frac{3}{2}}H^2\beta + \frac{\eta B^2 H}{\beta^2}\right) = O\left(\left(B^2 H^5 d^3 \eta\right)^{\frac{1}{3}}\right). \qquad\text{(choosing optimal }\beta)$$

**(2)** For the off-policy case, we have

$$6\sqrt{3dH\mathbb{E}_{\phi\sim\nu}\mathbb{E}_{(P,R)\sim\nu}\left[(V_{\phi,R}(\pi_\phi) - V_{P,R}(\pi_\phi))^2\right]} - \frac{2}{9\eta}\mathbb{E}_{\phi'\sim\nu}\mathbb{E}_{M\sim\nu}\mathbb{E}_{\phi\sim\nu}\left[D_{\mathsf{av}}^{\pi_{\phi'}^\alpha}(\phi\|M)\right]$$

$$\leq 6\sqrt{d^{\frac{3}{2}}H^2\sum_{h=1}^{H}\mathbb{E}_{(P,R)\sim\nu}\left[\sqrt{\mathbb{E}_{\phi\sim\nu}\mathbb{E}_{\phi'\sim\nu}\left[\left(\mathbb{E}^{\pi_{\phi'}\circ_h\pi_{\phi'}^{\mathrm{est}},P}\left[\ell_h(\phi,R;o_h)\right]\right)^2\right]}\right]}$$

$$- \frac{2}{9\eta B^2 H}\sum_{h=1}^{H}\mathbb{E}_{\phi\sim\nu}\mathbb{E}_{\phi'\sim\nu}\mathbb{E}_{(P,R)\sim\nu}\left[\sum_{j=1}^{d}\left(\mathbb{E}^{\pi_{\phi'}^\alpha,P}\left[\ell_h(\phi;o_h)_j\right]\right)^2\right]$$

$$\leq O\left(d^{\frac{3}{2}}H^2\beta\right) + \frac{1}{4\beta}\sum_{h=1}^{H}\mathbb{E}_{(P,R)\sim\nu}\left[\sqrt{\mathbb{E}_{\phi\sim\nu}\mathbb{E}_{\phi'\sim\nu}\left[\left(\mathbb{E}^{\pi_{\phi'}\circ_h\pi_{\phi'}^{\mathrm{est}},P}\left[\ell_h(\phi,R;o_h)\right]\right)^2\right]}\right]$$

$$- \frac{\alpha}{3H}\cdot\frac{2}{9\eta B^2 H}\sum_{h=1}^{H}\mathbb{E}_{\phi\sim\nu}\mathbb{E}_{\phi'\sim\nu}\mathbb{E}_{(P,R)\sim\nu}\left[\left(\mathbb{E}^{\pi_{\phi'}\circ_h\pi_{\phi'}^{\mathrm{est}},P}\left[\ell_h(\phi,R;o_h)\right]\right)^2\right]$$

$$\leq O\left(d^{\frac{3}{2}}H^2\beta + \frac{\eta B^2 H^2}{\alpha\beta^2}\right) = O\left(\left(B^2 H^6 d^3 \eta/\alpha\right)^{\frac{1}{3}}\right), \qquad\text{(with the optimal }\beta)$$

where the second-to-last inequality is because with probability $(1 - \frac{\alpha}{H})^{h-1}\frac{\alpha}{H} \geq \frac{\alpha}{3H}$, policy $\pi_{\phi'}^\alpha$ chooses the policy $\pi_{\phi'}\circ_h\pi_{\phi'}^{\mathrm{est}}$. $\qquad\square$

**Lemma 44.** *Let $(\mathcal{M},\Phi)$ be a hybrid bilinear class (Assumption 8). Then*

- $\mathsf{dig\text{-}dec}_\eta^{\Phi,\overline{D}_{\mathsf{av}}} \leq O\left(B^2 H^2 d\eta + \left(B^2 H^5 d^3 \eta\right)^{\frac{1}{3}}\right)$ *in the on-policy case;*
- $\mathsf{dig\text{-}dec}_\eta^{\Phi,\overline{D}_{\mathsf{av}}} \leq O\left(\sqrt{B^2 H^3 d\eta} + \left(B^2 H^6 d^3 \eta\right)^{\frac{1}{4}}\right)$ *in the off-policy case.*

*Proof.* This can be obtained by directly combining Lemma 39, Lemma 40, Lemma 42, Lemma 43. In the on-policy case,

$$\mathsf{dig\text{-}dec}_\eta^{\Phi,\overline{D}_{\mathsf{av}}} = O\left(B^2 H^2 d\eta + \left(B^2 H^5 d^3 \eta\right)^{\frac{1}{3}}\right).$$

In the off-policy case,

$$\mathsf{dig\text{-}dec}_\eta^{\Phi,\overline{D}_{\mathsf{av}}} = O\left(\alpha + B^2 H^3 d\eta/\alpha + \left(B^2 H^6 d^3 \eta/\alpha\right)^{\frac{1}{3}}\right)$$

$$= O\left(\sqrt{B^2 H^3 d\eta} + \left(B^2 H^6 d^3 \eta\right)^{\frac{1}{4}}\right). \qquad\text{(with optimal }\alpha)$$

$\square$

### I.3 RELATING dig-dec TO COVERABILITY UNDER BELLMAN COMPLETENESS

**Lemma 45.** *For hybrid MDPs with Bellman completeness and coverability bounded by $d$, it holds that*

$$\max_{\nu} \mathbf{TermA}_\eta^{\Phi, \overline{D}_{\mathsf{sq}}}(\nu) \leq O\left(\eta d B^2 H^2\right).$$

*Proof.* For $M = (P, R)$, define

$$g_h(s, a, \phi; R, P) = f_\phi(s, a; R) - R(s, a) - \mathbb{E}_{s' \sim P(\cdot | s, a)}[f_\phi(s'; R)],$$

$$d_h^{\nu, P}(s, a) = \mathbb{E}_{\phi \sim \nu}\left[d_h^{\pi_\phi, P}(s, a)\right].$$

By the AM-GM inequality, for any $\lambda > 0$,

$$
\begin{aligned}
&\mathbb{E}_{\phi \sim \nu} \mathbb{E}^{\pi_\phi, P}\left[g_h(s_h, a_h, \phi; R, P)\right] \\
&= \mathbb{E}_{\phi \sim \nu} \mathbb{E}_{(s, a) \sim d_h^{\pi_\phi, P}}\left[g_h(s, a, \phi; R, P)\right] \\
&= \mathbb{E}_{\phi \sim \nu} \mathbb{E}_{(s, a) \sim d_h^{\nu, P}}\left[\frac{d_h^{\pi_\phi, P}(s, a)}{d_h^{\nu, P}(s, a)} g_h(s, a, \phi; R, P)\right] \\
&\leq \mathbb{E}_{\phi \sim \nu} \mathbb{E}_{(s, a) \sim d_h^{\nu, P}}\left[\frac{\lambda}{4} \frac{d_h^{\pi_\phi, P}(s, a)^2}{d_h^{\nu, P}(s, a)^2} + \frac{1}{\lambda} g_h(s, a, \phi; R, P)^2\right] \\
&= \frac{\lambda}{4} \mathbb{E}_{\phi \sim \nu}\left[\sum_{s, a} \frac{d_h^{\pi_\phi, P}(s, a)^2}{d_h^{\nu, P}(s, a)}\right] + \frac{1}{\lambda} \mathbb{E}_{\phi \sim \nu} \mathbb{E}_{\phi' \sim \nu} \mathbb{E}^{\pi_{\phi'}, M}\left[g_h(s_h, a_h, \phi, R, P)^2\right]. \quad (30)
\end{aligned}
$$

Note that

$$\sum_{h=1}^{H} \mathbb{E}_{\phi \sim \nu} \mathbb{E}^{\pi_\phi, P}[g_h(s_h, a_h, \phi; R, P)] = \mathbb{E}_{\phi \sim \nu}\left[V_{\phi, R}(\pi_\phi) - V_M(\pi_\phi)\right],$$

and

$$
\begin{aligned}
&\sum_{h=1}^{H} \mathbb{E}^{\pi_{\phi'}, P}\left[g_h(s_h, a_h, \phi; R, P)^2\right] \\
&\leq \sum_{h=1}^{H} \sum_{j=1}^{d} \mathbb{E}^{\pi_{\phi'}, P}\left[g_h(s_h, a_h, \phi; \boldsymbol{e}_j, P)^2\right] \\
&= \sum_{h=1}^{H} \sum_{j=1}^{d} \mathbb{E}^{\pi_{\phi'}, P}\left[\left(f_\phi(s_h, a_h; \boldsymbol{e}_j) - \varphi(s_h, a_h)^\top \boldsymbol{e}_j - \mathbb{E}_{s' \sim P(\cdot | s, a)}[f_\phi(s'; \boldsymbol{e}_j)]\right)^2\right], \\
&= \sum_{h=1}^{H} \sum_{j=1}^{d} \mathbb{E}^{\pi_{\phi'}, P}\left[\left(f_\phi(s_h, a_h; \boldsymbol{e}_j) - f_{\mathcal{T}_M \phi}(s_h, a_h; \boldsymbol{e}_j)\right)^2\right] \\
&= \sum_{h=1}^{H} \mathbb{E}^{\pi_{\phi'}, P}\left[\|f_\phi(s_h, a_h) - f_{\mathcal{T}_M \phi}(s_h, a_h)\|^2\right] \\
&= \sum_{h=1}^{H} \mathbb{E}^{\pi_{\phi'}, P}\left[\xi_h(\phi, \phi; o_h) - \xi_h(\mathcal{T}_M \phi, \phi; o_h)\right] \quad \text{(by Eq. (23))} \\
&= B^2 H \overline{D}_{\mathsf{sq}}^{\pi_{\phi'}}(\phi \| M). \quad (31)
\end{aligned}
$$

Thus,

$\mathbf{TermA}_\eta^{\Phi, \overline{D}_{\mathsf{sq}}}(\nu)$

$= \mathbb{E}_{M \sim \nu} \mathbb{E}_{\phi' \sim \nu} \mathbb{E}_{\phi \sim \nu} \left[ V_{\phi, R}(\pi_\phi) - V_M(\pi_\phi) - \frac{1}{\eta} \overline{D}_{\mathsf{sq}}^{\pi_{\phi'}} (\phi \| M) \right]$

$\leq \mathbb{E}_{\phi \sim \nu} \mathbb{E}_{\phi' \sim \nu} \mathbb{E}_{M \sim \nu} \left[ \sum_{h=1}^H \mathbb{E}^{\pi_\phi, P}[g_h(s_h, a_h, \phi; R, P)] - \frac{1}{\eta B^2 H} \sum_{h=1}^H \mathbb{E}^{\pi_{\phi'}, P} \left[ g_h(s_h, a_h, \phi, R, P)^2 \right] \right]$

$\leq \frac{\eta B^2 H}{4} \mathbb{E}_{M \sim \nu} \mathbb{E}_{\phi \sim \nu} \left[ \sum_{h=1}^H \sum_{s,a} \frac{d_h^{\pi_\phi, P}(s,a)^2}{d_h^{\nu, P}(s,a)} \right].$  (by Eq. (30))

Let $\mu_h^P$ be any occupancy measure over layer $h$ that depends on $P$. Then

$$\mathbb{E}_{\phi \sim \nu} \left[ \sum_{s,a} \frac{d_h^{\pi_\phi, P}(s,a)^2}{d_h^{\nu, P}(s,a)} \right] = \mathbb{E}_{\phi \sim \nu} \left[ \sum_{s,a} \frac{d_h^{\pi_\phi, P}(s,a) \mu_h^P(s,a)}{d_h^{\nu, P}(s,a)} \cdot \frac{d^{\pi_\phi, P}(s,a)}{\mu_h^P(s,a)} \right]$$

$$\leq \mathbb{E}_{\phi \sim \nu} \left[ \sum_{s,a} \frac{d_h^{\pi_\phi, P}(s,a) \mu_h^P(s,a)}{d_h^{\nu, P}(s,a)} \right] \cdot \max_{s,a,\pi} \frac{d_h^{\pi, P}(s,a)}{\mu_h^P(s,a)}$$

$$= \left( \sum_{s,a} \mu_h^P(s,a) \right) \cdot \max_{s,a,\pi} \frac{d_h^{\pi, P}(s,a)}{\mu_h^P(s,a)}$$

$$= \max_{s,a,\pi} \frac{d_h^{\pi, P}(s,a)}{\mu_h^P(s,a)}.$$  (32)

We let $\mu_h^P$ be the minimizer of $\max_{s,a,\pi} \frac{d_h^{\pi, P}(s,a)}{\mu_h^P(s,a)}$. The coverability in MDP $M$ is defined as $\min_\mu \max_{s,a,\pi,h} \frac{d_h^{\pi, P}(s,a)}{\mu_h^P(s,a)}$ [XFB+23]. Combining the inequalities proves $\mathbf{TermA}_\eta^{\Phi, \overline{D}_{\mathsf{sq}}}(\nu) \leq O\left( \eta d B^2 H^2 \right)$. □

**Lemma 46.** *For hybrid MDPs with Bellman completeness and coverability bounded by d, it holds that*

$$\max_\nu \mathbf{TermB}_\eta^{\Phi, \overline{D}_{\mathsf{sq}}}(\nu) \leq O\left( \left( B^2 H^5 d^3 \eta \right)^{\frac{1}{3}} \right).$$

*Proof.* By definition,

$\mathbf{TermB}_\eta^{\Phi, \overline{D}_{\mathsf{sq}}}(\nu) = 6 \sqrt{dH} \sqrt{3 \mathbb{E}_{\phi \sim \nu} \mathbb{E}_{(P,R) \sim \nu} \left[ (V_{\phi, R}(\pi_\phi) - V_{P,R}(\pi_\phi))^2 \right]} - \frac{2}{9\eta} \mathbb{E}_{\phi' \sim \nu} \mathbb{E}_{M \sim \nu} \mathbb{E}_{\phi \sim \nu} \left[ \overline{D}_{\mathsf{sq}}^{\pi_{\phi'}} (\phi \| M) \right]$

Define

$$g_h(s, a, \phi; R, P) = f_\phi(s, a; R) - R(s, a) - \mathbb{E}_{s' \sim P(\cdot | s, a)}[f_\phi(s'; R)],$$

$$d_h^{\nu, P}(s, a) = \mathbb{E}_{\phi \sim \nu} \left[ d_h^{\pi_\phi, P}(s, a) \right].$$

We have

$$\mathbb{E}_{\phi \sim \nu} \left[ \left( V_{\phi,R}(\pi_\phi) - V_{P,R}(\pi_\phi) \right)^2 \right]$$

$$= H \sum_{h=1}^{H} \mathbb{E}_{\phi \sim \nu} \mathbb{E}_{(s,a) \sim d_h^{\pi_\phi, P}} \left[ g_h(s, a, \phi; R, P)^2 \right]$$

$$\leq H \sum_{h=1}^{H} \mathbb{E}_{\phi \sim \nu} \mathbb{E}_{(s,a) \sim d_h^{\pi_\phi, P}} \left[ |g_h(s, a, \phi; R, P)| \right]$$

$$= H \sum_{h=1}^{H} \mathbb{E}_{\phi \sim \nu} \mathbb{E}_{(s,a) \sim d_h^{\nu, P}} \left[ \frac{d_h^{\pi_\phi, P}(s,a)}{d_h^{\nu, P}(s,a)} |g_h(s, a, \phi; R, P)| \right]$$

$$\leq H \sum_{h=1}^{H} \sqrt{\mathbb{E}_{\phi \sim \nu} \mathbb{E}_{(s,a) \sim d_h^{\nu, P}} \left[ \frac{d_h^{\pi_\phi, P}(s,a)^2}{d_h^{\nu, P}(s,a)^2} \right]} \sqrt{\mathbb{E}_{\phi \sim \nu} \mathbb{E}_{(s,a) \sim d_h^{\nu, P}} \left[ \left( g_h(s, a, \phi; R, P) \right)^2 \right]}$$

$$\leq H \sum_{h=1}^{H} \sqrt{d \mathbb{E}_{\phi \sim \nu} \mathbb{E}_{(s,a) \sim d_h^{\nu, P}} \left[ g_h(s, a, \phi; R, P)^2 \right]}. \qquad \text{(by Eq. (32) and that coverability } \leq d)$$

Thus,

$$6\sqrt{dH} \sqrt{3 \mathbb{E}_{\phi \sim \nu} \mathbb{E}_{(P,R) \sim \nu} \left[ \left( V_{\phi,R}(\pi_\phi) - V_{P,R}(\pi_\phi) \right)^2 \right]} - \frac{2}{9\eta} \mathbb{E}_{\phi' \sim \nu} \mathbb{E}_{M \sim \nu} \mathbb{E}_{\phi \sim \nu} \left[ \overline{D}_{\mathsf{sq}}^{\pi_{\phi'}} (\phi \| M) \right]$$

$$\leq \sqrt{d^{\frac{3}{2}} H^2 \sum_{h=1}^{H} \mathbb{E}_{(P,R) \sim \nu} \left[ \sqrt{\mathbb{E}_{\phi \sim \nu} \mathbb{E}_{(s,a) \sim d_h^{\nu, P}} \left[ g_h(s, a, \phi; R, P)^2 \right]} \right]} - \frac{2}{9\eta} \mathbb{E}_{\phi' \sim \nu} \mathbb{E}_{M \sim \nu} \mathbb{E}_{\phi \sim \nu} \left[ \overline{D}_{\mathsf{sq}}^{\pi_{\phi'}} (\phi \| M) \right]$$

$$\leq d^{\frac{3}{2}} H^2 \beta + \frac{1}{4\beta} \sum_{h=1}^{H} \mathbb{E}_{(P,R) \sim \nu} \left[ \sqrt{\mathbb{E}_{\phi \sim \nu} \mathbb{E}_{(s,a) \sim d_h^{\nu, P}} \left[ g_h(s, a, \phi; R, P)^2 \right]} \right]$$

$$\qquad - \frac{2}{9\eta B^2 H} \sum_{h=1}^{H} \mathbb{E}_{\phi' \sim \nu} \mathbb{E}_{M \sim \nu} \mathbb{E}_{\phi \sim \nu} \mathbb{E}_{(s,a) \sim d_h^{\pi_{\phi'}, P}} \left[ g_h(s, a, \phi; R, P)^2 \right] \qquad \text{(Eq. (31))}$$

$$\leq O \left( d^{\frac{3}{2}} H^2 \beta + \frac{\eta B^2 H}{\beta^2} \right) = O \left( \left( B^2 H^5 d^3 \eta \right)^{\frac{1}{3}} \right).$$

$$\square$$

**Lemma 47.** *For hybrid MDPs with Bellman completeness and coverability bounded by d, it holds that*

$$\mathsf{dig\text{-}dec}_\eta^{\Phi, \overline{D}_{\mathsf{av}}} = O \left( B^2 H^2 d\eta + \left( B^2 H^5 d^3 \eta \right)^{\frac{1}{3}} \right).$$

*Proof.* This can be obtained by directly combining Lemma 39, Lemma 40, Lemma 45, Lemma 46.

$$\square$$

# J OMITTED DETAILS IN SECTION 6

## J.1 PROOF OF THEOREM 14

In this section, we will use $\text{Ber}(p)$ to denote Bernoulli distribution with success probability $p$. We consider parameters $\epsilon$ and $\Delta$ with $\epsilon < \Delta = \frac{1}{16\sqrt{T}} \le \frac{1}{16}$. Define $p^+ = \frac{1}{2} + \Delta$ and $p^- = \frac{1}{2} - \Delta$. Let $\mathbb{H}(\nu)$ denote the entropy of distribution $\nu$. We assume learning rate $\eta \le 1$.

Consider a three-arm bandit environment with model class $\mathcal{M} = \{M_1, M_2\}$ where

- $M_1 = (\text{Ber}(p^-), \text{Ber}(p^+), \epsilon\text{Ber}(0.5))$. The reward distribution is $\text{Ber}(p^-)$ for arm $a_1$ and $\text{Ber}(p^+)$ for arm $a_2$. Arm $a_3$'s reward is $0$ and $\epsilon$ with equal probability.
- $M_2 = (\text{Ber}(p^+), \text{Ber}(p^-), 0.5\epsilon)$. The reward distribution is $\text{Ber}(p^+)$ for arm $a_1$ and $\text{Ber}(p^-)$ for arm $a_2$. Arm $a_3$'s reward is $0.5\epsilon$ deterministically.

In this setting, $\Phi$ contains two infosets (based on Assumption 1):
$$\phi_1 = \{(M_1, \pi_{M_1})\}, \quad \phi_2 = \{(M_2, \pi_{M_2})\}.$$
In the rest of this proof, we compare the optimistic E2D algorithm [FGQ+23] and our algorithm in this environment.

**Optimistic DEC algorithm [FGQ+23]** Given $\rho_t \in \Delta(\Phi)$, the algorithm chooses action distribution via
$$p_t = \underset{p \in \Delta(\Pi)}{\text{argmin}} \max_{\nu \in \Delta(\Psi)} \mathbb{E}_{a \sim p} \mathbb{E}_{\phi \sim \rho_t} \mathbb{E}_{M \sim \nu} \left\{ V_\phi(a_\phi) - V_M(a) - \frac{1}{\eta} D^a(\phi \| M) \right\} \tag{33}$$
where $a_\phi$ is the optimal action of infoset $\phi$. In this simple bandit setting, the bilinear divergence and the squared Bellman error coincide with
$$D^a(\phi \| M) = \left( \mathbb{E}^{a,M}[V_\phi(a) - r] \right)^2 = (V_\phi(a) - V_M(a))^2.$$
We first consider the divergence term, for action $a \in \{a_1, a_2\}$, we have
$$\mathbb{E}_{\phi \sim \rho_t} \mathbb{E}_{M \sim \nu} [D^a(\phi \| M)] = \rho_t(\phi_1)\nu(M_2)(V_{\phi_1}(a) - V_{M_2}(a))^2 + \rho_t(\phi_2)\nu(M_1)(V_{\phi_2}(a) - V_{M_1}(a))^2$$
$$= 4\left( \rho_t(\phi_1)\nu(M_2) + \rho_t(\phi_2)\nu(M_1) \right) \Delta^2 \tag{34}$$
For action $a = a_3$, we have
$$\mathbb{E}_{\phi \sim \rho_t} \mathbb{E}_{M \sim \nu} [D^a(\phi \| M)] = \rho_t(\phi_1)\nu(M_2)(V_{\phi_1}(a) - V_{M_2}(a))^2 + \rho_t(\phi_2)\nu(M_1)(V_{\phi_2}(a) - V_{M_1}(a))^2$$
$$= 0 \tag{35}$$
Thus, for any $\rho_t$ and $\nu$, we have
$$\mathbb{E}_{a \sim p} \mathbb{E}_{\phi \sim \rho_t} \mathbb{E}_{M \sim \nu} \left[ -\frac{1}{\eta} D^a(\phi \| M) \right] = -\frac{4(1 - p(a_3))\Delta^2}{\eta} \left( \rho_t(\phi_1)\nu(M_2) + \rho_t(\phi_2)\nu(M_1) \right)$$
which is monotonically increasing in $p(a_3)$.

We then consider the regret term. For any $p \in \Delta(\Pi)$, define $\tilde{p} = \left( \frac{p(a_1)}{1 - p(a_3)}, \frac{p(a_2)}{1 - p(a_3)}, 0 \right)$ if $p(a_3) < 1$, and $\tilde{p} = (\frac{1}{2}, \frac{1}{2}, 0)$ otherwise. For any $M \in \mathcal{M}$, when $p(a_3) < 1$ we have
$$\mathbb{E}_{a \sim p}[V_M(a)] - \mathbb{E}_{a \sim \tilde{p}}[V_M(a)] = \sum_{a \in \{a_1, a_2\}} (p(a) - \tilde{p}(a)) V_M(a) + p(a_3)V_M(a_3)$$
$$= \frac{-p(a_3)}{1 - p(a_3)} \sum_{a \in \{a_1, a_2\}} p(a)V_M(a) + p(a_3)V_M(a_3)$$
$$\le \frac{-p(a_3)}{1 - p(a_3)} (p(a_1) + p(a_2)) p^- + p(a_3)V_M(a_3)$$
$$(V_M(a) \ge p^- \text{ for any } M \text{ and } a \in \{a_1, a_2\}, \text{ and } p(a_3) < 1)$$
$$= p(a_3) \left( V_M(a_3) - \frac{1}{2} + \Delta \right)$$
$$\le p(a_3) \left( 0.5\epsilon + \Delta - \frac{1}{2} \right) \le 0, \qquad (\epsilon < \Delta \le \frac{1}{16})$$

and when $p(a_3) = 1$ we also have $\mathbb{E}_{a \sim p}[V_M(a)] - \mathbb{E}_{a \sim \tilde{p}}[V_M(a)] \leq 0$. Thus, for any $\rho_t$, $\nu$, and $p$,
$$\mathbb{E}_{a \sim \tilde{p}} \mathbb{E}_{\phi \sim \rho_t} \mathbb{E}_{M \sim \nu} \{V_\phi(a_\phi) - V_M(a)\} \leq \mathbb{E}_{a \sim p} \mathbb{E}_{\phi \sim \rho_t} \mathbb{E}_{M \sim \nu} \{V_\phi(a_\phi) - V_M(a)\}.$$
Combining the discussion of the above two terms, for any $\rho_t, \nu$ and $p$, we have

$$\mathbb{E}_{a \sim \tilde{p}} \mathbb{E}_{\phi \sim \rho_t} \mathbb{E}_{M \sim \nu} \left\{ V_\phi(a_\phi) - V_M(a) - \frac{1}{\eta} D^a(\phi \| M) \right\} \leq \mathbb{E}_{a \sim p} \mathbb{E}_{\phi \sim \rho_t} \mathbb{E}_{M \sim \nu} \left\{ V_\phi(a_\phi) - V_M(a) - \frac{1}{\eta} D^a(\phi \| M) \right\}.$$
(36)

Given Eq. (36), the minimax solution of Eq. (33) must have $p_t(3) = 0$ for any $\rho_t$ and any $t$. This implies that the optimistic DEC algorithm will never choose $a_3$ and the problem degenerate to standard two-arm bandit, so the policy derived from optimistic DEC objective Eq. (33) must suffer standard regret lower bound $\mathbb{E}[\text{Reg}(\pi_{M^\star})] \geq \Omega(\sqrt{T})$ given $\Delta = \Theta\left(\frac{1}{\sqrt{T}}\right)$.

**Our algorithm**  Given $\rho_1$ is a uniform distribution, we consider our first step optimization where

$$p_1 = \operatorname*{argmin}_{p \in \Delta(\Pi)} \max_{\nu \in \Delta(\Psi)} \mathbb{E}_{a \sim p} \mathbb{E}_{\phi \sim \rho_1} \mathbb{E}_{M \sim \nu} \left\{ V_M(a_M) - V_M(a) - \frac{1}{\eta} \mathbb{E}_{o \sim M(\cdot|a)} [\text{KL}(\nu_\phi(\cdot|a, o), \rho_1)] - \frac{1}{\eta} D^a(\phi \| M) \right\}.$$
(37)

Below, we discuss the four terms in Eq. (37).

The $V_M(a_M)$ term  For any $\nu$, we have $\mathbb{E}_{M \sim \nu}[V_M(a_M)] = p^+$, which is a constant. Therefore, this term can be ignored in the objective.

The $V_M(a)$ term  By direct calculation, we have

$$\mathbb{E}_{a \sim p} \mathbb{E}_{M \sim \nu}[V_M(a)] = \frac{p(a_1) + p(a_2)}{2} + (p(a_1) - p(a_2))(\nu(M_2) - \nu(M_1))\Delta + 0.5 p(a_3)\epsilon.$$
(38)

For any $p = (p(a_1), p(a_2), p(a_3))$, consider $\hat{p} = (\frac{p(a_1) + p(a_2)}{2}, \frac{p(a_1) + p(a_2)}{2}, p(a_3))$. By Eq. (38) we have

$$\max_{\nu \in \Delta(\Psi)} \mathbb{E}_{a \sim \hat{p}} \mathbb{E}_{M \sim \nu}[-V_M(a)] \leq \max_{\nu \in \Delta(\Psi)} \mathbb{E}_{a \sim p} \mathbb{E}_{M \sim \nu}[-V_M(a)].$$
(39)

The $D^a(\phi \| M)$ term  Given $\rho_1$ is a uniform distribution, for action $a \in \{1, 2\}$, from Eq. (34), for any $\nu$ we have $\mathbb{E}_{\phi \sim \rho_1} \mathbb{E}_{M \sim \nu}[D^a(\phi \| M)] = 2\Delta^2$. For action $a = 3$, from Eq. (35), for any $\nu$, we have $\mathbb{E}_{\phi \sim \rho_1} \mathbb{E}_{M \sim \nu}[D^a(\phi \| M)] = 0$. Hence, $\mathbb{E}_{a \sim p} \mathbb{E}_{\phi \sim \rho_1} \mathbb{E}_{M \sim \nu}[D^a(\phi \| M)] = 2(1 - p(a_3))\Delta^2$. Note that now this is independent of $\nu$, and only related to $p(a_3)$ or $p(a_1) + p(a_2)$ but not $p(a_1)$ or $p(a_2)$ individually.

The KL term  Notice that
$$\nu_o(\cdot|a_1, \phi_1) = \text{Ber}\left(p^-\right), \quad \nu_o(\cdot|a_2, \phi_1) = \text{Ber}\left(p^+\right), \quad \nu_o(\cdot|a_1, \phi_2) = \text{Ber}\left(p^+\right), \quad \nu_o(\cdot|a_2, \phi_2) = \text{Ber}\left(p^-\right),$$
$$\nu_o(\cdot|a_1) = \text{Ber}(m_1), \qquad \nu_o(\cdot|a_2) = \text{Ber}(m_2),$$
where $m_1 = \nu(\phi_1)p^- + \nu(\phi_2)p^+$ and $m_2 = \nu(\phi_1)p^+ + \nu(\phi_2)p^-$ and it holds that $m_1 + m_2 = 1$. Given that $\text{KL}(\text{Ber}(p), \text{Ber}(q)) = \text{KL}(\text{Ber}(1-p), \text{Ber}(1-q))$, we have

$$\mathbb{E}_{a \sim p} \mathbb{E}_{M \sim \nu} \left[ \mathbb{E}_{o \sim M(a)}[\text{KL}(\nu_\phi(\cdot|a, o), \rho_1)] \right]$$
$$= \mathbb{E}_{a \sim p} \mathbb{E}_{\phi \sim \nu} \left[ \text{KL}(\nu_o(\cdot|a, \phi), \nu_o(\cdot|a)) \right] + \text{KL}(\nu_\phi, \rho_1)$$
$$= p(a_1)\nu(\phi_1)\text{KL}\left(\text{Ber}\left(p^-\right), \text{Ber}(m_1)\right) + p(a_2)\nu(\phi_1)\text{KL}\left(\text{Ber}\left(p^+\right), \text{Ber}(m_2)\right) + \text{KL}(\nu_\phi, \rho_1)$$
$$\quad + p(a_1)\nu(\phi_2)\text{KL}\left(\text{Ber}\left(p^+\right), \text{Ber}(m_1)\right) + p(a_2)\nu(\phi_2)\text{KL}\left(\text{Ber}\left(p^-\right), \text{Ber}(m_2)\right)$$
$$\quad + p(a_3)\mathbb{E}_{\phi \sim \nu}[\text{KL}(\nu_o(\cdot|a_3, \phi), \nu_o(\cdot|a_3))]$$
$$= (p(a_1) + p(a_2))\left(\nu(\phi_1)\text{KL}\left(\text{Ber}\left(p^-\right), \text{Ber}(m_1)\right) + \nu(\phi_2)\text{KL}\left(\text{Ber}\left(p^+\right), \text{Ber}(m_1)\right)\right)$$
$$\quad + p(a_3)\mathbb{H}(\nu) + \text{KL}(\nu_\phi, \rho_1)$$
$$= (1 - p(a_3))\left(\mathbb{H}(\text{Ber}(m_1)) - \mathbb{H}\left(\text{Ber}\left(p^+\right)\right)\right) + p(a_3)\mathbb{H}(\nu) + \text{KL}(\nu_\phi, \rho_1).$$

Note that this term is only related to $p(a_3)$ or $p(a_1) + p(a_2)$, but not $p(a_1)$ or $p(a_2)$ individually.

**Combining terms** Combining the case discussions above, for any $p = (p(a_1), p(a_2), p(a_3))$, with $\hat{p} = (\frac{p(a_1)+p(a_2)}{2}, \frac{p(a_1)+p(a_2)}{2}, p(a_3))$, we have

$$
\max_{\nu \in \Delta(\Psi)} \left\{ \mathbb{E}_{a \sim \hat{p}} \mathbb{E}_{M \sim \nu} \left[ -V_M(a) - \frac{1}{\eta} \mathbb{E}_{o \sim M(a)} \left[ \mathrm{KL}(\nu_{\boldsymbol{\phi}}(\cdot|a, o), \rho_1) \right] - \frac{1}{\eta} \mathbb{E}_{\phi \sim \rho_1} [D^a(\phi \| M)] \right] \right\}
$$
$$
\leq \max_{\nu \in \Delta(\Psi)} \left\{ \mathbb{E}_{a \sim p} \mathbb{E}_{M \sim \nu} \left[ -V_M(a) - \frac{1}{\eta} \mathbb{E}_{o \sim M(a)} \left[ \mathrm{KL}(\nu_{\boldsymbol{\phi}}(\cdot|a, o), \rho_1) \right] - \frac{1}{\eta} \mathbb{E}_{\phi \sim \rho_1} [D^a(\phi \| M)] \right] \right\}.
$$

To calculate the max value of the left-hand-side, consider policy distribution $p_s = (\frac{1-s}{2}, \frac{1-s}{2}, s)$. We have

$$
\mathbb{E}_{a \sim p_s} \mathbb{E}_{M \sim \nu} \left[ -V_M(a) - \frac{1}{\eta} \mathbb{E}_{o \sim M(a)} \left[ \mathrm{KL}(\nu_{\boldsymbol{\phi}}(\cdot|a, o), \rho_1) \right] - \frac{1}{\eta} \mathbb{E}_{\phi \sim \rho_1} [D^a(\phi \| M)] \right]
$$
$$
= \frac{s-1}{2} - \frac{s\epsilon}{2} - \frac{1}{\eta} \left( (1-s) \left( \mathbb{H}\left( \mathrm{Ber}(m_1) \right) - \mathbb{H}\left( \mathrm{Ber}(p^+) \right) + 2\Delta^2 \right) + \mathrm{KL}\left( \nu_{\boldsymbol{\phi}}, \rho_1 \right) + s\mathbb{H}(\nu) \right)
\tag{40}
$$

where $m_1 = \nu(\phi_1) p^- + \nu(\phi_2) p^+$. Define

$$
G(\nu) = (1-s)\mathbb{H}(\mathrm{Ber}(m_1)) + \mathrm{KL}\left( \nu_{\boldsymbol{\phi}}, \rho_1 \right) + s\mathbb{H}(\nu).
$$

To calculate $\max_\nu$ of Eq. (40), we only need to consider $\min_\nu \{G(\nu)\}$. By setting $\nu(\phi_2) = 1 - \nu(\phi_1)$, function $G$ is only related to $\nu(\phi_1)$ and we denote it as $G(\nu(\phi_1))$, after taking derivative, we have

$$
G'(\nu(\phi_1)) = (1-s) \ln\left( \frac{1-m_1}{m_1} \right) (p^- - p^+) + \log\left( \frac{\nu(\phi_1)}{1-\nu(\phi_1)} \right) + s \log\left( \frac{1-\nu(\phi_1)}{\nu(\phi_1)} \right)
$$
$$
= -\Delta(1-s) \ln\left( \frac{1-m_1}{m_1} \right) + \log\left( \frac{\nu(\phi_1)}{1-\nu(\phi_1)} \right) + s \log\left( \frac{1-\nu(\phi_1)}{\nu(\phi_1)} \right)
$$

where $m_1 = \nu(\phi_1) p^- + (1 - \nu(\phi_1)) p^+$ and we use the fact that $\frac{d\mathbb{H}(\mathrm{Ber}(p))}{dp} = \ln\left( \frac{1-p}{p} \right)$. Note that when $\nu(\phi_1) = \frac{1}{2}$ we have $m_1 = \frac{1}{2}$ and $G'(\frac{1}{2}) = 0$. Thus, $\frac{1}{2}$ is a stationary point. On the other hand, we have $G''(\frac{1}{2}) = 4(1 - s - 2(1-s)\Delta^2) \geq 0$ and $G(\nu(\phi_1)) = G(1 - \nu(\phi_1))$. This implies $\nu(\phi_1) = \frac{1}{2}$ is the unique minimizer and the minimal value is $G(\frac{1}{2}) = \ln(2)$.

Thus,

$$
\max_{\nu \in \Delta(\Psi)} \left\{ \mathbb{E}_{a \sim p_s} \mathbb{E}_{M \sim \nu} \left[ -V_M(a) - \frac{1}{\eta} \mathbb{E}_{o \sim M(a)} \left[ \mathrm{KL}(\nu_{\boldsymbol{\phi}}(\cdot|a, o), \rho_1) \right] - \frac{1}{\eta} \mathbb{E}_{\phi \sim \rho_1} [D^a(\phi \| M)] \right] \right\}
$$
$$
= \frac{s-1}{2} - \frac{s\epsilon}{2} - \frac{1}{\eta} (1-s) \left( -\mathbb{H}\left( \mathrm{Ber}(p^+) \right) + 2\Delta^2 \right) - \frac{1}{\eta} \ln(2)
$$
$$
= (1-s) \left( -\frac{1-\epsilon}{2} + \frac{\mathbb{H}(\mathrm{Ber}(p^+)) - 2\Delta^2}{\eta} \right) - \frac{\ln 2}{\eta} - \frac{\epsilon}{2}.
\tag{41}
$$

Note that

$$
\mathbb{H}(\mathrm{Ber}(p^+)) - 2\Delta^2
$$
$$
= -\mathrm{KL}(\mathrm{Ber}(p^+), \mathrm{Ber}(\tfrac{1}{2})) + \ln 2 - 2D_{\mathrm{TV}}^2(\mathrm{Ber}(p^+), \mathrm{Ber}(\tfrac{1}{2}))
$$
$$
\geq \ln 2 - 5\mathrm{KL}(\mathrm{Ber}(p^+), \mathrm{Ber}(\tfrac{1}{2})) \qquad \text{(Pinsker's inequality)}
$$
$$
\geq \ln 2 - 15\Delta^2 \qquad (\mathrm{KL}(\mathrm{Ber}(\tfrac{1}{2} + \Delta), \mathrm{Ber}(\tfrac{1}{2})) \leq 3\Delta^2 \text{ for } \Delta \leq \tfrac{1}{2})
$$
$$
\geq \frac{1}{2}. \qquad \text{(by the assumption } \Delta = \frac{1}{16\sqrt{T}} \leq \frac{1}{16})
$$

Hence, the minimum value of Eq. (41) is achieved at $s = 1$ when $\frac{1}{2\eta} - \frac{1-\epsilon}{2} \geq 0$. By the condition $\eta \leq 1$, this indeed holds. This means that our algorithm always picks the third arm in the first round. After picking arm $a_3$, the belief of $\phi$ will be deterministic, since $\nu_1(\phi|a_3, o) = 0$ for any $\phi \neq \phi^\star$. This means the algorithm will always choose the optimal action in the following rounds, ensuring that $\mathbb{E}\left[ \mathrm{Reg}(\pi_{M^\star}) \right] \leq p^+ < 1$.

## K    USE OF LARGE LANGUAGE MODELS IN PREPARATION

We did not use large language models at all for this project.

