# OpenReview forum: "An Improved Model-free Decision-estimation Coefficient with Applications in Adversarial MDPs"
_ICLR.cc/2026/Conference — ICLR 2026 Poster_

### Official Review · Reviewer_Z8zP · 2025-10-25

**Soundness:** 3
**Presentation:** 3
**Contribution:** 4
**Rating:** 8
**Confidence:** 2

**Summary:**

The author proposed the information-theoretical notion of complexity and a model-free algorithm for the general decision-making problem with structured observations (DMSO). The proposed Dig-DEC mechanism works seamlessly in both stochastic and adversarial environments (with stochastic transitions and adversarial rewards). It achieves the first model-free regret guarantees with bandit feedback in many general settings, resolving the existing open problem.  Additionally, the proposed approach improves the regret bound for Bellman-Complete MDPs to match classical optimism-based approaches.

**Strengths:**

- General framework that covers many existing RL settings and improves upon the existing optimistic DEC bounds.
- Guarantees for model-free algorithms in adversarial settings.
- Interesting online learning technique for the posterior update that seems to be of independent interest.

**Weaknesses:**

- Restrictive linear reward with known feature assumption for an adversarial hybrid setting;

**Questions:**

- What are typical values of $\log|\Phi|$ in the examples described in Section 5 (for example, in the case of worst-case finite MDPs)?
- How is DigDEC connected to a classical DEC in terms of regret lower bounds?

---

> ### Author Response · Authors · 2025-11-22
>
> We thank the reviewer for the insightful feedback. We have revised our manuscript and your questions are elaborated below.
>
>
> **Q1:** What are typical values of $\log|\Phi|$.
>
> **A:** We assume that $\mathcal{M}$ is the model-class, $\mathcal{F}$ is the value function class and $\Pi$ is the policy class. For model-based method in stochastic settings, $\log|\Phi| = \log|\mathcal{M}|$. For model-free method in stochastic settings, $\log|\Phi| = \log|\mathcal{F}|$. For model-based method in hybrid settings, $\log|\Phi| = \log|\mathcal{P}||\Pi|$. For model-free method in hybrid settings, $\log|\Phi| = \log|\mathcal{F}||\Pi|$. In all cases, the resulting $\log|\Phi|$ term matches the order obtained by previous methods specifically designed for each setting.
>
> **Q2:** How is DigDEC connected to a classical DEC in terms of regret lower bounds?
>
> **A:** In the stochastic setting, DigDEC recovers classical DEC with KL divergence if choosing $\overline{D}=0$ and $\Phi=\{(M, \pi_M), M\in\mathcal{M}\}$ (i.e., each partition corresponds to a single model). In general, DigDEC is larger than DEC because the two divergence terms in DigDEC measures the divergence between $\rho\in\Delta(\Phi)$ and $\nu\in\Delta(\mathcal{M})$, where $\Phi$ is a partition of $\mathcal{M}$, while the divergence term in DEC measures the divergence between $\rho\in\Delta(\mathcal{M})$ and $\nu\in\Delta(\mathcal{M})$. The latter divergence is generally larger, as the model distribution is more fine-grained than the partition distribution. Hence,
>
> $\mathrm{DigDEC}\_\eta$
>
> $= \max\_{\rho\in\Delta(\Phi)} \min\_{p\in\Delta(\Pi)} \max\_{\nu\in\Delta(\mathcal{M})}
>     \mathbb{E}\_{\phi\sim \rho} \mathbb{E}\_{\pi\sim p} \mathbb{E}\_{M\sim \nu}
>     \left[V_M(\pi_M) - V_M(\pi) - \frac{1}{\eta} D^\pi(\nu\|\rho)\right] $
>
> $\gtrsim \max\_{\rho\in\Delta(\mathcal{M})} \min\_{p\in\Delta(\Pi)} \max\_{\nu\in\Delta(\mathcal{M})}
>     \mathbb{E}\_{\phi\sim \rho} \mathbb{E}\_{\pi\sim p} \mathbb{E}\_{M\sim \nu}
>     \left[V_M(\pi_M) - V_M(\pi) - \frac{1}{\eta} D^\pi_{\text{KL}}(\nu\|\rho)\right] $
>
> $=\mathrm{DEC}^{\mathrm{KL}}\_\eta$
>
>
> where  $D^\pi(\nu\|\rho)$ is the sum of the two divergence terms in our Eq.(7), and the last equality is by Lemma 2.8 in [Xu and Leevi, 2023].
>
> As shown by [Foster et al., 2021], DEC characterizes the lower bound. Their upper bound is $T\cdot$DEC$\_\eta$ + $\frac{1}{\eta} \log|\mathcal{M}|$ while ours is $T\cdot$Dig-DEC$_\eta$ + $\frac{1}{\eta}\log|\Phi|$ and are thus incomparable.
>
> [Foster et al., 2021] Foster D J, Kakade S M, Qian J, et al. The statistical complexity of interactive decision making.
>
> [Liu et al., 2025] Liu H, Wei C Y, Zimmert J. Decision Making in Hybrid Environments: A Model Aggregation Approach. COLT, 2025.
>
> [Xu and Zeevi, 2023] Xu Y, Zeevi A. Bayesian design principles for frequentist sequential learning. ICML, 2023.

---

> > ### Comment · Reviewer_Z8zP · 2025-11-25
> >
> > I would like to thank the authors for their answer. All my questions were well answered, and I am happy to keep my positive score.

---

### Official Review · Reviewer_B9o3 · 2025-10-26

**Soundness:** 3
**Presentation:** 2
**Contribution:** 3
**Rating:** 6
**Confidence:** 3

**Summary:**

This paper positions itself as a bridge between two major approaches : model-based methods (which learn a full world model) and optimistic, model-free methods (which guess the best outcome). Its main contribution is a new, improved framework called Dig-DEC that removes the "optimism" mechanism. Instead of blindly chasing high rewards, Dig-DEC drives exploration  by seeking out information that helps it distinguish between different high-level "theories" about how the environment works. This shift leads to better performance guarantees in standard settings—improving regret bounds—under certain assumptions.
It aims at replacing a simple heuristic (optimism) with a  possibly more fundamental principle (targeted information gain) to solve harder problems more efficiently.

**Strengths:**

- the paper presents impressive theoretical developments (though I was far from being able to check all the details of the 30 pages of the appendix),
- the paper introduces Dig-DEC, a new complexity measure for decision-making. By removing the "optimism" principle and replacing it with pure "information gain," it provides an interesting exploration driver.
- The theoretical proof that Dig-DEC is always smaller than or equal to the prior optimistic DEC is a strong, clean result. This gives improved regret bounds but more fundamentally could give new perspectives of what can be done structured MDPs.

**Weaknesses:**

Implementation challenge
The first major challenge lies in the computational feasibility of solving the core minimax optimization (point 2 in the algorithm) at each round. This problem seems exceptionally difficult: the learner must optimize over distributions of policies against an adversary optimizing over distributions of models, with an objective function that involves nested expectations and KL divergences over trajectories. For any non-trivial state space, the policy and model classes are likely to be enormous, making an exact solution intractable. The objective is also non-convex and non-concave in general (?). While the paper provides a theoretical blueprint, it offers no practical implementation or approximation scheme. Bridging this gap would require major algorithmic innovations,


Assumptions
 The  core assumptions of the paper present a significant gap between the theory and the reality of most RL problems. The most restrictive assumption is possibly the requirement for a pre-defined, finite partition of the model space into infosets where all models within an infoset share a unique optimal policy and value function (Assumptions 1 & 3). In practice, such a discrete and perfectly aligned partition seems unavailable; the "optimal policy" may not be uniquely defined or may change during learning. Furthermore, Assumption 4 (linear rewards with known features) for the hybrid setting is a strong structural limitation, as it assumes the learner has perfect knowledge of the reward representation, which is often the very thing that needs to be learned in adversarial settings.

Presentation
A major weakness is the presentation. The paper is very difficult to read. Many notions are supposed known to the reader and there is no attempt to aim at a bit of self contained presentation.  For instance, the assumptions 5,6 are difficult to grasp. This makes the contribution of the paper less valuable because more difficult to gauge.
The presentation of the assumptions could be done differently by first describing informally the context and main restrictions and then making them mathematically precise.

**Questions:**

- What would be the complexity of the minimal AIR optimisation problem? Can you give approximate solutions at a reasonable complexity? (This is one of the most crucial point that the paper does not discuss...)

- Related to that last point, Im a bit lost with the "model free learning" terminology.  I guess for a large part of the community, model free means that you do not have access to the model ( and hence to means) but only to observations.
When solving the optimisation problem, you are not model free?
So,  given that you claim that optimism principles should be dropped, a lot more details should be given on how a practical estimation scheme can be leveraged  for your proposal?  The regret bounds are MDPs bound? not RL bounds?
In conclusion,  some precisions should be given on what you call (along with other papers) model free learning and more importantly what are the contours of your results...

- In Lemma 12, what does: "In the stochastic setting, Assumption 1 together with -completeness" mean?

- It would have been helpful to have a very simple toy example to underline that the assumptions are useful...

---

> ### Author Response · Authors · 2025-11-22
>
> We thank the reviewer for the insightful feedback. We have revised our manuscript and your questions are elaborated below.
>
> **Q1:** The algorithm is challenging to implement. The objective is also non-convex and non-concave in general (?)
>
> **A:** Our work focuses on the statistical complexity, which is already highly nontrivial. Resolving statistical and computational aspects simultaneously would require addressing two substantial challenges at once, which we believe exceeds what can be reasonably expected in a single work. We therefore leave the computational aspect to future study. We note, however, that this difficulty is common to many DEC-based approaches. That said, there are several bandit settings where DEC-based algorithms admit efficient implementations, including closed-form or tractable updates with strong empirical performance (e.g., [Foster et al., 2020], [Zhu et al., 2022], and Section 4.2 of [Xu et al., 2023]). Extending such efficient approaches to MDPs remains open and is an important direction.
>
> We note that the AIR objective is convex for the min-player in $\Delta(\Pi)$ and concave for the max-player in $\Delta(\Psi)$ (which is $\Delta(\mathcal{M})$ in the stochastic setting and $\Delta(\mathcal{M}\times \Pi)$ in the hybrid setting). Therefore, if the decision space is small, we can also directly apply standard algorithms for convex-concave saddle-point problem.  Suppose that given any decision of the min-player, the max-player can find its best response with a computational time of $C$.  Then the approximate saddle-point algorithm in [Arora et al., 2012] can find an $\epsilon$-approximate minimax solution within $\frac{|\Pi|\ln(|\Pi|)}{\epsilon^2}\times C$ time.
>
> **Q2:** The most restrictive assumption is possibly the requirement for a pre-defined, finite partition of the model space into infosets where all models within an infoset share a unique optimal policy and value function (Assumptions 1 \& 3). In practice, such a discrete and perfectly aligned partition seems unavailable; the "optimal policy" may not be uniquely defined or may change during learning.
>
> **A:**  We would like emphasize that while our paper uses a slightly different language (to fit in the DEC framework), our assumptions are aligned with those in prior model-free work (e.g., [Jin et al., 2021; Du et al., 2021]) and pose no additional restrictions. In fact, Assumption 1-3 act more like "definitions" than assumptions. Please also see our response to **Q1** of Reviewer L37f.
>
> In the stochastic setting, the learner only needs access to a value-function class (like [Jin et al., 2021; Du et al., 2021]). The model class, as well as the partition over the model class, are *induced* by the value-function class and constructed *internally* by the learner. As a result, it is always “perfect’’ by construction: two models belong to the same group only if they agree on the value function. Importantly, the learner never relies on any external access to perfect partition. Each value function $f$ and the corresponding greedy policy $\pi_f(s)=\arg\max_a f(s,a)$ (with arbitrary tie-breaking) are fixed over time, exactly same as in standard value-based formulations [Jin et al., 2021; Du et al., 2021].
>
> In the hybrid case, our formulation is also aligned with prior work in hybrid MDPs. For each partition $\phi_{\pi,f}$, the associated policy $\pi$ is not required to be the optimal policy for models in the group. Instead, it serves as a *fixed-over-time regret comparator*: the learner's performance is measured against a single fixed benchmark policy, even though the environment may change over time. This is the standard notion of regret in adversarial learning, and our use of $\pi$ plays exactly this role.
>
> Finally, our work do not require $|\mathcal{M}|$ to be finite. The finiteness of $|\mathcal{F}|$ and $|\Pi|$ is assumed for simplicity, as in prior work on model-free learning (e.g., [Jin et al., 2021], [Du et al., 2021]). The analysis can be extended to infinite classes using covering-number arguments (see Section 3.2.3 in [Foster et al., 2021]).

---

> ### Author Response · Authors · 2025-11-22
>
> **Q3:** Assumption 4 (linear rewards with known features) for the hybrid setting is a strong structural limitation, as it assumes the learner has perfect knowledge of the reward representation, which is often the very thing that needs to be learned in adversarial settings.
>
> **A:**  We agree that removing the assumption of known reward features in the hybrid setting is an important open problem. However, even the case with known reward feature is already technically challenging (as we deal with general transition structure), and this assumption is adopted in almost all prior work on bandit-feedback hybrid MDPs, with the exception of [Liu et al., 2024]. We also note that [Liu et al., 2024]'s approach relies on an explore-then-commit strategy to learn features, which is less desirable and potentially not generalizable beyond low-rank MDP. Therefore, we intentionally avoid such approach. We have added Footnote 2 on Page 5 to clarify this point.
>
>
> **Q4:** The assumptions 5,6 are difficult to grasp.
>
> **A:** Assumption 5 corresponds to standard realizability assumption, and Assumption 6 corresponds to realizability + Bellman-completeness assumption. The "estimation function" there can be simply regarded as average Bellman error and squared Bellman error (see our Lemma 8 and Lemma 12 for more details), which are two standard loss functions used in model-free training. Calling them ``estimation function'' is just a more general way to refer to them, as in [Du et al., 2021]. We have added more explanations in the revised manuscript.
>
> **Q5:** Im a bit lost with the "model free learning" terminology. When solving the optimisation problem, you are not model free?
>
> **A:**  Following [Foster et al. 2023], the main difference between model-based and model-free in the discussion of statistical complexity is whether the estimation error scale with $\log|\mathcal{M}|$ or  $\log|\mathcal{F}|$.  As explained in our response to **Q2**, our approach does not require the learner to access a model class --- although our optimization takes a maximum over a model class,  this model class can be constructed internally by the learner from the value function class $\mathcal{F}$  by setting $\mathcal{M} = \\{M \mid Q_M^\star \in \mathcal{F}\\}$ (also see the discussion in [Foster et al., 2023] after their Assumption 2.2). Overall, the precise distinction between model-based and model-free learning in our framework is whether the statistical complexity scales with the complexity of the model class or only that of the value function class / policy class. This is discussed at the end of Section 1.
>
>
> **Q6:**    The regret bounds are MDPs bound? not RL bounds?
>
> **A:** We would appreciate if the reviewer could elaborate more on the definitions and distinctions of MDPs bound and RL bounds.  The regret notion we use conforms with standard ones for stochastic and hybrid MDPs in prior work.
>
> **Q7:** In Lemma 12, what does: "In the stochastic setting, Assumption 1 together with -completeness" mean?
>
> **A:**  This is a typo and "-completenes" should be ``Bellman-completeness''. We have fixed it in the latest version.
>
> [Arora et al., 2012]  The Multiplicative Weights Update Method: a Meta Algorithm and Applications.
>
> [Jin et al., 2021]  Bellman Eluder Dimension: New Rich Classes of RL Problems, and Sample-Efficient Algorithms.
>
> [Du et al., 2021]  Bilinear Classes: A Structural Framework for Provable Generalization in RL.
>
> [Foster et al., 2021] Foster D J, Kakade S M, Qian J, et al. The statistical complexity of interactive decision making.
>
> [Foster et al., 2023] Foster D J, Golowich N, Qian J, et al. Model-free reinforcement learning with the decision-estimation coefficient. NeurIPS, 2023.
>
> [Dai et al., 2023] Dai Y, Luo H, Wei C Y, et al. Refined regret for adversarial mdps with linear function approximation.  ICML, 2023.
>
> [Luo et al., 2021] Luo H, Wei C Y, Lee C W. Policy optimization in adversarial mdps: Improved exploration via dilated bonuses.  NeurIPS, 2021.
>
> [Liu et al., 2024] Liu H, Wei C Y, Zimmert J. Towards optimal regret in adversarial linear mdps with bandit feedback. ICLR, 2024.
>
> [Lattimore and Szepesvári, 2020]  Lattimore T, Szepesvári C. Bandit algorithms. 2020.
>
> [Foster et al., 2020] Foster D, Rakhlin A. Beyond ucb: Optimal and efficient contextual bandits with regression oracles. ICML, 2020.
>
> [Zhu et al., 2022] Zhu Y, Foster D J, Langford J, et al. Contextual bandits with large action spaces: Made practical. ICML, 2022.
>
> [Xu et al., 2023] Xu Y, Zeevi A. Bayesian design principles for frequentist sequential learning. ICML, 2023.

---

### Official Review · Reviewer_L37f · 2025-11-01

**Soundness:** 3
**Presentation:** 2
**Contribution:** 4
**Rating:** 8
**Confidence:** 2

**Summary:**

This paper introduces a new, model-free complexity measure Dig-DEC (Dual Information Gain Decision-Estimation Coefficient), which removes the optimism-based exploration mechanism (allows adversal setting ) and is upper bounded by optimistic DEC. Then, this work adopts this framework to the stochastic and hybrid adversarial MDP setting and achieves a series of SOTA results with different MDP structures (bilinear classes, Bellman-complete, etc.).

**Strengths:**

* This work achieves a series of improved results under the Dig-DEC framework.

**Weaknesses:**

* Since this work has provided a series of new SOTA results, it would be very helpful to add some discussion and intution after each results or assumption (for example, Assumption 1).

**Questions:**

Q1: Can this work provide experiments (even simulation experiments) to show the relationship between Dig-DEC and optimistic DEC.

---

> ### Author Response · Authors · 2025-11-22
>
> We thank the reviewer for the insightful feedback. We have revised our manuscript and your questions are elaborated below.
>
> **Q1:** Since this work has provided a series of new SOTA results, it would be very helpful to add some discussion and intution after each results or assumption (for example, Assumption 1).
>
> **A:** We have revised the paper to provide clearer intuition. First, we emphasize that although our assumptions are stated in a more abstract way to fit in the DEC framework, they are natural generalizations or restatements of standard assumptions.
>
>
> Take the stochastic setting as an example. In standard model-free value-based learning, the learner is given a value-function class. As many different underlying MDPs (transition and reward models) can induce the same value function, we can partition the models according to the value function they induce. This is exactly what Assumption 1 wants to formalize. It is in fact more like a ``definition'' describing how to define the partition, and does not set additional restrictions than in prior work.
>
> Assumptions 2 and 3 extend this idea to the more involved hybrid setting. They similarly describe a natural partition over the model-policy space, and conform with existing model-free learning paradigms in hybrid MDPs, stated in a more abstract form to fit our DEC framework.
>
> Assumption 4 imposes a linear structure with known features in the hybrid setting. This is a standard assumption in the literature on linear MDPs and linear bandits.
>
> Assumption 5 corresponds to standard realizability assumption, and Assumption 6 corresponds to realizability + Bellman-completeness assumption. The "estimation function" in Assumption 5 can be simply regarded as the average Bellman error and in Assumption 6 the squared Bellman error (see our Lemma 8 and Lemma 12 for more details). These are two standard surrogate loss functions used in model-free training. Calling them "estimation function" is just a more general way to refer to them, as in [Du et al., 2021].
>
> We have added more explanations on these assumptions in the revised manuscript.
>
> **Q2:** Can this work provide experiments (even simulation experiments) to show the relationship between Dig-DEC and optimistic DEC?
>
> **A:** Thanks for the suggestion. Inspired by Xu and Zeevi (2023), we believe it may be possible to simulate both algorithms in simple settings such as Bernoulli bandits. Our case is slightly more involved, however, since our objective involves two divergence terms, so obtaining a clean closed-form solution may require additional work. Given the tight rebuttal timeline, it is unclear whether we can complete this investigation during the discussion phase, but if we succeed in resolving the technical difficulties, we will incorporate the experiments in the final version.
>
> [Du et al., 2021] Bilinear Classes: A Structural Framework for Provable Generalization in RL.
>
> [Xu et al., 2023] Xu Y, Zeevi A. Bayesian design principles for frequentist sequential learning. ICML, 2023.

---

### Official Review · Reviewer_NZXe · 2025-11-01

**Soundness:** 3
**Presentation:** 3
**Contribution:** 4
**Rating:** 8
**Confidence:** 4

**Summary:**

This work proposes a new decision-estimation coefficient (DEC) notion, which enables conducting exploration via information gain instead of the optimism principle in a model-free manner. The first benefit of the new DEC notion is the improvement of previous results in cases of bilinear classes or Bellman-complete MDPs with bounded Bellman eluder dimension. The second benefit is that this new DEC notion leads to the first model-free algorithm for MDPs with stochastic transitions and adversarial loss functions in the bandit feedback setting.

**Strengths:**

1. **Novelty**: The proposed new notion seems interesting and fundamental, enabling the exploration solely based on information gain.
2. **Results**: The model-free algorithm leads to a series of new results in stochastic MDPs, with matched or even improved results. This work also resolves the open problem of previous work for solving MDPs with stochastic transitions and adversarial loss functions in the bandit feedback setting.
3. **Presentation**: This paper is well-written.

**Weaknesses:**

If any, I would feel that some parts could be clearer and more thoroughly explained. For instance, from Table 1, I notice that the regret bound of off-policy exploration might be inferior to that of on-policy exploration. Could the authors explain why this happens?

Also, most previous works for tabular and linear MDPs with adversarial loss functions use occupancy measure (OM)-based or policy optimization (PO)-based methods, both of which are model-based and require to learn the transitions explicitly to construct the loss estimator. Could the authors intuitively explain how to construct the “loss estimator” without explicit learning of transitions?

**Questions:**

Please see my questions in the weakness part.

---

> ### Author Response · Authors · 2025-11-22
>
> We thank the reviewer for the insightful feedback. We have revised our manuscript and your questions are elaborated below.
>
> **Q1:** From Table 1, I notice that the regret bound for off-policy exploration appears to be worse than that for on-policy exploration. Could the authors explain why this happens?
>
> **A:** The "on-policy" and "off-policy" here should not be confused with the on-policy and off-policy RL commonly used in the community. Our “on-policy’’ and “off-policy’’ refer to two subclasses of bilinear class [Du et al., 2021] or Bellman-eluder dimension [Jin et al., 2021] that require different ways to measure the mismatch between a function and the ground-truth model. We added more explanation in the manuscript. One can also find a concise description of them in Section~4.1 of [Du et al., 2021].
>
> The high level reason for the difference in regret bound is that in the on-policy bilinear subclass, the policy induced by the belief ($\pi_{\phi}$ with $\phi \sim \rho_t$) already provides sufficient exploration to notice a model mismatch, while for the off-policy cases, the learner needs to execute an additional exploration policy for this purpose. Technically, in the on-policy case, the learner can simply execute the policy induced by the belief, which is sufficient to control Dig-DEC complexity to  $O(\eta)$ as shown in Lemma 33. However, in the off-policy case, the learner is required to execute a mixture policy that combines the belief-induced policy with the additional uniform exploration over actions. Such a mixture leads to $O(\sqrt{\eta})$ order for the Dig-DEC and thus leads to a worse regret.
>
>
> **Q2:** Most previous works on tabular and linear MDPs with adversarial loss functions use occupancy-measure (OM)–based or policy-optimization (PO)–based methods. Both are model-based and require explicit transition estimation to construct a loss estimator. Could the authors intuitively explain how to construct the "loss estimator" without explicitly learning the transitions?
>
> **A:**  It is indeed a standard approach in adversarial bandits/MDPs to create loss estimators. However, inspired by the Exploration-by-Optimization approach [Lattimore and Szepesvari, 2019], recent literature on general adversarial decision making [Foster et al., 2022; Xu and Zeevi, 2023, Liu et al., 2025] has developed a shortcut that bypasses the explicit construction of loss estimators. In fact, the standard loss estimators can be thought of as a mapping from observation to policy update, which specifies ``how should the policy distribution be updated based on the observation.'' In the approach of [Foster et al., 2022; Xu and Zeevi, 2023, Liu et al., 2025] and our work, such mapping is directly incorporated in the min-max optimization problem (specifically, the $\nu_\phi(\cdot|\pi, o)$ term in our AIR objective) in an end-to-end manner, without explicitly constructed. While such an optimization approach is computationally more involved, it gives a general and conceptually simpler algorithm when the explicit reward estimator construction might be complicated.
>
>
> In certain special cases, such as contextual bandits with a finite action space and linear rewards, the corresponding minimax optimization problem admits a closed-form solution. In these cases, the optimal solution can be related to certain loss estimators ((see [Foster et al., 2020] and its practical variant [Zhu et al., 2022], as well as Section 4.2 and Appendix C.2 of [Xu et al., 2023]).
>
> [Lattimore and Szepesvari, 2019] Exploration by Optimisation in Partial Monitoring, 2019.
>
> [Foster et al., 2022] Foster D J, Rakhlin A, Sekhari A, et al. On the complexity of adversarial decision making. NeurIPS, 2022.
>
> [Foster et al., 2023] Foster D J, Golowich N, Qian J, et al. Model-free reinforcement learning with the decision-estimation coefficient. NeurIPS, 2023.
>
> [Jin et al., 2021] Bellman Eluder Dimension: New Rich Classes of RL Problems, and Sample-Efficient Algorithms.
>
> [Du et al., 2021] Bilinear Classes: A Structural Framework for Provable Generalization in RL.
>
> [Liu et al., 2025] Liu H, Wei C Y, Zimmert J. Decision Making in Hybrid Environments: A Model Aggregation Approach. COLT, 2025.
>
> [Foster et al., 2020] Foster D, Rakhlin A. Beyond ucb: Optimal and efficient contextual bandits with regression oracles. ICML, 2020.
>
> [Zhu et al., 2022] Zhu Y, Foster D J, Langford J, et al. Contextual bandits with large action spaces: Made practical. ICML, 2022.
>
> [Xu et al., 2023] Xu Y, Zeevi A. Bayesian design principles for frequentist sequential learning. ICML, 2023.

---

### Meta-Review · Area_Chair_61rP · 2026-01-03

**Summary:**

This paper studies model-free DEC methods for reward-adversarial MDPs. The authors establish the first model-free regret bounds with bandit feedback in several important settings, including bilinear function classes and Bellman-complete MDPs with bounded Bellman–Eluder dimension. All reviewers agree that the paper resolves a significant open problem. Therefore, I would like to recommend acceptance.

**Reviewer Concerns:**

The rebuttal addresses some concerns regarding the problem setting. However, several issues remain. The major concern raised by the reviewer is about the practical implementation of the algorithm. First, the core minimax optimization required at each round is computationally intractable in practice; second, the theoretical assumptions are strong and potentially unrealistic. A minor concern is about the writings. The paper still requires effort to achieve a self-contained presentation.

**Reviewer Scores:**

I expect the reviewers to maintain their scores after fully engaging in the discussion.

---

### Decision · Program_Chairs · 2026-01-26

Accept (Poster)